# SUMOylated SNF2PH promotes variant surface glycoprotein expression in bloodstream trypanosomes

Andreu Saura[1], Paula A Iribarren[2], Domingo Rojas-Barros[1], Jean M Bart[1], Diana López-Farfán[1], Eduardo Andrés-León[1], Isabel Vidal-Cobo[1], Cordula Boehm[3], Vanina E Alvarez[2], Mark C Field[3,4] & Miguel Navarro[1,*] (ID)

## Abstract

**SUMOylation is a post-translational modification that positively regulates monoallelic expression of the trypanosome variant surface glycoprotein (VSG). The presence of a highly SUMOylated focus associated with the nuclear body, where the VSG gene is transcribed, further suggests an important role of SUMOylation in regulating VSG expression. Here, we show that SNF2PH, a SUMOylated plant homeodomain (PH)-transcription factor, is upregulated in the bloodstream form of the parasite and enriched at the active VSG telomere. SUMOylation promotes the recruitment of SNF2PH to the VSG promoter, where it is required to maintain RNA polymerase I and thus to regulate VSG transcript levels. Further, ectopic overexpression of SNF2PH in insect forms, but not of a mutant lacking the PH domain, induces the expression of bloodstream stage-specific surface proteins. These data suggest that SNF2PH SUMOylation positively regulates VSG monoallelic transcription, while the PH domain is required for the expression of bloodstream-specific surface proteins. Thus, SNF2PH functions as a positive activator, linking expression of infective form surface proteins and VSG regulation, thereby acting as a major regulator of pathogenicity.**

**Keywords** antigenic variation; plant homeodomain; post-translational modification; SUMO; variant surface glycoprotein
**Subject Categories** Chromatin, Transcription & Genomics; Microbiology, Virology & Host Pathogen Interaction; Post-translational Modifications & Proteolysis

## Introduction

Antigenic variation, the major mechanism by which African trypanosomes evade the host immune response, is mediated by switching expression between immunologically distinct variant surface glycoprotein (VSG) genes [1]. The active VSG gene is transcribed polycistronically by RNA polymerase I, together with several expression site-associated genes (ESAGs), from a large telomeric locus (40–60 kb), known as a VSG expression site (VSG-ES), currently named bloodstream ES (BESs) [2]. In the mammalian bloodstream form (BF), where antigenic variation occurs, only one of ~15 VSG-ES genes is transcribed at a given time, resulting in monoallelic expression and a dense surface coat comprised of a single VSG type [3–5]. The active VSG-ES is recruited to a nuclear compartment, the expression site body (ESB), which facilitates monoallelic transcription [6–8]. Interestingly, small ubiquitin-like modifier (SUMO) post-transcriptionally modified proteins are associated with the ESB within a highly SUMOylated focus (HSF) [9]. However, in the insect or procyclic form, VSGs are not expressed and procyclin glycoproteins cover the parasite surface [10].

SUMOylation is a large and reversible post-translational modification (PTM) that regulates many critical processes, including transcription, protein–protein interactions, protein stability, nuclear localization, DNA repair, and signaling [11]. In *Trypanosoma brucei*, there is a single SUMO ortholog, which is essential for cell cycle progression of the procyclic form [12]. Proteomic analyses of SUMO substrates in this life stage identified 45 proteins involved in multiple cellular processes, including epigenetic regulation of gene expression [13]. Transcription factors are well known SUMO targets, whose activity can be modulated in both gene silencing and activation [14]. In *T. brucei* BF, SUMO-conjugated proteins were detected highly enriched in the nucleus in a single focus (HSF) associated with the ES body (ESB) and in the active VSG-ES chromatin, suggesting chromatin SUMOylation acts as a positive epigenetic mark to regulate VSG expression [9]. Chromatin SUMOylation to the active VSG-ES locus is required for efficient recruitment of RNA polymerase I in a SUMO E3 ligase (TbSIZ1/PIAS)-dependent manner, suggesting protein SUMOylation facilitates the accessibility of additional transcription factors [9].

1 Instituto de Parasitología y Biomedicina "López-Neyra", CSIC (IPBLN-CSIC), Granada, Spain
2 IIB-UNSAM, Buenos Aires, Argentina
3 School of Life Sciences, University of Dundee, Dundee, UK
4 Biology Centre, Institute of Parasitology, Czech Academy of Sciences, Ceske Budejovice, Czech Republic
*Corresponding author. Tel: +34 958181651; Fax: +34 958181633; E-mail: miguel.navarro@ipb.csic.es

Thus, we sought to identify major SUMO-conjugated proteins in the mammalian infective form and found a novel protein annotated as a transcription activator in the database (Tb927.3.2140). Structural conserved domain predictions suggest that Tb927.3.2140 is a member of the Snf2 (Sucrose Nonfermenting Protein 2) SF2 helicase-like superfamily 2 of chromatin remodelers [15–17] and also contains a plant homeodomain (PHD). Thus, we designate the protein SNF2PH.

Here, we show that SNF2PH is a developmentally regulated protein enriched at chromatin of the VSG-ES (BES) telomere, particularly at promoter regions when modified by SUMO. SNF2PH depletion leads to reduced VSG transcription and upregulation of developmental markers for the insect stage. ChIP-seq data suggest SNF2PH binds to selective regions in chromatin, in addition to the active VSG-ES, like developmentally regulated loci, rDNA, SL-RNA, and, interestingly, also to clusters of tRNA genes, which function as insulators for repressed and active chromatin domains in other eukaryotes. SNF2PH is strongly downregulated in quiescent (pre-adapted to host transition) trypanosomes generated in both pleomorphic (differentiation-competent) and monomorphic (by AMPKα1-activation) strains. Further, SNF2PH expression is negatively regulated in the insect procyclic form. Most importantly, overexpression of SNF2PH in the insect form triggers the expression of bloodstream stage-specific surface protein genes, suggesting a role as positive regulator of differentiation. Thus, SNF2PH links immune evasion with pathogenicity.

# Results

## Trypanosome SNF2PH is SNF2_N-related protein that contains an unusual plant homeodomain

SUMOylation is a hallmark of epigenetic VSG regulation at the level of chromatin and nuclear architecture [9]. The highly SUMOylated focus (HSF) detected by a specific mAb against TbSUMO in the nucleus of bloodstream form (BF) trypanosomes was recently associated with the nuclear body ESB [9], the site for VSG-ES monoallelic expression [6]. Recognition of HSF together with the detection of highly SUMOylated proteins at the active VSG-ES chromatin by ChIP analysis suggests that a number of SUMOylated proteins are mechanistically involved in regulation of VSG expression [9]. Therefore, identifying these proteins is a novel approach for the discovery of factors involved in VSG regulation. To identify abundant SUMOylated proteins, we performed a non-exhaustive proteomic analysis utilizing BF protein extracts from a cell line expressing an 8His-HA-tagged SUMO (see Materials and Methods). LC-MS/MS analyses of His-HA-affinity-purified extracts robustly identified several proteins (see Appendix Table S1). Particularly, interesting was Tb927.3.2140 (length 948 aa), a protein annotated in the TriTrypDB database as a transcription activator, which contains a conserved SNF2 family N-terminal domain.

Comparative analyses of Tb927.3.2140 at CDART [18] and the NCBI CDD domain database identified three conserved domains: DEXHc_Snf, e-value $9.4e^{-74}$, SF2_C_SNF, e-value $8.0e^{-50}$, PHD5_NSD, e-value $6.2e^{-14}$. Structural CD predictions suggest than Tb927.3.2140 is a member of the Snf2 family (Sucrose Nonfermenting Protein 2) from the SF2 helicase-like superfamily 2

of chromatin remodelers [15–17], which regulate DNA accessibility to facilitate central cellular processes as transcription, DNA repair, DNA replication and cell differentiation [15,16]. Next, searching for Tb927.3.2140 homologues using DELTA-BLAST against UniProtKB/SwissProt database, identified a protein member of the SWI/SNF family, SMARCA1 (e-value, $4e^{-157}$) (SWI/SNF-related matrix-associated actin-dependent regulator of chromatin subfamily A member 1) also known as the global transcription activator SNF2L1 (length, 1054 aa) (homonyms SWI; ISWI; SWI2; SNF2L; SNF2L1; SNF2LB; SNF2LT; hSNF2L; NURF140) all described to be involved in transcription for either gene activation or gene repression [16]. In addition to the SNF2 N domain, a conserved helicase C-terminal domain was also detected, known to function as a chaperon-like in the assembly of protein complexes. Interestingly, Tb927.3.2140 also contains a plant homeodomain (PHD) that is absent from other known trypanosome chromatin remodelers. The PHD is a conserved homeodomain involved in development [19] that binds H3 tails and reads unmodified H3 tails [20] as well as H3 trimethylated at Lys4 (H3K4me3) [21] or acetylated at Lys8 and Lys14 [22,23]. The PH domain is conserved in histone methyltransferases, including murine HMT3 and human NSD3 (Appendix Fig S1). Thus, we named this chromatin-remodeling factor trypanosome SNF2PH.

## SNF2PH is developmentally regulated and associated with the ESB nuclear body

In order to investigate SNF2PH protein expression, we raised a monoclonal antibody (mAb) (11C10E4) against the recombinant protein expressed in bacteria. Western blot of total protein extracts and immunofluorescence (IF) showed that SNF2PH protein levels are developmentally regulated, with higher expression in the bloodstream compared to the insect form (Fig 1A and Appendix Fig S2). The specificity of mAb11C10E4 was demonstrated as SNF2PH levels in whole cell extracts were markedly reduced in RNAi cells (Fig 1B).

Subcellular localization of SNF2PH by 3D-deconvolution IF (3D-IF) microscopy with mAb11C10E4 showed a nuclear localization, with disperse distribution in the nucleoplasm, including puncta and enrichment at the nucleolar periphery (Fig 1C). To determine whether SNF2PH associates with the active VSG-ES locus, we used a GFP-lacI targeted VSG-promoter cell line [6]. We detected 38.8% (n = 55) colocalization (Pearson's correlation coefficient) with GFP-tagged active VSG-ES using an anti-GFP rabbit antiserum and mAb11C10E4 anti-SNF2PH (Fig 1D). Statistical analysis showed even lower association in 2K1N cells (S-G2) (Fig EV1A), suggesting that SNF2PH association with the active VSG-ES locus occurs in a cell cycle-dependent manner.

To investigate the association with the HSF, we stained cells with anti-TbSUMO mAb [9] and anti-SNF2PH antiserum (Materials and Methods). 3D-IF showed colocalization between SNF2PH and HSF in 53.7% of the cells (n = 67)) (Figs 1E and EV1B), likely due to the highly dynamic nature of protein SUMOylation. Similar colocalization in 53.49% of the cells was observed between SNF2PH and the ESB (Figs 1F and EV1C), indicated by extranucleolar pol I signal visualized with a YFP-tagged RPB5z (specific RNA pol I subunit 5z [24]) (Figs 1F and EV1C).

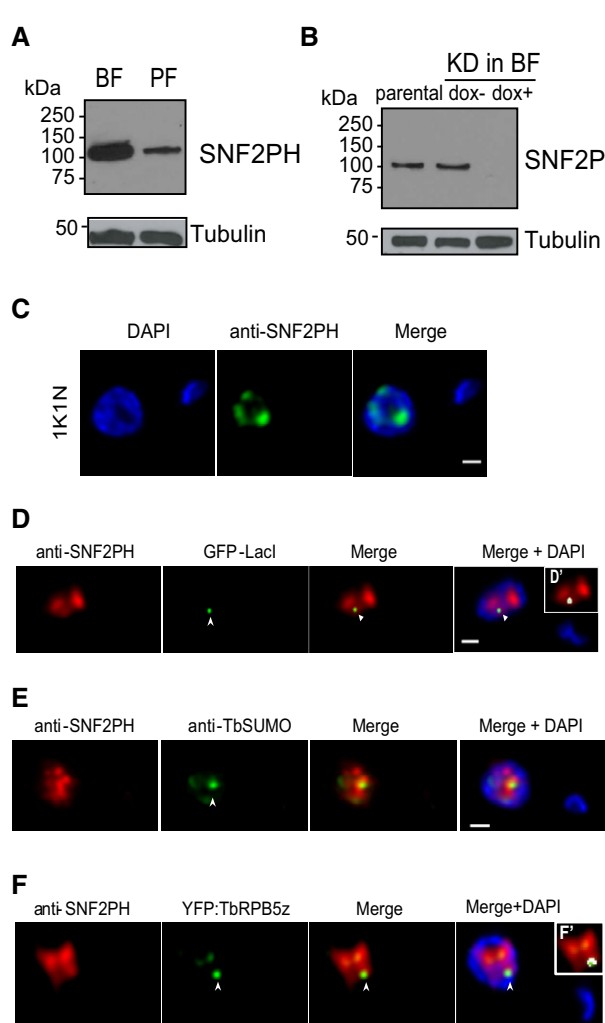

**Figure 1. SNF2PH is a developmentally regulated protein associated with the expression site body (ESB) and highly SUMOylated focus (HSF).**

A SNF2PH is differentially expressed in *T. brucei* developmental stages. The mammalian infective form, bloodstream form (BF), and the insect form, procyclic form (PF).

B Knockdown of SNF2PH by inducible RNA interference in bloodstream form leads to protein depletion after 24 h. (5 × 10⁶ cells/lane): parental, uninduced (dox⁻), and induced (dox⁺). Total cell extracts were analyzed by Western blotting using the anti-SNF2PH mAb.

C SNF2PH is diffusely distributed in the nucleoplasm with certain enrichment in the nucleolus. Panels show DAPI and green channels after IF with the anti-SNF2PH mAb (11C10E4). Scale bar, 1 μm.

D SNF2PH associates with the active VSG-ES. A cell line with a GFP-LacI tag in the active VSG-ES [6] was subjected to double 3D-IF using anti-SNF2PH mAb (red), anti-GFP rabbit antiserum (green) and DAPI staining. Maximum intensity projections of deconvolved slices containing the GFP signal are shown (arrowhead). (D') Inset shows a higher magnification of the nucleus including anti-SNFPH and anti-GFP fluorescence signals colocalization mask (white). Scale bar, 1 μm.

E Colocalization analysis of SNF2PH with the Highly SUMOylated focus (HSF). SNF2PH associates with the HSF (arrowhead) in bloodstream form nucleus. Indirect 3D-IF analyses were carried out using the rabbit anti-SNF2PH antiserum (red) and the anti-TbSUMO mAb 1C9H8 (green) [9]. Scale bar, 1 μm.

F CSNF2PH partially colocalizes with YFP-tagged TbRPB5z in the ESB. A cell line expressing an N-terminal fusion of a Yellow Fluorescent Protein (YFG) with the RNA pol I-specific subunit RPB5z described previously [24] was used to analyze by double 3D-IF a possible association of SNF2PH with the ESB. The 3D-IF was performed using the anti-SNF2PH mAb (red) and rabbit anti-GFP antiserum (green) that recognizes the Yellow GFP variant. Deconvolved slices containing both SNF2PH and the extranucleolar ESB (arrowhead) are represented as maximum intensity projections. (F') Inset shows a higher magnification of the nucleus including anti-SNF2PH and anti-GFP fluorescence signals colocalization mask (white). Scale bar, 1 μm.

nucleolus and nuclear periphery in one (84.12% ± 0.25%) or two puncta (15.88% ± 0.25%) (Appendix Fig S3).

The low signal of SNF2PH antibody in TbSUMO IP experiments is likely a consequence of the dynamic nature of SUMOylation, yielding a small population of SUMOylated SNF2PH form at any given time; similar behavior has been demonstrated for TbRPAI (RNA Polymerase I largest subunit) [9] and additional SUMO proteins in *T. brucei* [25]. To determine which domains of SNF2PH are SUMOylated, we used an *E. coli* strain expressing the complete *T. brucei* SUMOylation system [13]. We evaluated two different constructs encompassing the SNF2PH N-terminal or C-terminal domain (SNF2PH-N and SNF2PH-C, respectively), bearing a Flag tag. We co-expressed SNF2PH-N and SNF2PH-C in *E. coli* with TbSUMO (already exposing the diGly motif) and both activating enzyme subunits (TbE1a/TbE1b) plus the conjugating enzyme (TbE2). SNF2PHN appears as a single band migrating at the expected position when expressed alone in *E. coli* (Fig 2C, lane 1) or when co-expressed with a partially reconstituted system (lanes 2 and 3). However, when co-expressed with the complete SUMOylation system, additional slower-migrating bands can be detected (lane 4), suggesting that the N-terminal domain of SNF2PH can be SUMO conjugated. In contrast, the C-terminal domain is not a target of SUMOylation since it is only detected as a single protein band at the expected position of the unmodified protein (Fig 2D).

To confirm heterologous SUMOylation of SNF2PHN, we performed *in vitro* deconjugation reactions using the specific *T. brucei* SUMO isopeptidase TbSENP. As shown in Fig 2E, the

## SNF2PH is SUMO conjugated

To investigate whether SNF2PH is a *bona fide* SUMO-modified protein, we carried out immunoprecipitation (IP) utilizing the anti-TbSUMO mAb [9] and anti-SNF2PH antiserum under denaturing conditions to capture only proteins with covalent SUMO modifications. IP suggested that SNF2PH is SUMOylated when analyzed by Western blotting using the anti-TbSUMO mAb on a SNF2PH immunoprecipitate (Fig 2A). The reciprocal experiment, using anti-SNF2PH antiserum on a TbSUMO immunoprecipitate reproducibly detected SNF2PH conjugated to TbSUMO (Fig 2B).

While IP demonstrates that SNF2PH is SUMOylated, it is unknown whether nuclear conjugation with SUMO is associated with dispersed nuclear foci or localization to a specific subnuclear site. We performed Proximity Ligation assays (PLA) (O-link Bioscience), an IF method where a signal is produced only if two proteins, or a protein and its PTM, are within 40 nm. After a first IF experiment using anti-SUMO mAb and the SNF2PH antiserum, secondary species-specific antibodies conjugated with oligonucleotides are hybridized to the two PLA probes to produce a DNA by rolling circle replication. As the PLA assay detected positive amplification this suggests that SNF2PH is SUMOylated *in situ* in both the

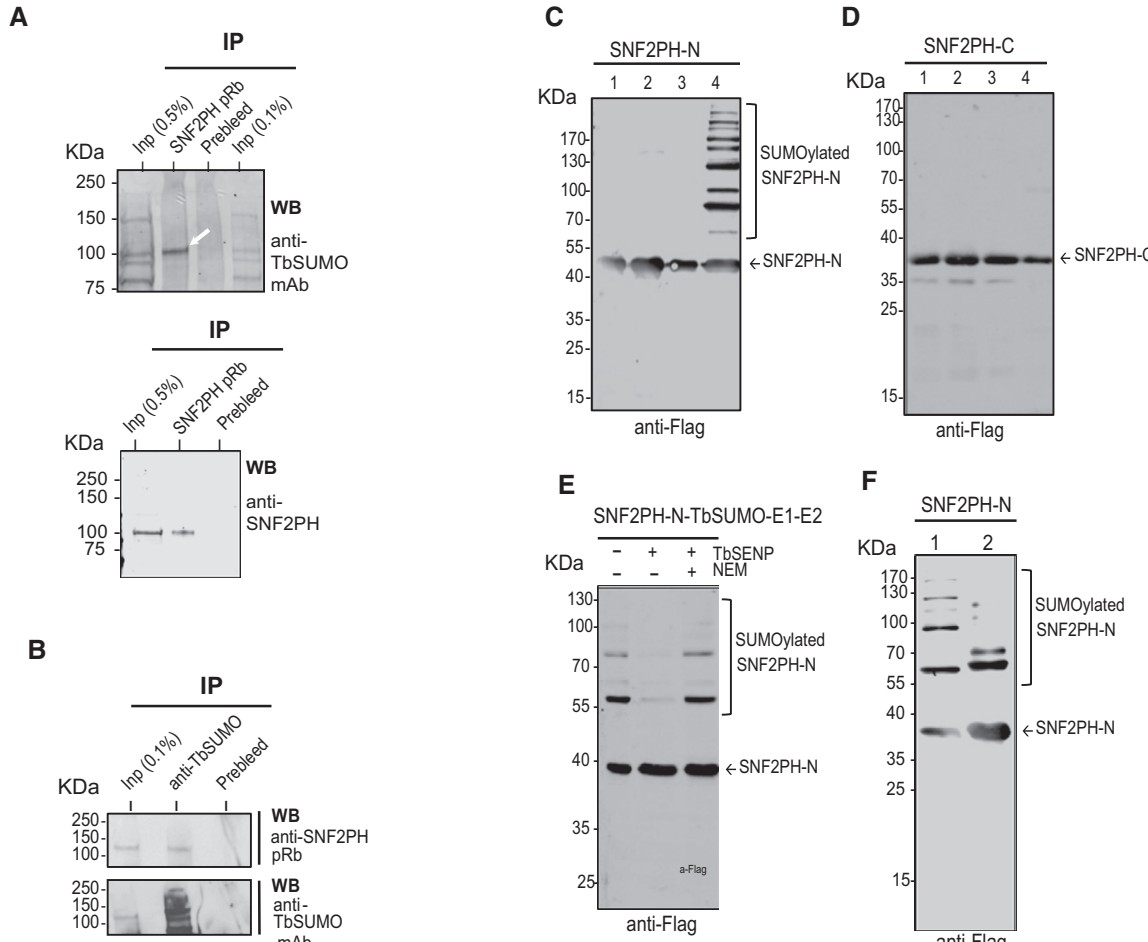

**Figure 2. SNF2PH is SUMOylated *in vivo* in trypanosomes and *in vitro* using a SUMOylation heterologous system.**

A  Immunoprecipitation (IP) of bloodstream SUMOylated proteins revealed that SNF2PH is SUMOylated. A nuclear fraction was lysed in urea-containing buffer, and proteins were immunoprecipitated with rabbit anti-SNF2PH antiserum or unspecific antiserum (prebleed) and probed with anti-TbSUMO mAb 1C9H8 (arrow). As a control, IP samples were reprobed with SNF2PH antiserum (below).

B  A reciprocal IP experiment was performed using anti-TbSUMO mAb and probed with SNF2PH antiserum. As a control, the blot was reprobed with anti-TbSUMO mAb (lower panel). Inp: Input, IP (0.5%).

C  Anti-Flag Western blot analysis of SNF2PHN performed on soluble cell extracts from induced cultures of *E. coli* transformed with pET28-SNF2PHN-3xFlag alone (lane 1), or in the background of an incomplete (lane 2, pACYCDuet-1-TbE1a-TbE1b; lane 3, pCDFDuet-1-TbSUMO-TbE2) or a complete (lane 4, pCDFDuet-1-TbSUMO-TbE2 plus pACYCDuet-1-TbE1a-TbE1b) SUMOylation system.

D  Similar samples as described in (C) were analyzed for SNF2PHC.

E  Cell lysates of *E. coli* heterologously expressing SNF2PH and the complete *T. brucei* SUMOylation system were incubated at 28°C in the absence (−) or presence (+) of recombinant TbSENP. The deconjugation activity of TbSENP was specifically inhibited by the addition of 20 mM NEM. Reaction mixtures were analyzed by Western blot using anti-Flag monoclonal antibodies.

F  Western blot analysis of SUMOylated SNF2PHN pattern performed on soluble cell extracts from a complete bacterial SUMOylation system using a wild type version of SUMO (lane 1) or a Lys-deficient version of SUMO (TbSUMO K9R) unable to form chains (lane 2).

additional slowly migrating bands observed when SNF2PHN was co-expressed with the *T. brucei* SUMOylation bacterial system (lane 1) completely disappear upon treatment of cell lysates with TbSENP (lane 2), and the deconjugation ability of TbSENP was specifically inhibited by addition of 20 mM NEM (lane 3). To investigate the nature of SUMOylation of SNF2PHN, we compared the patterns obtained in the bacterial system when replacing wild-type SUMO with a variant unable to form SUMO chains (Fig 2F). In the latter case, a doublet near the 55 kDa marker can be detected, suggesting that there are at least two major sites for SUMOylation in the SNF2PH N-terminus.

**SNF2PH is highly enriched at active VSG-ES promoter chromatin**

To investigate SNF2PH occupancy at VSG-ES loci, we performed chromatin IP (ChIP) using anti-SNF2PH antiserum in a promoter-tagged cell line. To overcome the problem of highly homologous sequences at the promoter region among the 15 telomeric VSG-ESs, we developed a tagged cell line (<u>D</u>ual-reporter <u>R</u>enilla <u>A</u>ctive <u>L</u>uci-ferase <u>I</u>nactive or DRALI) (loci of interest schematic in Fig 3A). The reporter genes in the DRALI cell line allowed us to determine SNF2PH occupancy at the region downstream of the promoter in either active or inactive VSG-ESs. First, we analyzed occupancy of

SNF2PH by ChIP and quantitative PCR (qPCR), which detect significant SNF2PH enrichment at the *RLuc* gene downstream of the active VSG-ES promoter (*P* < 0.001) as well as at the active VSG221 gene located in the telomere of BES1 (*P* < 0.01). However, *FLuc* located downstream of an inactive VSG-ES promoter (Fig 3A) was not significantly detected (Fig 3B). SNF2PH enrichment was also not

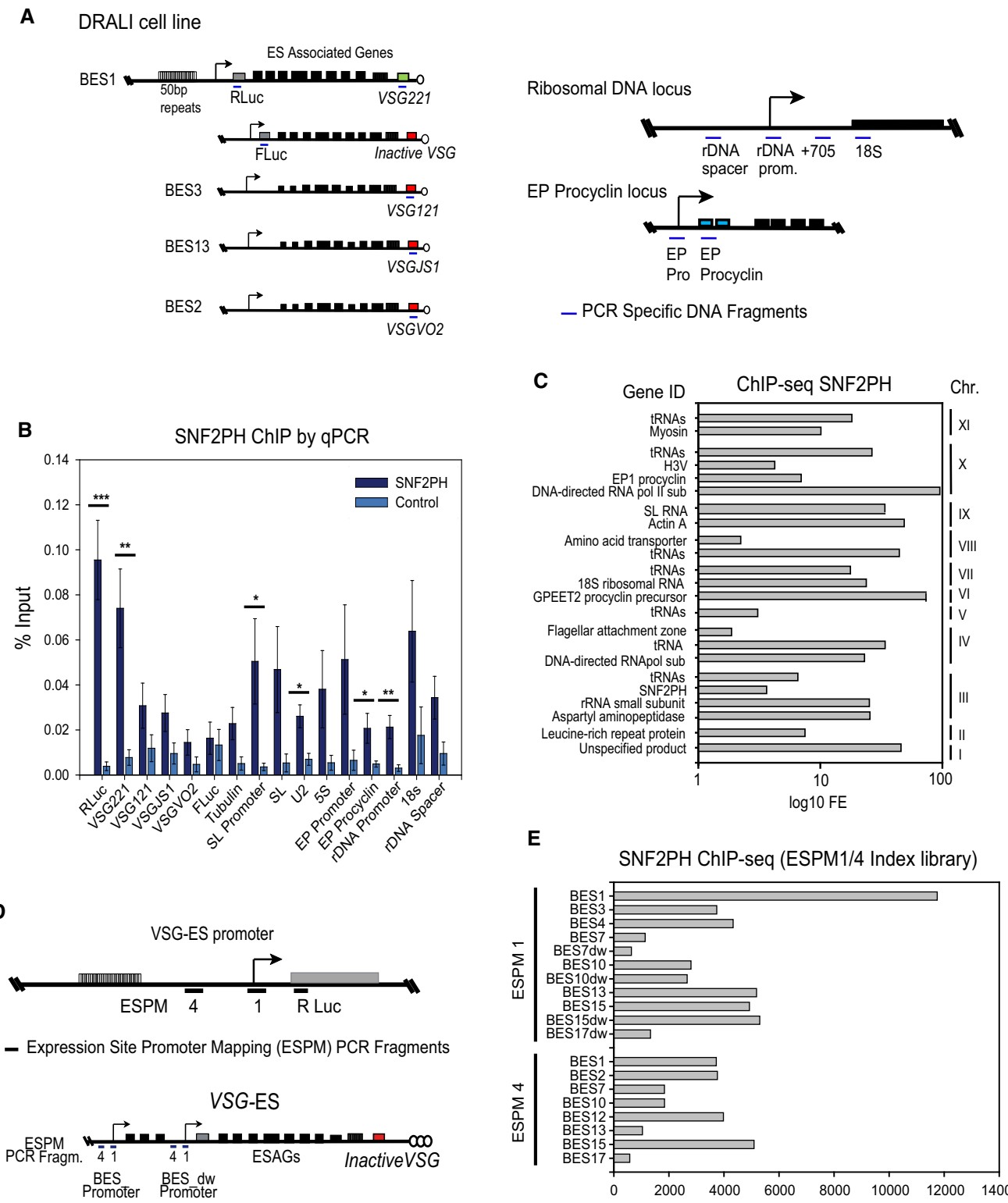

**Figure 3.**

**Figure 3. SNF2PH is highly enriched upstream of the active VSG-ES chromatin while is detected to a lesser extent in silent promoters.**

A  Schematic representation of loci of interest in DRALI, the dual-reporter cell line (not to scale). Two reporters were inserted, Renilla luciferase gene (RLuc) downstream of the Active VSG221-ES (BES1) promoter and firefly Luciferase gene (FLuc) downstream of an Inactive VSG-ES (DRALI). Few other inactive VSGs known to be telomeric in this strain are also represented (VSG121 (BES3), VSGJS1 (BES13), and VSGVO2 (BES2)). Schematic representations for other chromosomal loci (ribosomal DNA and procyclin locus) are shown. Color code: gray (reporters), green (active VSG-ES), red (Inactive VSG-ESs), blue (procyclin locus). Arrow (promoters).

B  Chromatin at the active VSG-ES is enriched for SNF2PH. Chromatin immunoprecipitation (ChIP) analysis by quantitative PCR of reporter sequences inserted downstream of the VSG-ES promoters indicates SNF2PH is highly enriched at the active VSG-ES (RLuc) (BES1) compared to an inactive VSG-ES promoter (FLuc) (***$P < 0.001$). SNF2PH enrichment on the active telomeric VSG221 (BES1) compared to inactive VSGs (VSG121 (BES3), VSGJS1 (BES13), and VSGVO2 (BES2)) was also significant (*$P < 0.05$–**$P < 0.01$). SNF2PH occupancy was detected at the splice leader promoter (SL promoter, pol II-transcribed) and EP procyclin ($P < 0.05$), as well as the rDNA promoter ($P < 0.01$) (Student's $t$-test) *$P < 0.05$ **$P < 0.01$, ***$P < 0.001$). ChIP analyses are shown as the average of at least three independent experiments with standard error of the mean (SEM). Data are represented as percent of input immunoprecipitated (% input).

C  Distribution of SNF2PH across the genome. ChIP-seq analysis using the SNF2PH antiserum and *T. b. brucei* 427 genomic library (v4) excluding the telomeres. Histogram illustrates peak enrichment of representative genes expressed as $\log_{10}$ fold enrichment (FE). This global analysis confirmed previous ChIP data locating SNF2PH on developmentally regulated genes (EP and GPEET), RNA pol I driven rDNA (ribRNAs) and the SL cluster of small RNAs that are trans-spliced in every mRNA. Interestingly, beside those essential genes for cell growth, SNF2PH occupies few other coding genes; noteworthy is H3V protein recently linked to the regulation of VSG monoallelic expression [28]. In addition, SNF2PH was consistently enriched at tRNA gene clusters in 7 chromosomes. Due to highly homologous sequences among ESAGs, all ES-related sequences as ESAGs genes, VSG basic copies located in chromosomal internal positions were excluded of this graph since we cannot rule out whether the ChIP-seq reads came from the active VSG-ES (all sequences are included in Dataset EV1).

D  Schema of VSG-promoter region indicating the location of ESPM PCR fragments amplified (upper panel). Detailed schema of the promoter region showing both upstream and downstream (dw) BES from the tandem repeated promoters ESPM 1 and 4 (lower panel).

E  Chromatin at the core promoter of the active VSG-ES is highly enriched in SNF2PH. ChIP-seq data using SNF2PH antiserum reveal a higher number of reads corresponding to the sequence polymorphism of the BES1 at the PCR fragment 1, (ESPM1) mapping at the VSG-ES promoter (Fig 3D) described before in [9]. dw, downstream promoter.

detected at *VSG* genes known to be located at silent telomeric ES position in this strain [2], such as *VSG121* (VSG in BES3), *VSGVO2* (BES2) and *VSGJS* (BES13) (Fig 3B). Altogether, the active *VSG221* gene immunoprecipitated more efficiently than all inactive *VSG* telomeric loci analyzed, suggesting SNF2PH associates preferentially with the active ES telomere. Additionally, SNF2PH was detected at other RNA pol I-transcribed loci, including rDNA and EP procyclin promoters. Occupancy of SNF2PH at the two promoters of the surface glycoprotein genes characteristic of mammalian and insect forms (VSG-ES and EP) implicates SNF2PH in regulation of developmental gene expression. Enrichment was also detected for the splice leader (SL) promoter ($P < 0.05$) and coding regions. However, SNF2PH was most enriched at the active VSG-ES chromatin compared with EP and rDNA promoters.

In eukaryotes, chromatin remodelers are detected at RNA pol II promoters and play important roles in their activity [26,27]. We investigated the presence of SNF2PH in chromatin across the genome, aside from the multiallelic VSG-ESs, to identify additional genes targeted by this protein. We compared quantitative enrichment profiles with ChIP-seq peak distribution and considered 0-mismatch error to avoid bias in polymorphic sequences within repetitive chromosomal loci, leading to defined peaks ($q$ value $< 0.05$, Dataset EV1; Fig 3C). As demonstrated by quantitative ChIP, the site of enrichment corresponded to developmentally regulated loci EP and GPEET2 procyclin, and 18S ribosomal DNA and SL-RNA-related sequences. Interestingly, SNF2PH localizes at H3.V, a histone variant recently associated with VSG monoallelic expression [28] and its own coding sequence. SNF2PH was also significantly enriched at a substantial number of tRNAs gene arrays located in chromosomes XI, X, VIII, VII, V, IV, and III (Fig 3C); interestingly, tRNA clusters are known to function as a chromatin insulators in eukaryotes from yeast to human, reviewed in Ref. [29].

Figure 3B shows that SNF2PH is enriched at the active VSG-ES (BES1) locus by ChIP qPCR analysis using unique sequences like the Rluc reporter inserted downstream of the promoter and the

VSG221 gene at the telomeric end. However, we wished to investigate in detail a possible SNF2PH occupancy at the area adjacent to the promoter; nevertheless, highly homologues sequences shared among most of the VSG-ESs (BES1 to BES17 in ref. [2]) prevented this analysis using a simple ChIP-seq alignment. We have previously reported polymorphisms in the sequence at particular regions located at the core promoter and upstream the promoter region, referred to as ES promoter PCR fragments 1 and 4 (ESPM1 and ESPM4) (schematically represented in Fig 3D) [9]. These minor polymorphisms in the sequences allowed us to differentiate among different BES promoter regions. In particular, PCR fragment ESPM 4 and 1 yielded 14 different sequences when genomic DNA was used as template (Appendix Fig S5A in [9]) providing considerable covering of most of the BESs [2]. ChIP-seq data were generated from the immunoprecipitated chromatin with the anti-SNF2PH serum, which was then PCR amplified with ESPM1 and 4 PCR primers, and the products were deep-sequenced (see Materials and Methods). Reads were aligned to BES promoter sequences, and an index file was built by combining the sequences from the ESPM1 and 4 together in a single lane with the sequences from the corresponding BESs (BES1, 2, 3, 4, 7, 7dw, 10, 10dw, 12, 13, 15, 15dw, 17, 17dw described previously [2]). Next, using Bowtie software, the alignments of the reads were assigned to the BES promoter index file, and the number of reads aligning to each BES is shown in Fig 3D. These data showed that SNF2PH is enriched at the active BES1, at the ESPM1, which corresponds to the core promoter of the active BES1 (VSG-ES221). SNF2PH was detected to a lesser extent at other BES promoters, suggesting that SNF2PH is controlling inactive promoters as well (Fig 3D). Interestingly, the most prominent increase of read counts was found at the ESPM1 fragment region, where the actual ES promoter is located, suggesting SN2PH is associated with the active core ES promoter rather than the upstream promoter region (Dataset EV1). Together, these data indicate that SNF2PH is located at several BES promoters; however, it is most enriched at the active VSG-ES promoter (ESPM1 of the BES1) (Fig 3E).

## SUMOylation functions to recruit SNF2PH at the active VSG-ES

SUMOylated chromatin-associated proteins are enriched upstream of the promoter region in *T. brucei* [9]. We assessed the importance of SUMOylation on SNF2PH targeting by mutating lysine residue 2 to alanine (K2A). K2 was selected as it was predicted as a modification site (SUMO V2.0 Webserver (http://sumosp.biocuckoo.org/) and is contained within the N-terminal region modified by the *in*

*bacteria* SUMOylation system (Fig 2). SNF2PH K2A was expressed with an HA tag from the endogenous locus (Fig EV2). We performed ChIP qPCR analysis using anti-HA rabbit antiserum after 48-h expression of HA-SNF2PH K2A, using the parental cell line as a control (Fig 4A). Expression of HA-tagged SNF2PH showed similar occupancy to SNF2PH, but by contrast the HA-SNF2PH K2A mutant was reduced in the active VSG221 telomeric locus (BES1) up to 4.6-fold compared to inactive VSG genes, VSG121, VSGJS1, VSGVO2

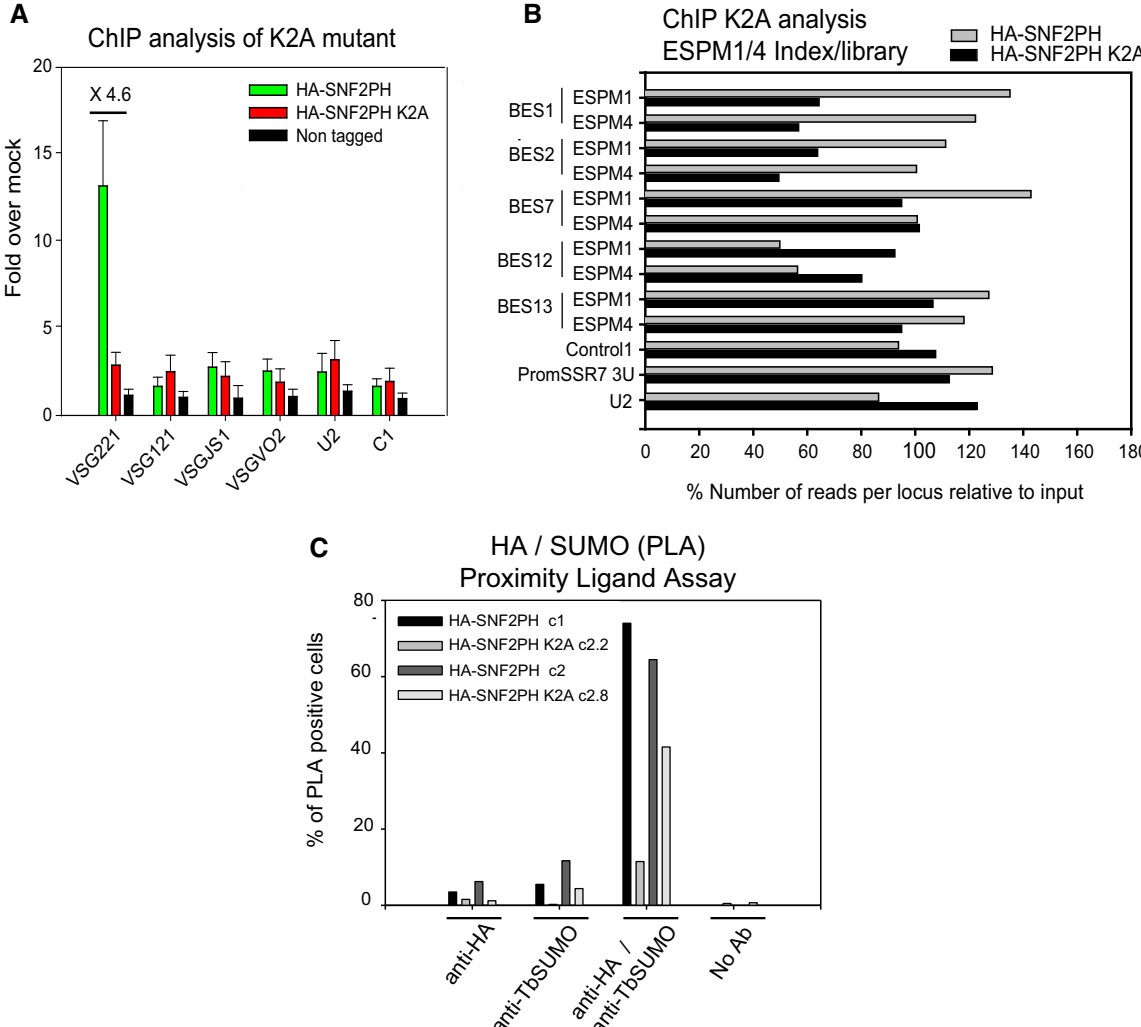

**Figure 4. Expression of a SUMO-deficient mutant reduces SNF2PH occupancy at the active VSG-ES.**

A ChIP analysis of mutant HA-SNF2PH K2A shows a reduction in the active telomeric VSG221 gene occupancy compare to the WT HA-SNF2PH. ChIP experiments were carried out using the anti-HA antibody and chromatin isolated from 3 independent clones expressing HA-SNF2PH K2A compared with a cell line expressing WT HA-SNF2PH. The mean of the ChIP analyses from three clones is represented as fold enrichment. A non-tagged cell line (Single Marker) was included as a negative HA control. Histogram shows the mean from three independent clones expressing the K2A mutant, and error bars represent means ± standard error of the mean (SEM).

B Reads per locus relative to input (%). ChIP analysis using anti-HA antibody in HA-SNF2PH and HA-SNF2PH K2A cell lines aligned with the ESPM1/4 index library containing the sequences from all the BES amplicons. ChIP analysis of the HA-SNF2PH versus the mutant HA-SNF2PH K2A generated ChIP libraries. Control 1; C1, PromSSR7 3U; Switch Strand Region promoter, Chr.7.

C *In situ* detection of SUMOylated SNF2PH using a Proximity Ligation Assay (PLA) is reduced in mutant SNF2PH K2A. Percentage of nuclei showing positive amplification signal in PLA assay analyzed as described before [9]. Histogram comparing the % PLA positive cells is represented separately for two independent clones overexpressing the HA-SNF2PH K2A mutant. Trypanosome transgenic cell lines usually vary considerably in protein expression level and kinetic, leading to a considerable degree of variability. Notwithstanding, both cell lines showed a decrease in PLA-positive nuclei upon expression of SNF2PH K2A. WT HA-SNF2PH c1 PLA-positive 73.98% (n = 246 total cells) versus HA-SNF2PH K2A Clone 2.2, positive PLA 11.47% (n = 514). WT HA-SNF2PH c2 PLA-positive cells 64.43% (n = 298) versus HA-SNF2PH K2A Clone 2.8 PLA-positive cells 41.39% (n = 387).

(located at the telomeric end of the BES3, BES13, and BES2, respectively [2]).

In order to determine whether K2A mutation also affects to SNF2PH occupancy at the promoter region, we constructed a ChIP-seq index library including all the BES sequences from PCR regions ESPM1 and 4 (Fig 3D) and map the reads PCR amplified from DNA immunoprecipitated using anti-HA antibodies and chromatin from cell lines expressing HA-SNF2PH and HA-SNF2PH K2A (Fig 4B). As controls we used gene Control 1, a promoter in chromosome 7, SSR7 (Strand Switch Region, SSR) (sequence defined by the prom SSR7 3U primers in Appendix Table S4) a RNA pol III (U2) were included after input normalization (Fig 4B and Dataset EV2). Sequence alignments using *Bowtie1* and 0 mismatch error yielded a number of reads aligned in BES ESPM1/4 libraries at the active promoter in BES1 and BES2 (highly sequence homologue) that were reduced for the cell line expressing the mutant K2A, to approximately ~0.5-fold. Conversely, at inactive promoters of BES the number of reads was increased (BES12) or no significantly changed (BES7 and BES13). This result is consistent with mutation K2A reduced protein SUMOylation, which decreases the occupancy of SNF2PH in the active promoter chromatin (Fig 4B).

We also assessed the effect of the K2A mutation on SUMO conjugation by PLA. The PLA signal for 3HA-SNF2PH (69.20% ± 4.77) was considerably greater than for the HA-SNF2PH K2A mutant (26.49% ± 15.02, $n = 2$), suggesting that K2 is required for efficient SUMOylation *in situ* (Fig 4C). We conclude that SNF2PH is mainly located at the active VSG-ES and that this specifically requires SUMO modification.

## SNF2PH is a transcriptional activator that regulates VSG expression

To obtain direct evidence for function, we knocked down SNF2PH and analyzed the effect using the DRALI cell line. Western blot demonstrated depletion of the protein after 24 h RNAi (Fig 1B and Appendix Fig S4A) and a reduction to proliferation was also observed suggesting SNF2PH is essential for normal fitness (Appendix Fig S4B). RT–qPCR analysis after 48 h RNAi indicated significantly reduced levels of *RLuc* and *VSG221* mRNAs ($P < 0.05$), without changes to RNA pol II- or pol III-transcribed control loci C1 and U2, respectively. No reduction was detected in mature or pre-spliced rDNA + 780 RNAs (Fig 5A), suggesting SNF2PH depletion

decreases *VSG* expression specifically. Next, we analyzed TbRPAI (pol I largest subunit) occupancy in VSG-ES chromatin in cells depleted of SNF2PH by ChIP using anti-TbRPAI (Fig 5B). Upon SNF2PH depletion, we detected lower levels of TbRPAI recruitment to the active VSG-ES (BES1) including the active *VSG221* telomeric locus ($P < 0.05$). This also detected for the *RLuc* gene inserted downstream of the active promoter (2.94-fold). A lower decrease at the rDNA promoter and ribosomal 18S locus was not significant. Hence, SNF2PH is specifically involved in recruitment of the RNA polymerase to the active VSG-ES, suggesting SNF2PH is required for active VSG-ES transcription.

While a reduction of the reporter at the active VSG-ES promoter was detected upon RNAi in three independent clones, we found that *Fluc* activity from inactive VSG-ES promoters was clone dependent (Appendix Fig S4C). Relative expression of *FLuc* transcripts correlates with the FLuc expression level (Appendix Fig S4D). We performed RNA-seq analysis on paired groups of individual clones (Appendix Fig S4E and Dataset EV3), which showed variability in the VSG that is derepressed. Some, like VSG427-15 (BES10) and VSG-14 (BES8), but not all were upregulated, while the active BES1 telomeric *VSG221* gene was consistently downregulated, suggesting that where SNF2PH was depleted, random derepression of inactive VSG-ES promoters occurred. Thus, SNF2PH depletion induced derepression of a cluster of silent BES but not all, similar to recently reported for H3V-H4V KO [28]. As the PH domain is known to bind Histone 3 tails, the H3V KO [28] and SNF2PH depletion phenotypes may be related.

Next, we asked whether VSG protein expression was also reduced in SNF2PH depleted cells. VSG221 protein levels in three independent SNF2PH RNAi clones (Fig 5C) were significantly decreased ($P < 0.05$) (Fig 5D). Decreased VSG221 at the cell surface was also detected after SNF2PH RNAi by fluorescence-activated cell sorting (FACS) (Fig 5E). These results indicate that SNF2PH functions as positive transcription factor for VSG expression.

To identify factors associated with SNF2PH, we ectopically expressed triple-HA-tagged SNF2PH followed by affinity purification and LC-MSMS (Appendix Table S2 and Table EV1). Among the identified proteins, we found mRNA splicing factor TbPRP9 and nucleosome assembly protein, AGC kinase 1 (AEK1), a kinase essential for the bloodstream form stage [30] and a TP-dependent RNA helicase SUB2, (Tb927.10.540). Importantly, several previously identified VSG transcription factors, including Spt16 included in FACT complex (Facilitates Chromosome Transcription) [31], the

**Figure 5. SNF2PH depletion results in a reduction of active VSG expression.**

A  Reduced VSG-ES transcripts upon SNF2PH 48 h RNAi. Quantitative RT–qPCR analysis indicates 43% reduction of VSG221 mRNA, validated by RLuc reporter (*$P < 0.05$). No significant changes in ribosomal RNA transcripts were detected (18s and rDNA + 780).

B  Reduced RNA pol I occupancy in the active VSG-ES upon SNF2PH depletion. TbRPAI analysis was carried out in 48 h RNAi-induced cell lines and the parental cell line (DRALI). Statistical analysis shows a significant reduction of TbRPAI occupancy levels between parental and SNF2PH-depleted cells at the active VSG-ES (*$P < 0.05$). Data from three independent clones with standard error of the mean (SEM) are represented as fold over non-specific antiserum. NS; nonsignificant.

C  Quantitative Western blots of VSG221 expression in three independent SNF2PH RNAi clones using IR fluorescence. Anti-VSG221 and tubulin antibodies were incubated with the same blot and developed using goat anti-rabbit IgG 800 and anti-mouse IgG 700 Dylight (Thermo Fisher). A standard curve based on tubulin-normalized anti-VSG221 signal intensity was generated using different concentrations of parental cell extracts ($R^2$=0.99).

D  Quantitation of VSG221 expression relative to the parental cell line. Relative VSG221 protein levels appear to be reduced compared with the parental cell line (*$P < 0.05$).

E  FACS analysis of VSG221 expressing cells shows a decreased active VSG221 population upon SNF2PH depletion (continuous red line) performed on DRALI cell line. Control: DRALI cell line population expressing VSG221. SNF2PH-dox⁺: induced SNF2PH-depleted cells. SNF2PH-dox⁻: uninduced SNF2PH-depleted cells. Secondary Ab: population incubated with secondary antibodies as a negative control.

Data information: All data are reported as the mean ± SEM for biological replicates ($n = 3$). *$P < 0.05$ using two-tailed Student's *t*-test for paired observations.

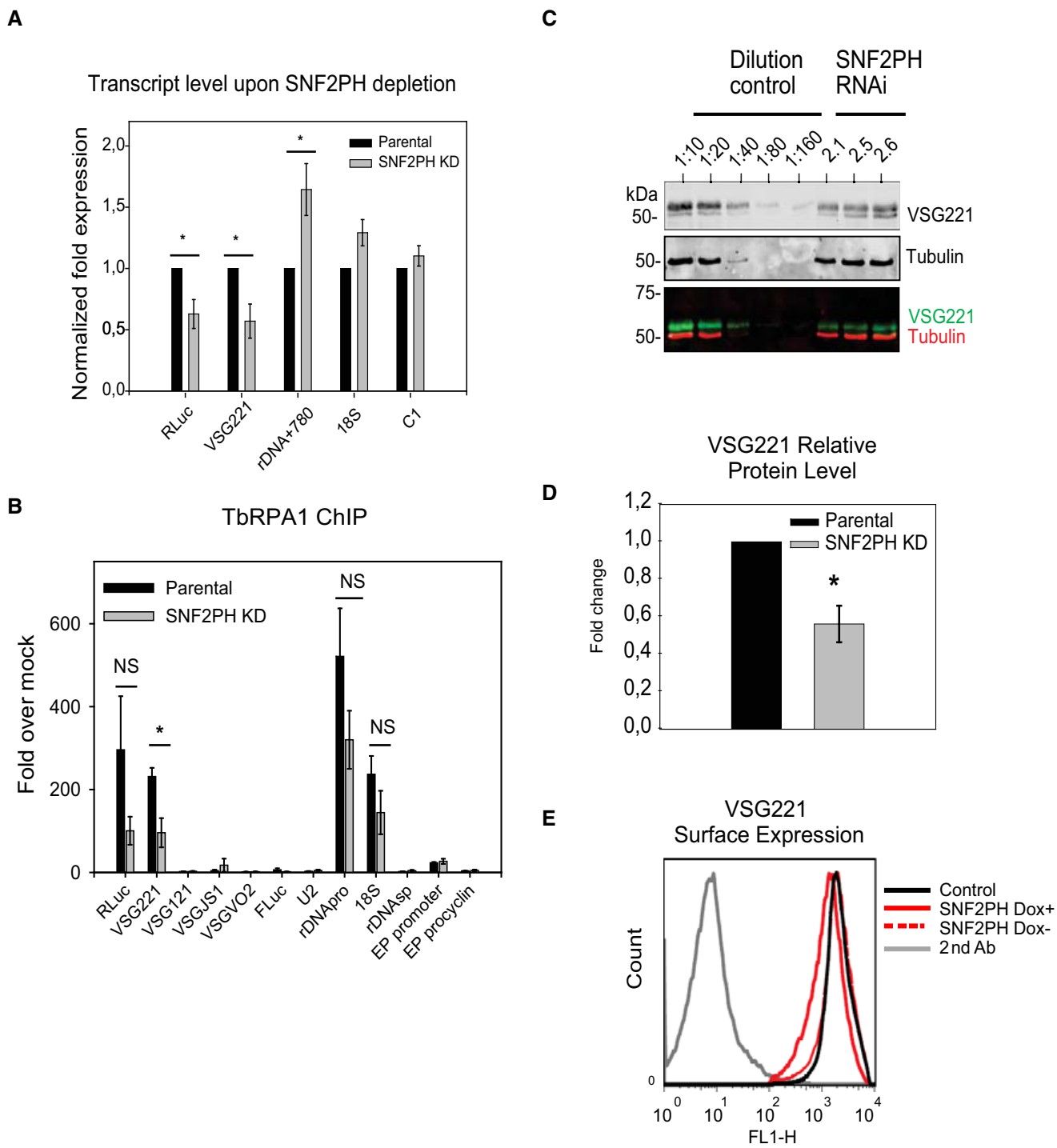

**Figure 5.**

proliferative cell nuclear antigen (PCNA) and a subunit from the Class I transcription factor A [32] were also identified (Appendix Table S2). Furthermore, RNA pol I subunit RPA135 and the RNA pol II RPB1 were also found to co-purify with SNF2PH suggesting possibly a transient interaction with RNA polymerases subunits, as previously described for yeast SNF2. Together, these data suggest that SNF2PH occupies a central position in VSG transcription regulation and links to RNA pol II transcription.

**SNF2PH is required for the maintenance of the bloodstream stage expression profile**

The above results suggest that SNF2PH regulates VSG expression however also interacts with a RNA pol II subunit and RNA binding proteins suggesting SNF2PH may regulates the expression of additional genes. In addition, SNF2PH protein levels in the bloodstream form are decreased in the insect form suggesting a possible function

in development. To investigate that possibility, we knocked down SNF2PH in bloodstream trypanosomes. Transcript levels for EP procyclin ($P < 0.01$), PAD1 and PAD2 (Protein Associated with Differentiation 1 and 2) were significantly upregulated ($P < 0.05$), whereas MyoB levels decreased ($P < 0.01$) (Fig 6A). Relative mRNA levels of ribosomal 18S and C1 were unchanged. An increase in EP procyclin and PAD1/2 transcription followed by a reduction in MyoB, similar to the transcript profile of the stumpy form, a transition to the insect stage [33,34].

Furthermore, we investigated whether SNF2PH depletion influences EP procyclin replacement dynamics during *in vitro* differentiation from the BF to the PF. We triggered differentiation with 3 mM *cis*-aconitate (Fig 6B). Under these conditions, SNF2PH-depleted cells expressed higher procyclin transcripts at 4 h of induction, whereas VSG221 mRNA levels were reduced compared to the parental cell line. Surface protein analyses of EP procyclin and VSG221 in SNF2PH-depleted cells by IF, found a ~2-fold faster replacement of VSG by EP procyclin (Fig EV3).

To investigate additional genes regulated by SNF2PH, we carried out RNA-seq upon knockdown in the bloodstream form (Fig 6C). Eighteen representative differentially expressed genes (DEG) with a false discovery rate (FDR) < 0.05 were found out of 8673 genes from two experimental replicates and 221 transcripts with a $P < 0.05$ (Dataset EV4). Among them, clear upregulation of genes related to the procyclic form, including procyclin genes (EP1-2, GPEET2), followed by procyclin-associated genes (PAG1, 4 and 5) and expression site-associated genes (ESAG2 and 8), some of them consistent with previous analysis in procyclic forms [33] (Appendix Table S3). We also found upregulation of protein associated with differentiation 1, 2 (PAD1 and PAD2), zing finger family ZC3H18, and the receptor-type adenylate cyclase GRESAG4-related transcripts. Conversely, genes with normally higher expression in bloodstream form were downregulated,

including the alternative oxidase and glucose transporter THT-2 (Dataset EV4).

Given that SNF2PH expression is clearly reduced in the procyclic stage (Fig 1A), we sought to investigate the gain-of-function phenotype induced by ectopic expression of SNF2PH in the procyclic form. Interestingly, we detected an increase in silent telomeric VSG121, VSGBn2, and VSGJS1 transcripts from two independent clones in the procyclic form where no VSG is normally expressed (Fig 6D).

To determine possible changes to global gene expression induced by SNF2PH overexpression, we used RNA-seq and compared the overexpressor and parental procyclic cell line (Fig 6E). We found increased expression of ESAG3 from several BES, ranging from 2.5 to 5.2- log FC, while the promoter adjacent transferrin-binding protein ESAG6/7 increased 1.7 to 3.2- log FC (Fig 6E and Dataset EV5). Of the total mapped reads, 55.08% were linked to telomeric bloodstream ESs (BESs), including BES1 (BES40 ~5.2- log FC), BES4 (BES28/98 ~4.5- log FC), BES2 (BES129/126 ~3.8- log FC), and BES13 (BES56/153/51/4 ~2.6- log FC) (Fig 6E and Dataset EV5). Importantly, no expression of VSG basic copies genes, located within internal chromosomal arrays, was detected, ruling out global chromatin deregulation. These results show that, in procyclic forms where no VSG is expressed, overexpression of SNF2PH induced upregulation of telomeric VSG-ES (BES) transcripts, suggesting that SNF2PH functions as a central regulator of the BES. Thus, in BF, where SNF2PH is highly expressed, this chromatin remodeler likely functions to promote and maintain the expression of VSG-ES (BES).

Next, we analyzed the relevance of the SNF2PH plant homeodomain and overexpressed a truncated form of SNF2PH lacking the PH domain (SNF2ΔPH) (Fig 6F). RNA-seq analysis detected 737 genes (FDR < 0.05 and $P < 0.01$, out of 7,918 genes) differentially expressed after overexpression of SNF2ΔPH versus 118 genes

**Figure 6. SNF2PH is required for maintaining infective form surface protein expression.**

A    Depletion of SNF2PH (48 h) in the bloodstream form results in an increase of procyclin and proteins associated with differentiation (PADs) transcripts. Procyclin, PAD1, and PAD2 mRNAs are upregulated when SNF2PH is depleted. Results are the average from three independent clones and data normalized with U2 mRNA. Error bars represent means ± SEM. (*$P < 0.05$, **$P < 0.01$) using two-tailed Student's *t*-test for paired observations.

B    SNF2PH KD cells differentiate to procyclics more efficiently. The procyclin transcript is increased in SNF2PH KD-depleted cells during *in vitro* differentiation compared with parental cell line. Parental and SNF2PH-depleted cells were treated with 3 mM cis-aconitate and temperature shift for 4 hours. Quantitative RT–PCR data from two independent clones were normalized against C1 (RNA pol II-transcribed) as a housekeeping gene. Error bars represent means ± standard deviation (SD).

C    Scatter plot for differentially expressed genes (DEG) from RNA-seq analysis upon SNF2PH depletion (FDR < 0.05). Non-DEG: Non-differentially expressed genes. Up: upregulated genes. Down: downregulated genes.

D    Ectopic expression of SNF2PH in procyclic form upregulates telomeric VSG mRNAs. Histogram showing the relative expression of mRNA measured by qRT–PCR of VSG genes located in different telomeric VSG-ESs (BES) genes after 48 h of induction of SNF2PH overexpression in procyclic form. This analysis included mRNAs from telomeric VSG genes (BF stage-specific) not expressed normally in the insect procyclic form, including the VSG121 (BES3), VSGJS1 (BES13) VSGBR2 (BES15), and VSGVO2 (BES2). Data from two independent clones and parental controls are represented as normalized fold expression relative to C1 (RNA pol II transcribed) as a housekeeping gene. Error bars represent means ± SD from technical replicates for each independent clone.

E    Ectopic expression of SNF2PH in procyclic induces expression of bloodstream form surface proteins. Scatter plot for differentially expressed genes (DEG) in RNA-seq analysis shows upregulation of telomeric BESs. A significant increase of BES-associated genes (ESAGs) linked to telomeric BESs was detected after full-length SNF2PH overexpression in the procyclic form. Data from at least two experimental replicates are represented as log₂ fold change (FC) for genes with FDR < 0.05 and $P < 0.001$ after correction with the uninduced procyclic cell line. Relative mRNA levels of procyclic cells after 48 h of SNF2PH overexpression increase telomeric BESs, including VSG221-ES (BES1; TAR40), VSG121-ES (BES3; TAR15), VSGJS1-ES (BES13; TAR56), and VSGVO2-ES (BES2; TAR129), VSGR2-ES ES (BES15; TAR126), see (2) for detailed BES and TAR nomenclature (Dataset EV5). Ectopic expression of SNF2PH full-length also induced invariant surface glycoprotein 65 (ISG65) genes (Dataset EV5). Data from two independent clones and parental controls are represented as normalized fold expression relative to C1 (RNA pol II transcribed) as a housekeeping gene.

F    Expression of bloodstream form surface proteins requires SNF2PH plant homeodomain. EP procyclins and other insect stage-specific markers (FDR < 0.05, $P < 0.001$) are expressed in the SNF2PH mutant form lacking of the plant homeodomain (SNF2ΔPH). No BF surface proteins like VSG-ES (BES related or ESAGs) neither invariant surface glycoproteins (ISG65) genes were induced under ectopic expression of the SNF2ΔPH mutant.

 

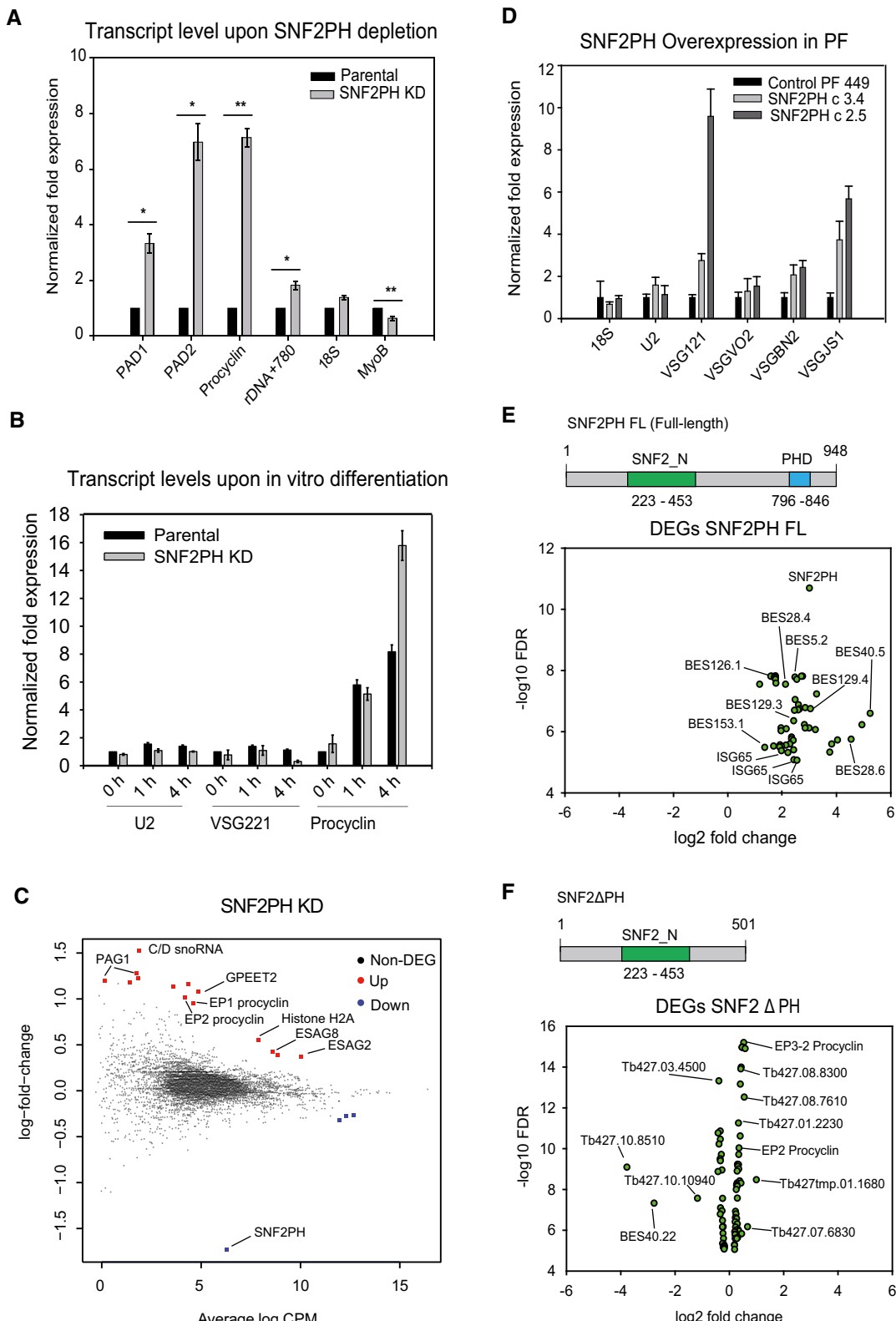

**Figure 6.**

(FDR < 0.05 and *P* < 0.001, out of 8,451 genes) for SNF2PH full length, illustrated in Fig EV3C and D. Most significantly, induction of BES-related genes did not occur in cells overexpressing the

SNF2ΔPH mutant lacking the PH domain (Fig 6F), suggesting that the PH domain is required to provide SNF2PH specificity to bind and recognizes VSG-ES (BES) chromatin (Fig 6F and Dataset EV5).

Significantly, beyond impacts to BES expression, invariant surface glycoprotein ISG65 genes, transcribed exclusively in the bloodstream form, were also significantly increased up to 6-fold in procyclics ectopically expressing SNF2PH compared to the parental procyclic cell line (Fig 6E). Importantly, ISG gene transcripts were not detected in cell lines expressing SNF2ΔPH (Dataset EV5). We do not know whether ISG65 is expressed at the surface of these cells, but consider it unlikely as we previously demonstrated that ectopically expressed ISGs are rapidly degraded in insect stage cells [35], as well as with ectopic expressed mRNA VSG. As VSG is transcribed by the RNA pol I and ISG65 by RNA pol II, this suggests that SNF2PH acts beyond the telomeric BES, and may act as a global transcriptional activator of surface protein genes irrespective of the RNA polymerase involved.

These results suggest that the PH domain is essential to direct SNF2PH to specific bloodstream form surface proteins genes, promoting transcriptional activation. In metazoans, transcription factors containing homeodomains determine cell fates during development, and this is the chromatin-binding domain necessary to regulate downstream target genes. Our results provide an important contribution of the plant homeodomain in recognition epigenetic chromatin patterns to regulate gene expression in trypanosomes, an early-branching eukaryote.

### SNF2PH is downregulated in quiescent stumpy forms

The stumpy form of pleomorphic trypanosomes is pre-adapted to the metabolism required by the procyclic form for survival within the insect vector. Given upregulation of stumpy form markers in SNF2PH knockdown cells, we examined SNF2PH dynamics during the stumpy to procyclic form transition. Recent evidences suggest that the AMP-dependent kinase, AMPKα1, is a key regulator of the development of quiescence in bloodstream form trypanosomes [42] reviewed in [43]. Upon AMPKα1 activation, stumpy-like differentiation was induced in a monomorphic cell line. Quantitative Western blots and RT–qPCR analyses of monomorphic cells treated with an AMP analog showed a significant decreased in SNF2PH proteins levels and transcripts (Appendix Fig S5A). Reduction of SNF2PH protein levels in stumpy-like cells obtained after AMPKα1 activation by AMP suggests AMPK pathway negatively regulates SNF2PH expression.

We also assessed the transcriptional profile in stumpy-like forms induced by AMP treatment and compared with SNF2PH knockdown. qRT–qPCR confirmed a transcriptional profile characteristic of the stumpy form in untreated SNF2PH knockdown versus AMP treated cells, in which relative mRNA levels for SNF2PH transcripts were downregulated ($P < 0.01$; Appendix Fig S5B). Interestingly, stumpy form-like transcriptome changes were more prominent in cells treated with 5′-AMP and depleted for SNF2PH compared with 5′-AMP alone and 5′-AMP untreated knockdown cells.

An important question, however, is whether SNF2PH downregulation occurs naturally during *in vivo* differentiation of a wild-type pleomorphic strain. Bloodstream pleomorphic trypanosomes undergo differentiation from the proliferative to the quiescent stumpy form throughout mice infection. SNF2PH protein levels analysis in the pleomorphic AnTAT 90.13 strain decreased at 4-5 days postinfection (Appendix Fig S5C), whereas AMPKα1 was

fully activated as previously described [42]. Taken altogether, these data suggest that SNF2PH is negatively regulated during transition to stumpy form and mechanistically linked to AMPKα1 activation by an undefined mechanism.

## Discussion

Antigenic variation in African trypanosomes is mediated by complex regulatory mechanisms that secure monoallelic expression of a single VSG, the fundamental basis of immune evasion. The transcriptional state of VSG genes is maintained through several generations and is the product of epigenetic mechanisms and post-translational modifications (PTM) that mark the active VSG genes. Several PTMs are associated with silencing the inactive VSG-ESs [3], but SUMOylation is the only known modification associated with the active VSG-ES [9]. A concentration of SUMOylated proteins [9] is located adjacent to the expression site body where VSG gene transcription occurs. Additionally, several chromatin-associated SUMO-modified proteins are enriched at the active VSG-ES, and chromatin SUMOylation is also required for efficient RPAI recruitment [9].

Epigenetic factors are likely regulators of VSG transcription; silencing of TbISWI, HDACs, and additional chromatin remodelers such as ASF1, CAF1, and FACT, consistently results in similar phenotypes, i.e., the derepression of normally silent VSG-ES subtelomeric regions (reviewed in [4]). By contrast, pharmacological inhibition of bromodomain chromatin remodelers leads to the upregulation of both silent and basic copy VSG genes, suggesting a role in maintaining global chromatin organization, rather than specific action at the active ES [36]. Further, post-translational histone modifications are associated with the repression of silent VSG-ESs (see [37] for a review). Interestingly, proteins regulating nuclear architecture also alter monoallelic VSG expression, including the lamina proteins NUP-1 [38] and cohesin, where silencing leads to a ~10-fold increase in VSG switch frequency [24]. Deletion of histone variants H3.V and H4.V highly increases VSG switch frequency *via* homologous recombination, and the telomeric VSG-ES loci are confined to distinct compartments within the nucleus [28]. Altogether, these data support a model where epigenetically marked VSG-ES chromatin occupies a unique subnuclear compartment, facilitating monoallelic expression [6].

The product of Tb927.3.2140 is a SUMO-conjugated protein unregulated in the bloodstream form. Tb927.3.2140 possesses domains homologous with the SWI/SNF family of chromatin remodelers, e.g., SMARCA1, also known as the global transcription activator SNF2L1, and SNF2 helicase domains. Thus, we named the gene product SNF2PH. The SNF2 family N-terminal domain participates in many processes [16], while the chromatin SWI/SNF remodeler superfamily regulates either specific or global gene expression. SNF2PH also contains helicase domains homologous to transcriptional activators HsSMCA5 and MsSMCA1 of the ISWI subfamily (Appendix Fig S1). Interestingly, SNF2PH contains a homeodomain that interacts directly with histone tails to regulate development, the plant homeodomain (PHD), which is present in histone methyltransferases, such as MmHMT3 and HsNSD3 (Appendix Fig S1). The *H. sapiens* chromatin remodeler CHD4-based NuRD also contains a PHD [39] and was shown to act in

both a positive and negative manner in regulating cell-specific differentiation [16]. Multiple lines of evidence suggest that SNF2PH has an architecture typical of transcriptional regulators and chromatin readers and similarly acts in both, positive and negative manners, to regulate developmental stage-specific gene expression of surface proteins in trypanosomes.

SUMO modification acts as a switch for the localization of SNF2PH, and modified SNF2PH is recruited to the active VSG-ES promoter, while unmodified SNF2PH is present at silent VSG promoters. While we cannot be certain if this is cause or consequence, loss of function associated with a non-SUMOylated mutant of SNF2PH suggests an active role for SUMOylation in targeting. SNF2PH depletion leads to downregulation of the active VSG-ES, indicating a possible function as a positive regulator. This conclusion is supported by an increased frequency of sequences related to BES1 in the ChIP-seq data, as well as upregulation of procyclin and PAD genes after SNF2PH knockdown, suggesting a suppressive function in the insect stage. Interestingly, ChIP-seq analysis revealed SNF2PH occupancy at the H3.V locus, which suggests that SNF2PH regulates H3.V expression, a histone variant crucial for regulating VSG expression [28].

SNF2PH associates with several proteins, including histone chaperone Spt16 [40] and proliferative cell nuclear antigen (PCNA), essential for replication [41], suggesting a link between SNF2PH and cell cycle progression. Another significant interactor is the CITFA-4 subunit of the RNA pol I promoter binding complex [31]. SNF2PH also shows an association with RNA pol I subunit RPA135 and RNA pol II subunit RPB1, the latter consistent with SNF2PH occupancy at pol II-transcribed loci (Fig 3B). This prompts us to speculate that SNF2PH-pol II association facilitates accessibility of transcription factors to regulate RNA pol II transcription.

SNF2PH depletion leads to expression of insect form genes, which suggests a role in maintaining the BF chromatin state by opposing control of the two major developmentally regulated surface gene families, procyclins and VSGs. Importantly, overexpression of SNF2PH in the procyclic form leads to the expression of bloodstream form telomeric VSGs and ESAGs, requiring the PH domain to promote telomeric expression in a repressed VSG transcriptional background. Consequently, this domain is apparently required to maintain bloodstream form through developmental regulation [19].

The stumpy form is a pre-adaptation to the insect host and requires AMPKα1 activation [42]. 5′-AMP analog treatment leading to enhanced AMPKα1 activation in monomorphic cells resulted in SNF2PH downregulation. Whereas AMPKα1 activation promotes differentiation, reduced SNF2PH expression in stumpy forms observed in wild-type pleomorphic trypanosomes during mice infection confirmed the biological relevance. This result rules out a possible SIF-independent induction of differentiation by VSG-ES transcription attenuation described previously [44].

Interestingly, SNF2PH is enriched at tRNAs clusters on several chromosomes (Fig 3C). tRNAs can act as insulators to prevent the spread of silencing in *S. cerevisiae* [45,46] and in mammals to prevent enhancers from activating promoters [47,48]. We detected low colocalization of SNF2PH with GFP-tagged active VSG-ES (Fig 1D), and even lower association in 2K1N cells (G2) (Fig EV1A), which may suggest that the association of SNF2PH with the active VSG-ES locus occurs preferentially in G1. Although such insulator function has not been described in trypanosomes, the presence of SNF2PH at tRNA loci during interphase, as some insulator proteins [49], suggests a tentative model where SNF2PH/tRNAs could act as a chromatin domains organizer to maintain bloodstream stage-specific surface protein gene expression.

The ISG65 genes are transcribed by RNA pol II from polycistronic arrays located in core chromosomal regions, while VSG genes are transcribed by RNA pol I from the VSG-ES (BES) telomeric locus. Notwithstanding, the chromatin remodeler SNF2PH, utilizing the PH domain as epigenetic reader, recognizes both distinct gene families. This is suggested by SNF2PH ectopic expression in procyclic forms, which led to a developmental epigenetic reprogramming, similar to homeodomain proteins in other organisms. Furthermore, ISGs are transcribed at similar levels for all allelic variants, while the VSG is monoallelically transcribed at one out of 15 different telomeric BESs [2]. Interestingly, SNF2PH overexpressed in procyclic forms lacking SUMO induced then expression of all BES (Fig 6E), and monoallelic expression of VSG genes was not achieved. We speculate that SNF2PH SUMOylation is likely the modification that SNF2PH acquires at the nuclear body ESB, where VSG-ES transcription occurs [6,9], allowing SNF2PH to recognize and activate a single BES telomere among the VSG multiallelic gene family.

In sum, SNF2PH requires SUMO modification to function as a transcriptional activator of VSG-ES monoallelic expression, and the PH domain is required for this and for maintaining the mammalian infective form surface protein coat, ensuring continuous and proper VSG and ISG surface display, essential for pathogenicity.

# Materials and Methods

### Trypanosome strains and cell lines

*Trypanosma brucei brucei,* bloodstream form (Lister 427, antigenic type MiTat 1.2, clone 221a), 427 procyclic form and the pleomorphic AnTAT 90.13 were used in this study. The dual-reporter cell line, DRALI, contains the *Renilla* luciferase (*RLuc*) gene inserted 405 bp downstream of the active VSG-ES promoter and the *Firefly* luciferase (*FLuc)* gene downstream of an inactive VSG-ES promoter. The insertion site was checked by sequencing the flanking region from DRALI genomic DNA confirming *RLuc* inserted in the active VSG221-ES (BES1), whereas *FLuc* was inserted downstream of the inactive VSG-ES promoter BES15/TAR126 VSGbR-2/427-11. The VSG221-ES GFP-tagging and YFP-TbRPB5z fusion were previously described [24].

### Recombinant proteins and monoclonal antibodies

C-terminal fragment of SNF2PH (Tb927.3.2140) was amplified by PCR, and the PCR product was cloned into *Bam*HI and *Hind*III sites of pET28a vector (Novagen) expressed as a C-terminal His tag (Appendix Tables S5 and S6). The recombinant protein was purified using NI Sepharose Fast flow 6 (GE Healthcare) and inoculated into mice to generate anti-SNF2PH (11C10E4) monoclonal antibody (mAb), using standard procedures. Hybridomas were screened against the recombinant protein by ELISA and further confirmed by Western blot analysis using trypanosome protein extracts that recognized the protein of the expected size.

Hybridoma cell line 11C10E4 was grown as ascites. SNF2PH polyclonal antibody was obtained by affinity purification from rabbit antiserum after several inoculations of the recombinant protein using an Aminolink column (Pierce), following the manufacturer's instructions. Anti-TbSUMO (1C9H8) monoclonal antibody was generated as previously described [9].

### 3xHA tagging of SNF2PH versions

A *T. brucei* bloodstream form cell line expressing a 3xHA-tagged SNF2PH was developed by replacing both copies of the endogenous gene. For the 3xHA K2A, one allele was replaced by the mutant version due to cell viability. Procyclic form cells carrying both 3xHA-tagged full-length and truncated SNF2PH isoforms were ectopically expressed from the ribosomal spacer. Cloning procedures are detailed in Appendix Tables S5 and S6.

### RNAi experiments

SNF2PH RNAi construct was made using the p2T7Bla vector [50]. Since most of the RNAi constructs using this vector are leaky, comparative analyses always included addition of the dox induced (+) and uninduced (−) RNAi in the parental cell line (DRALI). Amplified PCR fragment corresponding to 1113-bp of C-term SNF2PH ORF was cloned into *Bam*HI and *Hind*III sites of p2T7Bla and transfected into the dual-reporter cell line DRALI (Appendix Tables S5 and S6). Synthesis of dsRNA was induced by adding 1 μg/ml of doxycycline. At least three independent clones were analyzed, and protein depletion was confirmed by Western blot using specific antibodies.

### RT–qPCR

RNA isolation, cDNA synthesis, and qPCR were performed as previously described [9]. Relative expression levels were referred to a control (parental cell line) and normalized against a housekeeping gene (U2, pol III-transcribed gene), using the software Bio-Rad CFX Manager Software. Experimental conditions were performed in triplicate and analyzed by Student's *t*-test. A detailed primer list is found in Appendix Table S4.

### RNA-seq analysis

Total RNA from at least two independent biological replicates of both SNF2PH knock down (BF) and overexpression (PF) after 48 h of doxycycline induction was used to generate a library from poli (A) + mRNA isolated fragments. Libraries were sequenced on an Illumina NextSeq 500 platform (150 cycles) in paired-end mode with a read length of 2 × 76 bp and sequence depth of approximately 50 million reads per sample. The miARma-Seq pipeline [51] was used to analyze all transcriptomic data. In detail, this pipeline contains all needed software to automatically perform any kind of differential expression analysis. It uses fastqc to check the quality of the reads and aligned them using hisat2 on *T. brucei* TREU427 reference genome (TritrypDB release 39). Subsequently, the aligned reads are quantified and summarized for each gene using featurecounts. Finally, gene counts are analyzed using the edgeR package from Bioconductor. In such a way, all samples were size corrected in order to be comparable and then normalized using the TMM

method from the EdgeR package. TMM values for each gene were used for the differentially expression analysis. RPKM values for each gene were calculated from the normalized read counts values using the rpkm method from edgeR. Genes transcripts isolated form uninduced versus induced SNF2PH RNAi cells with a [$\log_2$FC] ≥ 1 ($\log_2$ of Fold Change) and FDR ≤ 0.05) were considered as differentially expressed. Additionally, miARma-Seq [51] generated a volcano plot to facilitate the identification of genes that felt higher variation in expression.

### Chromatin Immunoprecipitation (ChIP) and ChIP sequencing (ChIP-seq)

Bloodstream form *T. brucei* cultures were fixed and processed as previously described [9]. Pre-cleared chromatin ($5 \times 10^7$ cells per IP) was incubated with each antibody (90 μg anti-SNF2PH, 6 μg anti-TbRPAI, 6 μg of rabbit anti-HA tag antibody (abcam), and 90 μg of an unspecific antiserum). The immunoprecipitated products were reverse crosslinked, and the extracted DNA was analyzed by quantitative PCR (qPCR). For ChIP-seq analysis, the protocol was scaled for a final concentration of ~5 ng of immunoprecipitated DNA. To compare the amount of DNA immunoprecipitated to the total input DNA, 10% of the pre-cleared chromatin saved as input was processed with the eluted immunoprecipitated products before the crosslink reversal step. Quantitative PCR was performed using SYBR green Supermix (Bio-Rad) in a CFX96 cycler (Bio-Rad). IP percentages were determined as previously described [9]. At least three independent experimental assays were displayed and analyzed by Student's *t*-test. A detailed primer list is detailed in Appendix Table S4.

### Generation of the ChIP-seq library

#### ChIP-seq analysis in Fig 3D

Immunoprecipitated DNA (~5 ng) from each condition was evaluated in a 2100 BioAnalyzer to assess fragmentation size and subjected to end-repair enzymatic plus dA-tailing treatments further to be ligated to adapters using the Illumina TruSeq DNA Sample preparation kit, following the manufacturer's instructions. Adapter-ligated libraries were enriched with 15 cycles of PCR using Illumina PE primers and purified with a double-sided SPRI size selection in a range below 300 bp. Libraries were sequenced in an Illumina NextSeq 500 platform leading to a 650,000 reads per sample. The raw reads were processed using the miARma-Seq pipeline [51] to measure quality, adapter sequence removal and read alignment. Briefly, this software first assessed the quality of the sequences using FASTQC tool kit. After that, adapter sequences were removed using the cutadapt utility. Once reads were processed, they were aligned against the Trypanosoma brucei Lister 427 genome obtained from TriTrypDB version 34 using the BWA aligner with default parameters. Later, final results obtained from miARma-seq were processed with macs. Therefore, each paired sample (-f BAMPE) chip (-t) was processed against the input sample (-c) to eliminate general peaks in both types of samples using as organism size $2.7e^7$ (-g). The correspondence of peaks between both types of samples (HA-SNF2PH and HA-SNF2PH K2A) and with gene sequences was carried out with the intersectBed script from bedtools using the gene annotation provided by TriTrypDB version 34 and visualized in GB browser. A starting pool of 8 amplicons 18S, U2, C1, prom SSR7,

ESPM-1/4, VSG221 (BES1), and VSG121 (BES3)) were combined together in a single lane per condition (including respective inputs). Coding sequences for 18 and U2 were used as reference genes to evaluate the immunoprecipitation efficiency in each experimental case. Differential peak distribution was represented as fold enrichment relative to input and assessed by –logP value, considering a 0 nucleotide mismatch to discern telomeric sequences.

### ChIP-seq analysis in Fig 3E

To discriminate among the BES promoters, we carried out ChIP-seq analysis using selected PCR ES Promoter Mapping (ESPM) regions known to have sequence polymorphisms among different BESs, as previously described [9]. A pool of amplicons of sequences of the promoter regions ESPM1 and ESPM4 (Fig 3D, defined with the primer include in Appendix Table S4) containing the corresponding sequences from the BES1, BES2, BES3, BES4, BES7, BES7dw, BES10, BES10dw, BES12, BES13, BES15, BES 15dw, BES17, BES17dw, and 18S, U2, C1 as control were combined together in a single lane to build an index file (ebwt), next the alignment was done with bowtie -S -n0 command to consider a 0 nucleotide mismatch to distinguish among few nucleotide sequence differences in each BES, as described before [9]. The actual number of reads aligned on each BES in represented in the histogram of Fig 3E.

### Cell extracts and Immunoblots

Parasite cultures were collected by centrifugation and washed once in Trypanosome Dilution Buffer (TDB) with 1X protease inhibitor cocktail (Roche) and 20 mM N-ethylmaleimide (NEM) and pellets were processed as previously described [9]. Western blot membranes were incubated with anti-SNF2PH mAb ascites (1:1,000), anti-SNF2PH affinity-purified antiserum (1:1,000) and monoclonal anti-HA high affinity (1:500, clone 3F10, Roche Applied Science). Mouse monoclonal anti-TbSUMO mAb ascites (1:1,000), anti-Tubulin mAb (1:5,000), anti-MVP mAb (1:1,000), rabbit anti-VSG221 antiserum (1:50,000), and rabbit mAb Phospho-AMPKα (Thr172) (40H9) (1:1,000, Cell Signalling technologies) were used as described previously [9,34,42].

### Quantitative western blots

Quantitative Western blots analyses were performed as previously described [9]. Membranes were incubated with anti-VSG221 (1:50,000), anti-SNF2PH affinity-purified antiserum (1:1,000), Tubulin (1:5,000), and anti-MVP mAb (1:1,000). A standard curve based on Tubulin-normalized anti-VSG221 signal intensity was generated using different concentrations of parental cell extracts ($R^2 = 0.99$). The standard curve regression was used to determine VSG221 expression levels in SNF2-depleted cell lines. For both detection of AMPK phosphorylated levels and SNF2PH in in vitro and in vivo assays, a MVP-normalized anti-SNF2PH and/or anti-p-AMPK were used to quantify differences in signal intensity compared with the parental condition.

### Purification and identification of TbSUMO conjugates

To identify SUMO conjugates from a bloodstream form, T. brucei cell line expressing an 8xHis and HA-tagged version of SUMO (Tb927.5.3210) was developed by replacing both copies of the endogenous gene, see Appendix Tables S5 and S6. Purification of conjugates was performed in denaturing conditions by nickel affinity chromatography. After imidazole elution, the urea concentration was decreased and SUMOylated proteins were subjected to a second affinity purification step using an anti-HA agarose resin. Final conjugates were analyzed by mass spectrometry, and processing data was performed as previously described [34].

### Purification of protein complexes and identification by Nano-LC-MS/MS

A total cell mass of $4.0 \times 10^{10}$ procyclic form cells, strain T. brucei 449 expressing the 3HA-tagged version of TbSNF2PH was induced with 1 μg/ml doxycycline during 48 h and harvested at 1,400 g rpm during 10 min at 4°C. The pellet is washed in 50 ml PBS 1× including protease inhibitors and subjected to cryogenic grinding, resulting in a lyophilized powder with all nuclear components, as previously described [52]. Immunoprecipitation assays were performed with 50 mg of lyophilized powder and resuspended immediately in 1 ml of Lysis buffer (20 mM HEPES pH 7.4, 50 mM sodium citrate, 1 mM MgCl$_2$, 10 μM CaCl$_2$, 2× protease inhibitor cocktail (Roche), and 0.1% CHAPS), followed by three cycles of sonication of 15 s at 50 W and centrifuged at 20,000 g during 10 min at 4°C. The supernatant containing the nuclear fraction was incubated with 10 μl (0.1 mg) of previously equilibrated HA-Magnetic beads and incubated during 2 h at 4°C on rotation and washed three times, preserving the same buffer conditions. HA-Magnetic beads were eluted at 99°C during 5 min with 15 μl of NuPAGE SDS Sample buffer (Life technologies) with 1.5 μl of NU PAGE SDS Sample Reducing Agent (Life technologies) and denatured at 99°C during 5 min prior to being analyzed in an SDS–PAGE gel with silver stain. Sample preparation for MS analysis was eluted with 50 μl of 2% SDS and 20 mM Tris–HCl pH 8.0 at 72°C and precipitated with 100% ethanol. After centrifugation, the sample was subjected to tryptic digestion and reductive alkylation of Cys groups with 50 mM iodoacetamide and finally vacuum-dried to be dissolved in 1% acetic acid. Then, tryptic peptides mixtures were injected onto a C-18 reversed phase nano-column (100 mM ID, 12 cm, Teknokroma) and separated in a continuous acetonitrile gradient. Eluted peptides from the RP nano-column were fragmented in a LTQ-Orbitrap Velos Pro mass spectrometer (Thermo Scientific). For protein identification, the mass spectra were deconvoluted using MaxQuant version 1.5 searching the T. brucei427_927_Tritryp-3.1 annotate protein database (37,220 proteins). Search engine analysis was performed assuming the full trypsin digestion (strict trypsin) in Mascot version 2.4.1. with pre-established parameters (Fragment Tolerance: 0.60 Da (Monoisotopic) Parent Tolerance: 10.0 PPM (Monoisotopic) Fixed Modifications: +57 on C (Carbamidomethyl) Variable Modifications: -17 on n (Gln->pyro-Glu), +16 on M (Oxidation), +32 on M (Dioxidation), +42 on n (Acetyl)). To visualize MS-spec data, we used Scaffold Proteome Software version 4.4.6 with a 0.5% peptide threshold and 5% protein threshold and 1 peptide minimum. False discovery rates (FDR) were calculated for both peptide and protein levels. A non-tagged cell line (procyclic form 449) was used to subtract contaminant proteins.

## Immunoprecipitation in denaturing conditions to detect TbSUMO conjugates

For each immunoprecipitation (IP) experiment, $1.0 \times 10^{10}$ bloodstream form (BF) cells were used. Cells were washed in TDB with 1X protease inhibitor cocktail (Roche) and 20 mM NEM. Pellets were resuspended at $\sim 5.0 \times 10^9$ cells/ml in urea buffer (6M Urea, 50 mM HEPES pH 7.5, 500 mM NaCl, 20 mM NEM, 0.5% NP-40, 2× protease inhibitor cocktail (Roche)) and sonicated until their viscosity was lost. The cell lysate was centrifuged at 20,000 $g$ for 10 min at 4°C, and the supernatant with the nuclear enriched fraction was stored at −80°C. For immunoprecipitations, the nuclear extract was diluted 1:6 with dilution buffer (50 mM HEPES pH 7.5, 500 mM NaCl, 1% NP-40, 0.5% Lauryl Sarcosine, 0.1 mM EDTA, 10 mM NEM, 1× protease inhibitor cocktail (Roche)) followed by overnight antibody incubation at 4°C on rotation. Antibody concentration for IP experiments was 180 μg/ml anti-SNF2PH rabbit antiserum, 800 μg/ml of anti-TbSUMO mAb 1C9H8, and 180 μg/ml of unspecific IgGs (prebleed antiserum). Previously equilibrated Protein G Sepharose beads (Sigma-Aldrich) were added to each diluted extract ($\sim 5.0 \times 10^9$ cells) containing the antibodies and incubated during 1 hr at 4°C on rotation to capture specific IgGs. Beads were washed five times for 5 min at 4°C on a rotating wheel with 1 ml of wash buffer (1 M urea, 50 mM HEPES pH 7.5, 500 mM NaCl, 1% NP-40, 0.5% Lauryl Sarcosine, 0.1 mM EDTA, 10 mM NEM, 1× protease inhibitor cocktail (Roche)). Then, beads were eluted with 2× Laemmli sample buffer and boiled at 99°C for 5 min. IP samples and inputs were subjected to SDS–PAGE and quantitative Western blotting using the appropriated antibodies.

## 3D-Immunofluorescence

Three-dimensional immunofluorescence (3D-IF) was carried out on cells in suspension as previously described [24]. Mouse anti-TbSUMO mAb 1:2,000, mouse anti-SNF2PH mAb 1:1,000, rabbit anti-SNF2PH affinity-purified antiserum 1:1,000, rabbit anti-VSG221 antiserum 1:50,000, mouse anti-Procyclin mAb 1:500 (MyBioSource), and mouse anti-GFP mAb, 1:600 (Invitrogen) were used as primary antibodies. Alexa Fluor 488 and Alexa Fluor 594 goat anti-mouse or anti-rabbit (Invitrogen) were used as secondary antibodies. Cells were DAPI stained. Pseudocoloring, colocalization analysis, and maximum intensity projections were performed using ImageJ Fiji program version 1.51n software (National Institutes of Health), and one-way analysis of variance was used to compare the Pearson's coefficient value generated by the JACoP analysis plugin, available under ImageJ Fiji. For the colocalization mask, the plugin "Colocalization highlighter" was used where two points are considered as colocalized if their respective intensities are strictly higher than the threshold of their channels, which was set to 80% and if their ratio of intensity is higher than the ratio setting value of 80%.

## In vitro Trypanosoma brucei SUMOylation assay in bacteria

*In vitro* reconstituted SUMOylation system was performed in *Escherichia coli* BL21 (DE3) cells transformed with pCDFDuet-1-TbSUMO/TbE2, followed by pACYCDuet-1-TbE1a-TbE1b. Competent bacteria were transformed again with pET28a(+)-3xFlag-SNF2PHN or pET28a(+)-3xFlag-SNF2PHC (See Appendix Tables S5 and S6 for cloning details). Assessment of SUMOylation reaction and TbSENP deconjugation assays were performed according to [13], and samples were analyzed by Western blot using an anti-Flag M2 mouse monoclonal antibody 1:5,000 (Sigma-Aldrich). Horseradish peroxidase-conjugated goat anti-mouse secondary antibody 1:5,000 (Sigma) was detected by Chemiluminescence using SuperSignal West Pico Chemiluminescent Substrate (Pierce).

## Proximity ligation assay

The PLA assay was performed as previously described [9] by using the rabbit anti-SNF2PH affinity-purified antiserum (1:1,000) and mouse anti-TbSUMO 1C9H8 (1:2,000) as primary antibodies.

## Luciferase assay

Luciferase assays were carried out using the Luciferase Assay System (Promega®) following the manufacturer's instructions from bloodstream form culture ($3 \times 10^6$ cells) of control and SNF2PH depleted cells. Lectures were performed in a FB 12 Single Tube Luminometer (Titertek-Berthold) with pre-established parameters (2 s of delay time/10 s temp).

## Fluorescent-activated cell sorting (FACS) analysis

SNF2PH RNAi bloodstream form induced cultures ($1.5 \times 10^7$ cells) were collected and processed as previously described [9] using anti-VSG221 (1:3,000) as primary antibody. Alexa Fluor 488 goat anti-rabbit (Invitrogen) was used as secondary antibody.

## Differentiation to procyclic form

Differentiation from slender to insect procyclic form was induced by 3 mM cis-Aconitate (Sigma-Aldrich), with a temperature shift from 37°C to 28°C and switching the medium to Differentiating Trypanosome Medium (DTM) as previously described [34]. The assessment of the differentiation process was monitored by a double IF to detect the expression of the surface glycoproteins using anti-procyclin and anti-VSG221 antibodies.

## AMP Analog treatment and obtaining *in vivo* stumpy forms

Parasites in culture at a low density ($2 \times 10^5$ cells/ml) were incubated with 8-pCT-2′-O-Me-5′-AMP (1 μM) (c078; Biolog Life Science Institute) during 18 h. To avoid the AMPK activation caused by cell density, the control and treated cells were analyzed at the same cell density. Slender and stumpy forms of pleomorphic AnTat 90.13 were purified from Balb/c mice at 3–5 days postinfection as previously described [42].

## Ethics statement

Slender (Lister 427, antigenic type MiTat 1.2, clone 221a) and stumpy (pleomorphic AnTat 90.13) forms were isolated from Wistar and Balb/C mice rats, respectively, in compliance with policies approved by the Committee on Use and Care of Laboratory Animals of the Institute for Parasitology and Biomedicine López-Neyra, National Spanish Research Council (CSIC-IPBLN).

## Statistical analysis

Statistical analysis was performed using two-tailed Student's *t*-test for paired observations using SigmaPlot Systat Software.

# Data availability

RNA-Seq and ChIP-Seq datasets produced in this study are available in the database: Sequence Read Archive PRJNA562785 (https://www.ncbi.nlm.nih.gov/sra/PRJNA562785).

Expanded View for this article is available online.

## Acknowledgements

The authors thank Dr. Alicia Barroso Del Jesus for excellent assistance and input with NSG methodology at the Genomic Unit and Dr. Laura Montosa at the Microscopy Unit (IPBLN-CSIC). This work was supported by grants from the Spanish Ministerio de Ciencia, Innovación y Universidades (RTI2018-098834-B-I00) and the Wellcome Trust (WTI 204697/Z/16/Z to MCF) and the grant from the Argentinian National Agency for Promotion of Scientific and Technological Research to VEA (PICT/2016/0465).

## Author contributions

AS carried out and analyzed most experiments described in the manuscript. MN designed and directed research. PAI and VEA developed and performed the heterologous trypanosome SUMOylation system of SNF2PH fragments and wrote/interpreted these results. DR-B, DL-F, CB, and JMB performed distinct experiments included in this work. EA-L analyzed and helps to interpret the NGS data in RNA-seq and ChiP-seq and IV-C provided technical support. MN, VEA, and MCF supervised, edit, and founded the study. AS, MCF, and MN interpreted the data and wrote the final manuscript.

## Conflict of interest

The authors declare that they have no conflict of interest.

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
