## [Review Process File · EMBO Reports]

SUMOylated SNF2PH promotes variant surface glycoprotein expression in bloodstream trypanosomes

Andreu Saura, Paula Iribarren, Domingo Rojas-Barros, Jean M. Bart, Diana López-Farfán, Eduardo Andrés-León, Isabel Vidal-Cobo, Cordula Boehm, Vanina Alvarez, Mark C. Field and Miguel Navarro

Review timeline:

Submission date:	1 March 2019
Editorial Decision:	23 April 2019
Revision received:	30 July 2019
Editorial Decision:	21 August 2019
Revision received:	22 September 2019
Accepted:	26 September 2019

Editor: Achim Breiling

Transaction Report:

1st Editorial Decision

23 April 2019

Thank you for the transfer of your research manuscript to EMBO reports. I have so far received reports from two of the three referees that were asked to evaluate your study, which can be found at the end of this email. I am sorry for the unusual delay, but referee #3 has still not submitted his/her assessment, even though s/he promised to do that several times. In order to proceed in a timely manner, I have now decided to continue without a third report.

As you will see, both referees think the manuscript is of interest, but requires major revisions to allow publication in EMBO reports. As the reports are below, I will not further detail them here, also as I think that all points need to be addressed.

Given the constructive referee comments, we would like to invite you to revise your manuscript with the understanding that all referee concerns must be addressed in the revised manuscript and in a detailed point-by-point response. Acceptance of your manuscript will depend on a positive outcome of a second round of review. It is EMBO reports policy to allow a single round of revision only and acceptance or rejection of the manuscript will therefore depend on the completeness of your responses included in the next, final version of the manuscript.

Revised manuscripts should be submitted within three months of a request for revision; they will otherwise be treated as new submissions. Please contact me if a 3-months time frame is not sufficient so that we can discuss the revisions further.

Supplementary/additional data: The Expanded View format, which will be displayed in the main HTML of the paper in a collapsible format, has replaced the Supplementary information. You can submit up to 5 images as Expanded View. Please follow the nomenclature Figure EV1, Figure EV2 etc. The figure legend for these should be included in the main manuscript document file in a section called Expanded View Figure Legends after the main Figure Legends section. Additional

Supplementary material should be supplied as a single pdf labeled Appendix. The Appendix includes a table of content on the first page, all figures and their legends. Please follow the nomenclature Appendix Figure Sx throughout the text and also label the figures according to this nomenclature.

For more details please refer to our guide to authors:
<http://embor.embopress.org/authorguide#manuscriptpreparation>

The title is currently too long and reads rather complicated. Please provide a simpler and shorter title (with not more than 100 characters including spaces).

Please add a paragraph describing the author contributions to the manuscript (above the conflict of interest statement), and acknowledgements (if you want to add those).

Important: All materials and methods should be included in the main manuscript file.

See also our guide for figure preparation:
http://www.embopress.org/sites/default/files/EMBOPress_Figure_Guidelines_061115.pdf

Regarding data quantification and statistics, can you please specify, where applicable, the number "n" for how many independent experiments (biological replicates) were performed, the bars and error bars (e.g. SEM, SD) and the test used to calculate p-values in the respective figure legends. Please provide statistical testing where applicable. See:
<http://embor.embopress.org/authorguide#statisticalanalysis>

Please also follow our guidelines for the use of living organisms, and the respective reporting guidelines: <http://embor.embopress.org/authorguide#livingorganisms>

We now strongly encourage the publication of original source data with the aim of making primary data more accessible and transparent to the reader. The source data will be published in a separate source data file online along with the accepted manuscript and will be linked to the relevant figure. If you would like to use this opportunity, please submit the source data (for example scans of entire gels or blots, data points of graphs in an excel sheet, additional images, etc.) of your key experiments together with the revised manuscript. Please include size markers for scans of entire gels, label the scans with figure and panel number, and send one PDF file per figure.

Finally, please format the references according to our journal style. See:
<http://embor.embopress.org/authorguide#referencesformat>

- a complete author checklist, which you can download from our author guidelines (<http://embor.embopress.org/authorguide#revision>). Please insert page numbers in the checklist to indicate where the requested information can be found.
- a letter detailing your responses to the referee comments in Word format (.doc)
- a Microsoft Word file (.doc) of the revised manuscript text
- editable TIFF or EPS-formatted single figure files in high resolution (for main figures and EV figures)

I look forward to seeing a revised version of your manuscript when it is ready. Please let me know if you have questions or comments regarding the revision.

REFeree REPORTS

Referee #1:

In this manuscript, Saura and colleagues build upon the recent description of a highly SUMOylated focus (HSF) associated with monoallelic expression of the Variant Surface Glycoprotein (VSG) in *Trypanosoma brucei* by exploring the function of a protein, SHF2PH, which they show is

SUMOylated, whose expression is developmentally regulated, that interacts with the VSG transcription locus (as well as other loci), and whose loss affects VSG expression and differentiation.

A great deal of new data on SNF2PH is presented, much of it with rigour, but the reader is left with a lack of clarity regarding what new insight is provided. The authors present a somewhat confused picture in which they appear to be suggesting a potentially novel role for SNF2PH in maintaining transcription of the only active VSG expression site, as well as a wider role in life cycle differentiation. However, a simpler explanation may be that SNF2PH acts as a negative regulator of differentiation of parasite cells (from those that inhabit the mammal to those found in the fly vector). One facet of such a function would be ensuring continued expression of the VSG, perhaps through modification of the chromatin in the VSG expression site in ways similar to found at other loci. If so, such functions appear consistent with evidence linking PH domains and development in other organisms.

I suggest that the authors need to alter the text to provide a clear explanation of what role or roles they suggest are played by SNF2PH in *T. brucei*; in addition, if they are proposing a specific role for SNF2PH in transcription of the VSG expression site, distinct from differentiation, they need to provide a clear explanation for the evidence for this proposal. The lack of clarity stems, at least in part, from deficiencies in several experiments, which should be addressed:

1. Page 7 and Fig.1. What do they authors mean by 40% overlap between GFP and SNF2PH signals, and 50% overlap between SNF2PH and SUMO signals; do they mean that they see such overlap in only this proportion of cells analysed, or do they mean they mean that they detect such an extent of overlap within each cell's nucleus (in which case a measure of variation should be provided)? Clarification here is critical in order to understand if SNF2PH always localises to the VSG expression site and HSF, or if this localisation is only seen in some circumstances.
2. The analysis of the ChIP data for SNF2PH is very unclear and should be improved. Most importantly, the authors should show the pattern of ChIPseq signal across the active and silent VSG expression sites, as well as at important other loci (rRNA, procyclin), rather than levels of enrichment using small parts of these regions (as in Fig.3D), which may give a distorted impression of enrichment. Once these data are shown, the qPCR of necessarily small parts (Fig.3B) can then be seen in the wider context. In addition, presentation of the analysis needs to be made more accessible:
 - How do the PCR regions in Fig.3A relate to the VSGs detailed in Fig.3B?
 - Why has the active VSG ES promoter region been duplicated in Figs.3A and C?
 - What is the meaning of the numbered BES (e.g. BES4), dw, DES, VO2 in Fig.3D?
 - How was the ChIPseq data analysed (I could not find this in the methods)?
3. The validity of the conclusions reached using the K2A mutant of SNF2PH appear questionable. Though it is clear that SUMOylation of the mutant protein is reduced and leads to loss of ChIP to the active VSG, the suggestion that this mutation 'drives SNF2PH occupancy towards inactive VSG-ES promoters' is not well supported. Most importantly, how can the authors exclude that the mutation renders SNF2PH inactive, rather than merely impairing SUMOylation? If the protein was inactive, perhaps through incorrect folding, the conclusions reached would be incorrect.
4. What happens to the expression of VSGs from the silent expression sites after RNAi? At the moment, the only description is that there is variation between clones. What exactly do they mean, and why did they not perform RNAseq of the different clones to clarify these confusing effects?

Minor criticisms:

Page 5, paragraph 1. SNF2 proteins are helicase-like factors, most of which are DNA-dependent ATPases; not all are involved in chromatin remodelling, as is being suggested. A recent, broad review should be cited.

Page 5, paragraph 2 is data and should be removed from the introduction and left to the results.

Page 5. What do they authors mean by nucleosome remodelling 'translocates' DNA?

Page 6, Table S1. What control affinity purification was performed to allow the authors to list the

factors in the table? What life cycle stage was used for this experiment?

Referee #2:

Monoallelic expression of Variant Surface Glycoprotein, with transcription by RNA polymerase I, is critical for survival of bloodstream-form trypanosomes and is a key element in the control of antigenic variation. VSG transcription is known to take place in a focus called the "expression Site Body" which is outside of the nucleus. The authors previously found that SUMO is concentrated on the ESB and that SUMO ChIP pulls down the VSG promoter region. They identified a SUMO E3 ligase, SIZ1. Knockdown of SUMO, SIZ1 or UBC9 decreased polymerase I transcription of the expression site. There is therefore convincing evidence that SUMOylation plays an active role in promoting VSG transcription.

In the current paper the authors set out to try to find out which SUMOylated protein(s) are responsible for the transcription effect. They used cells expressing His-tagged SUMO to pull down covalently modified proteins and identified them by mass spectrometry. Among the candidates was a protein with promising-looking domains that they have called SNF2PH. They show convincingly that:

- 1) SNF2PH is spread over quite a large part (but not all of) of the nucleus, overlapping with the ESB and the nucleolus.
- 2) Some of the SNF2PH is SUMOylated (the proportion is not measured). An N-terminal fragment could also be SUMOylated by a lysate from *E. coli* expressing the trypanosome enzymes - a very useful experiment and set-up for future work.
- 3) ChIP using antibody to SNF2PH, followed by PCR, indicates that it is enriched not only on the active VSG promoter but also promoters for U2 (pol III), rRNA (also pol I) and the spliced leader or a strand-switch region (pol II). These results suggest that it might be a general mark for open chromatin. There was much less enrichment on inactive VSG promoters.
- 4) The results were followed up by ChIPSeq but most of the results are not shown. No data for any promoters other than VSG ones is shown.
- 5) Next the authors (I think) replace the wild-type gene with either an HA-tagged version, or an HA-tagged version that lacks lysine 2. They could only mutate one allele, suggesting that Lysine 2 is important. They expect that Lysine 2 is SUMOylated but I don't think they formally showed it. The mutant protein no longer preferentially localises to the active expression site promoter, suggesting that SUMOylation is important for association with the ES but not with other promoters (though I didn't understand some of these results).
- 6) The authors look for proteins associated with SNF2PH and indeed detect both pol I and pol II - associated proteins.
- 7) Depletion of SNF2PH results in a decrease in VSG mRNA, consistent with a need for the protein in VSG transcription.
- 8) Next the authors looked at effects of the RNAi on overall gene expression. RNASeq yielded extremely few significant changes, but one of them was an increase in EP procyclin mRNAs, which is diagnostic of the very beginning of differentiation. Unfortunately, it is also commonly seen after various other treatments that impair cell growth. Moreover, it is known that attenuating VSG RNA synthesis can initiate differentiation (Batram et al., not cited). The effects of SNF2PH depletion on differentiation-type gene expression are therefore likely to be secondary. There is no evidence that SNF2PH is a direct regulator of differentiation.
- 9) The Supplement also contains some data relating to phosphorylation of SNF2PH. These results are unconvincing and should either be removed, or included as negative data.

All of the results from points 1-7 are good and interesting, although my interpretation would not be quite the same as the authors'. I conclude from that SNF2PH is localised to many different promoters, and probably generally associated with open chromatin, but that SUMOylation is important specifically for its action at the VSG expression site. This in itself is very interesting.

One missing link is what difference the SUMOylation makes. This could be examined by a comparison of the interactions of mutant and wild-type, HA-tagged SNF2PH. Whether or not there are differences, this would make a stronger paper.

Apart from that I have numerous detailed criticisms, all listed below.

GENERAL

Overall the failure of the authors to bother with details led to me wasting a considerable amount of time. This impacts the overall tone of the review.

In my opinion most of the alterations or additional information requested below is essential before publication, but there are hardly any new experiments.

A. Data availability:

a) Where are the database deposition numbers for the sequencing results? (Maybe I missed them?) where are the detailed supplementary Tables for the OMICS results?

b) Mass spectrometry: Please supply full data for the mass spectrometry in Tables S1 -and S2, including the controls. Were the peptide numbers for both purifications identical? If not, which was chosen for the simplified Tables? Please also supply the peak quantitation for all replicates and do a statistical analysis with an appropriate program. This should be possible if the analysis was done in the past 4 years, and may reveal additional interactions.

c) RNASeq: Please supply the raw read counts for the RNASeq and show the full statistical analysis for all genes using an established package (see below).

STATISTICS

Standard errors should not be used at all for these sorts of results. Readers are interested in the variation in your measurements, not the possible error in your estimate of the mean. If you had enough measurements to know you had a normal distribution, standard deviation would be appropriate. But in this case you do not know you have a normal distribution since usually there are only about 3 measurements. Bar graphs with error bars are no longer regarded as appropriate low numbers of replicates, since they fail to illustrate the real distribution of the data. Please show dot plots for individual measurements. This often makes results more convincing because big error bars can be caused by just one out of 3-4 measurements. (You can add a line for the mean, if you like.)

FIGURE FORMATTING

Please give all text labels similar sizes without laterally (or vertically) squashing them to fit. It should be possible to read the whole thing at the same magnification. 1B, 2E, 5C, D, E and S4E are the worst, but not the only, offenders; 2B has italics for no obvious reason. This is just laziness (cut it, paste it, and squash it no matter what it looks like) and it is extremely irritating.

DETAILED COMMENTS

1. Title:

This has to be changed completely,

- a) The authors present no evidence that *T. brucei* SNF2PH is a chromatin remodeller.
- b) The differentiation data are weak - delete the second half of the title completely (Anyway, most - indeed perhaps all - stages of trypanosomes are infective for either mammals or Tsetse flies.)

2. Introduction

The results from the previous paper on SUMO and VSG transcription should be described in a lot more detail - beyond the single sentence that is currently there.

Results:

3. Mass spectrometry of SUMOylated proteins.

- a) The list of SUMOylated proteins from bloodstream forms has no overlap whatsoever with the previously published list for procyclic forms. Given that the patterns of SUMOylation in the two forms are similar (10.7717/peerj.180), this seems very odd. Although some of the proteins in the procyclic list might be high-abundance contaminants, quite a lot of them probably are not. Please comment, at least briefly. Also please, in the Supplement, say which peptides were SUMOylated. It must be clear from the MS results.
- b) Is there any other evidence for SUMOylation of any of the proteins on this list? It would be

expected to result in a mobility shift on SDS-PAGE - Have any of these proteins been examined by Western blotting? Several have at least been studied before (give references).

c) Please check - and correct - the annotations. There aren't many of them so it shouldn't take long. For example I couldn't find any evidence that Tb927.2.4950 has anything to do with translation. There is no evidence that Tb927.11.15830 has RNA methyltransferase activity, beyond motifs. Tb927.11.14190 has a Tudor domain, but there is no evidence that it is involved in RNA-induced silencing...

4. Figure 2, Immunoprecipitations:

a) Please show the unbound as well as the bound fractions. That way it will be possible to tell whether the immunoprecipitations were quantitative or not. If they were then the results suggest that only 0.5% of SNF2PH is SUMOylated. If they aren't quantitative no such conclusion can be drawn. This is important since the authors repeatedly claim - without (I think) any evidence, although it is likely - that the SUMOylation is highly dynamic.

b) Since the authors have both antibody and recombinant protein, they could also very simply measure the absolute amount of SNF2PD in cells using a quantitative Western blot. That would be useful information. (This is desirable, not essential, but where else would you publish it and it isn't much work.)

5. ChIP and ChIP-Seq

a) Please explain the expression site terminology better, anyone who is not thoroughly versed in it would get totally lost. Explain what the pseudoVSG gene is.

b) The Figure legend for the first set of experiments is inadequate. It must be possible to see what everything is without hunting in the main text. What does "dw" mean? (Is this explained anywhere?)

c) "however it is significantly more enriched at the active VSG221 promoter region" - delete "significantly".

I assume from the results that VSG221 is in ES1 but this is not stated.

d) How big are these ES promoter region fragments? Fragment 1 is actually over, and immediately downstream of, the start site whereas (depending on the scale) fragment 2 would be in the promoter itself. Where are the results for ESPM2, 3, 5, 6 and 7?

e) Please also give the CHIP-Seq results for the other promoters analysed in (B). Where is SSR 3U and where are the results for it? Is it a strand-switch region?

f) The statement "A minor enrichment was also detected for the splice leader (SL) promoter and coding region." is actively misleading. The enrichment is the same as at the EP and rRNA promoters and the enrichment on the ES promoter is only 2-fold higher. And in Fig 4B there is no difference at all, although...

g) I do not understand Fig 4B. Does it mean that the enrichment was the same on all of the tested promoters? This is very poorly explained.

h) In the RPA1 CHIP, Fig 5B, is Rluc difference significant or not? If not it too should be labelled "NS", like rDNA and 18S.

i) For the HA-tagged proteins - it is essential to confirm that the mutant is not SUMOylated - this should be possible by co-IP.

6. SNF2PH-associated proteins

a) This would be MUCH more interesting if you were to include similar results for the non-SUMOylated mutant. This needs to be quantitative to allow a comparison with wild-type. Label-free quantitation is fine.

b) The discussion is biased. Please mention that RNA polymerase II subunits were also identified! Finding a splicing factor is also intriguing.

c) "We found PH (Pleckstrin)-like domain and Zinc finger containing protein ZC3H22 as mediators that allow gene transcription by recognition of lysine methylated histones". What is the evidence for this statement? First, which is the protein with the Pleckstrin-like domain (not on the list)? Second, I looked at the TritypDB page (Incidentally the Gene ID is in tb427 format.) and it appears that ZC3H22 is a cytoplasmic protein which is expressed mainly in procyclic forms; and the zinc-finger domain appears, from the Interpro page, to bind mainly to RNA. Finding 7 peptides from this protein in a bloodstream-form preparation is remarkable given that, from the database, it appears never to have been detected previously in bloodstream forms. Perhaps it is moonlighting in bloodstream forms with another function?

d) Check the locations and annotations of the other proteins as well. Won't take long.

7. RNASeq methods:

- a) It is not clear to me how these data were analysed. The authors only say (in the Supplement): "We used R software package" to analyse the data. Which package? there are many different packages available for analysis of RNASeq data, please say which one you chose. If you did not use a custom package, you must do so since normalization is an extremely complex problem. Using RPKM for data analysis is not acceptable since it ignores the impact of gene length on data reliability. For example, you had 10 million reads, 1 RPKM on a 10 KB ORF is 100 reads, which gives reliable results, but 1 RPKM on a 300nt ORF is 3 reads, which is not usable.
- b) Please present comparisons of reads using a standard package such as EdgeR or DESeq2, supply the full results in a supplementary table; show the principal component analysis; and describe the results more carefully, including a comparison with the available recent RNA Seq datasets for stumpy forms, not older microarrays.
- c) The sentence "We used R software package to detect genes that were differentially expressed in uninduced versus induced SNF2PH RNAi cells after subtraction of the parental (DRALI) cell line." is incomprehensible. (How can you subtract a cell line?)

8. RNASeq results:

- a) When you describe the RNASeq results you say "Eighteen representative..genes were found". Why "representative"? Which other genes were found? Or were only 18 found altogether? If it was only 18, and the FDR was <0.05 , then there is nothing significant at all. You are comparing about 9000 genes, so with an FDR of 0.05 you expect at least this many false positives.
- b) FDR <0.03 is not appropriate for RNASeq data, please delete this table and discussion.
- c) Changes of less than 2-fold (or, at a stretch, 1.5-fold) are also usually judged not to be biologically significant.
- d) Check the annotations, using the Tb927 numbers. If you do this you will discover that you have even fewer regulated genes than you thought. NO fewer than EIGHT of them are almost identical (HSP83). The various procyclins and PAGs also have some shared sequences. There is also no evidence that Tb427.02.4950 has anything to do with translation.

9. Differentiation experiments. These are done with trypanosomes that are presumably not capable of making stumpy forms or differentiating to growing procyclic forms, so interpretation is virtually impossible.

Why are these experiments in the supplement, but nevertheless highlighted in the Abstract? From the data it seems that:

- a) S5 The knock-down results in an increase in procyclin and PAD1-2 mRNAs. The increase is, however, very modest.
- b) S5 There is no effect on MyoB but I am not sure why this gene was picked. Was it increased in procyclic forms in the recent RNASeq experiments?
- c) S5 Panel C: There is a slight increase in surface procyclin expression. There is also a decrease in VSG but that was already shown.
- d) S5 Panel D is invisible so I can't comment

After knocking down the expression of SBF2PH there is a growth defect, which might explain some or even all of the effects seen. Numerous studies - including decreasing VSG synthesis - have shown slight increases in some stumpy-form RNAs after growth retardation. This is not necessarily anything to do with differentiation, it could just be a stress response. Cite the relevant papers, especially Batram et al.

In summary, the effects seen here do not show that SNF2PH is a regulator of differentiation.

10. Finally, the authors indeed work on differentiation-competent trypanosomes.

Overall the experiments in Fig S6 do not demonstrate any convincing link between AMPK α 1 and SNF2PH, and have nothing to do with SUMOylation. So I'm not sure why they were included. If you want to include them to get rid of them, merely state in the results section that you could find no convincing evidence that SNF2PH is phosphorylated. Also if you include them the following changes are needed.

Panels A and D should be put together. Otherwise it is impossible to see whether there is an effect of the knock-down or not - please don't force the viewers to jump from one panel to the other in order to see that there probably isn't. (Also fix the labels on panel D).

Panel B shows that 5'AMP treatment causes a decrease in SNF2PH expression (presumably because of growth arrest) and an increase in phospho-AMPK α 1

Panel C is an immunoprecipitation of SNF2PH after AMP treatment, with and without inhibitor.

This is a single experiment and it is impossible to tell which phospho-Tyr band(s), if any, might be SNF2PH. IT all looks likel background to me. (I could see no difference between the lanes with and without inhibitor.) TThis panel was wholly unconvincing. I got a second opinion from a student, who just "WHAT? How can they claim that? More seriously - if the lower panel is the same blot, how come the markers have moved? If you don't even know where the markers are, how can you possibly even guess which background band might be the one you are looking for?. The claim "The phosphorylated band in upper panel corresponds to SNF2PH (arrow, lane 3)" is not justified, and neither is the conclusion "Cells treated with 5'-AMP contained phosphorylated SNF2PH". Panels E and F are missing a vital control - compound C alone - so cannot be interpreted.

11. Methods:

The rationale for including some methods, but not others, in the main text, escapes me. It would be better to mention all of them in the main text and refer to the Supplement for details. Meanwhile the Supplementary methods are missing quite a lot of details, such that repeating the experiments would not be possible.

FORMATTING ISSUES AND OTHER POINTS

Fig S4 also has a confusing legend in the inset. "hs vs" is peculiar. Why not just label these with the cell type? Please show the individual curves for the replicates as cumulative cell number. That way the degree of variation will be seen - at the moment this is invisible except for the last time point.

Some of the legends and statements are not very accurate - with a tendency towards over-statement. Examples are:

Fig 1A legend "highly expressed" - relative to what? The actual expression level was not measured. Is the upper band in bloodstream forms the SUMOylated version? Is there SUMOylation in porocyclic forms as well?

Fig 1B: "full" depletion is not an accurate description, the protein has merely dropped below the level that is detectable with the antibody - the detection limit of which was not measured. There's almost certainly still some protein there.

Fig 2A legend - the control panel is below, not to the right.

Fig 4 - "fold enrichment over no antibody control including a nontagged cell line". Doesn't make sense.

Page 7: "which is probably due to the highly dynamic nature of protein SUMOylation." This should be stated as a hypothesis, not (at this stage) an explanation.

"precise recognition determinants for SNF2PD are unlikely to resemble metazoan..." Well, no, especially as the domain resembles one from plants. How about targets from plants?

Would it be possible to eliminate some of the abbreviations? They are rather numerous and make the manuscript very difficult to read. For example "HSF" is very commonly used for "heat-shock factor", "PTM" isn't used that often. "IPed" is horrible...

References should be given for the identities of the different proteins involved in (de)-SUMOylation, in cases where they were previously studied. For the remainder, the evidence that this is the right protein should be stated briefly.

"mature or pre-spliced rDNA+ 780" DO you mean rRNA? Whether or not it is rRNA, what is the +780?

Oligonucleotide Tables - Give TritypDB numbers please.

TYPOS and OTHER ODDITIES

There are quite a lot of textual errors, especially in the Supplement (Did you think the reviewers wouldn't read it?)

Here are some examples but the whole text needs scrutiny.

p4: to positive regulate" (->"positively" or "to enhance")

p5 "unlikely to resemble metazoan." Dangling adjective. Should be "those from metazoa" or "metazoan ones".

"included too in transcriptional activators as HsSMCA5 and MsSMCA1". I think "as" should be "like"?

"The SNF2 N domain is conserved across the kinetoplastida, importantly, this factor... Sentence changes subject half-way through. Which factor?

"conserved with Histone Methyltransferases" - re-write.

"not a target of SUMOylation detected as a single protein band" doesn't make sense.

"Genomic was extracted by the phenol-chloroform method, previously treated with RNase" ! Also, in the same paragraph, "shared chromatin"

I was also puzzled by "Percentage of immunoprecipitated product was referred to 10% of input." Surely it should be referred to 100% of input, otherwise it's misleading?

"there are at least two major sites for SUMOylation in.SNF2PH N-terminal." N-terminal what ? (Should be "the N-terminus").

"downstream of the active VSG-ES to the telomeric VSG221 gene" ? Sentence also has mixed tenses.

"while in silent promoters is detected"

"showed both enrichment with the highest number of reads" - the "both" is hanging there without anything extra.

"suggested involvement of this transcription factor in.." You haven't actually shown it is a transcription factor.

Page 12 "Consecutively," - I think you mean "Consequently".

1st Revision - authors' response

30 July 2019

Referee #1

In this manuscript, Saura and colleagues build upon the recent description of a highly SUMOylated focus (HSF) associated with monoallelic expression of the Variant Surface Glycoprotein (VSG) in Trypanosoma brucei by exploring the function of a protein, SHF2PH, which they show is SUMOylated, whose expression is developmentally regulated, that interacts with the VSG transcription locus (as well as other loci), and whose loss affects VSG expression and differentiation. A great deal of new data on SNF2PH is presented, much of it with rigour....

We really appreciate the comment of this review when he acknowledged the above.

...One facet of such a function would be ensuring continued expression of the VSG, perhaps through modification of the chromatin in the VSG expression site in ways similar to found at other loci. If so, such functions appear consistent with evidence linking PH domains and development in other organisms.

We totally agree with the explanation proposed for the results we presented in this

manuscript. Whilst we have probably not defined the function as the reviewer specifically does, our intention was to transmit this precise idea. Thus, in the new manuscript, we have incorporated in the discussion section (page 20) a similar phrasing trying to convey this particular explanation.

I suggest that the authors need to alter the text to provide a clear explanation of what role or roles they suggest are played by SNF2PH in T. brucei; in addition, if they are proposing a specific role for SNF2PH in transcription of the VSG expression site, distinct from differentiation, they need to provide a clear explanation for the evidence for this proposal.

We have included in the new manuscript changes throughout the text to clearly explain SNF2PH function, although our new data suggests that VSG positive transcription regulation is not a distinct function from differentiation; rather both functions are linked. We think we provided new evidence for that in addition of the previous one.

1) CD analysis: VSG expression is a fundamental feature of the parasite bloodstream form (BF) and the fact that SNF2PH contains a homeodomain function demonstrated, in eukaryotes from yeast to human, to bind to Histone 3 modified-tails by a considerable number of publications (Mouriz et al., 2015; Musselman and Kutateladze, 2011; Saldivia et al., 2016; Sanchez and Zhou, 2011; Shi et al., 2006; Watson et al., 2012; Wysocka et al., 2006).

2) New RNAseq analysis of procyclic insect form cell lines overexpressing SNF2PH, led to specific derepression of bloodstream form stage-specific genes, including silent telomeric VSG-ES (BESs) and housekeeping genes (Glucose transporters, Glycosomal proteins, Glycolytic pathway, etc). Importantly no upregulation of VSG basic copies was detected, suggesting that SNF2PH function is specific and does not merely disorganize/open chromatin everywhere (as occur with other chromatin factors). These results suggest a specific function of SNF2PH to regulate positively the expression of the VSG-ES, since SNF2PH overexpression in the insect form induced telomeric BESs genes expression without affecting basic copies of the VSGs.

3) New global SNF2PH ChIP-seq analysis allow us to locate SNF2PH in precise positions in the chromatin (Fig 3C), Demonstrating that is associated to promoters of developmentally regulated genes as EP procyclin GPEEPT (including very interesting loci as the H3V, a histone variant linked to VSG monoallelic exclusion (Muller et al., 2018), or cluster of tRNA genes located in 7 chromosomes, sequences known to work as a global chromatin insulators to organize open and close status of the chromatin (Raab et al., 2012; Van Bortle et al., 2014).

This result altogether, strongly suggests that positive regulation of VSG transcription by SNF2PH is coupled to positive regulation of expression of housekeeping bloodstream genes. We hope we provide considerable and solid evidences to support this hypothesis through the manuscript. One of these new results (that also argues against the possibility of been a artifact or secondary effect due to loss of viability) is that overexpression of SNF2PH in the insect form increased the expression of VSG and ESAGs (BESs) and bloodstream developmentally regulated genes, while overexpression of mutant SNF2 Δ PH lacking of the PH domain, did not.

In the new manuscript the implication of these results are more clearly explained, to improve clarity of these new insights.

1. Page 7 and Fig.1. What do they authors mean by 40% overlap between GFP and SNF2PH signals, and 50% overlap between SNF2PH and SUMO signals; do they mean that they see such overlap in only this proportion of cells analysed, or do they mean they mean that they detect such an extent of overlap within each cell's nucleus (in which case a measure of variation should be provided)? Clarification here is critical in order to understand if SNF2PH always localizes to the VSG expression site and HSF, or if this localisation is only seen in some circumstances.

The percentage provided is referred to the proportion of cells analyzed that showed colocalization. In the new manuscript this aspect is now more clear throughout the manuscript and most importantly, we have included a whole set of statistic analyses colocalization for each of the nuclear structures (Figure EV1), the ESB, the HSF and the active VSG-ES tagged with the GFP-LacI. However, it is worth to mention that these double indirect IF microscopy analyses are not easy to perform and include a number of limitations. Probably, the most important is the fact the both, the ESB and the GFPtagged VSG-ES are detected in just a proportion of cells, for reason that we just start to understand. For example, the ESB undergoes a nuclear dynamic from nucleolar periphery positions (where can not be distinguish from the nucleolus) to the nucleoplasm where it is clearly detectable. In sum, as published before (Navarro & Gull, Nature 2001, Landeira et al. (2007)).

In addition, the GFP labeled active ES is detected in 40- 60% of cells depending of the cell line. Similarly, the ESB is detected in about 60% of nuclei, thus just the probability of both detection in a single nucleus is in about 36% of the cell population. Now, colocalization analyses of any of these nuclear structures to a protein under study is usually restricted to low percentages.

Moreover, SNF2PH is showed to be SUMO-conjugated, and by ChIP analysis and PLA we demonstrate that the modified version is the one associated to the active VSG-ES promoter and therefore, to the ESB where transcription of the VSG-ES occurs. Thus, it seems quite possible that many of the epitopes recognized by the anti-SNF2PH antiserum could be partially hidden by this large protein modification (12 kDa SUMO peptide). Probably, this PTM or possible any other possible protein modification associated with the SNF2PH that positively regulate VSG transcription affect to the antibody recognition, resulting in a low % of cells showing colocalization with either the ESB, the GFP-tagged VSG-ES or the actual HSF.

2. The analysis of the ChIP data for SNF2PH is very unclear and should be improved. Most importantly, the authors should show the pattern of ChIPseq signal across the active and silent VSG expression sites, as well as at important other loci (rRNA, procyclin), rather than levels of enrichment using small parts of these regions (as in Fig.3D), which may give a distorted impression of enrichment. Once these data are shown, the qPCR of necessarily small parts (Fig.3B) can then be seen in the wider context. In addition, presentation of the analysis needs to be made more accessible:

Trypanosome genome contains at least 15 different telomeric VSG-ES loci (BES) that share high homologues sequences, particularly at the at the promoter region (Hertz-Fowler et al., 2003), therefore, any alimnet tool used to allocate the reads from ChIPseq data will anneal the sort DNA read onto many different VSG-ES at random, instead to the one the protein is actually enriched (). This high sequence homology shared among all telomeric VSG-ES hampered many molecular studies of transcription until we had transgenic technology at our disposal, so inserting unique sequences (reporter, selectable markers, etc.) within different position along the VSG-ES to be able to investigate properly the complex regulation of this multi allelic family (see for example, Navarro & Cross, 1994, MCB). That is why we began our analysis with DRALI, a cell line that contains inserted closed to the promoter unique sequences, two reporter genes, to use as specific probes to locate a given protein by ChIP-qPCR (Figure 3B).

Additionally, we have recently utilized minor polymorphism sequence differences in the promoter area among different VSG-ESs (BESs), which allowed us to distinguish the position of a given protein in ChIP experiments (Lopez-FarfaÅLn, et al., Plos Pathog. 2014). Thus, exploiting now NGS possibilities combined with availability of the actual sequence for each telomeric VSG-ES (BES) from the 427 *T. brucei* strain, we are able to allocate with unprecedented precision the position that occupy a protein in the chromatin of each one of the 15 different VSG-ES (BESs), sequence published in (Hertz-Fowler et al., Plos One, 2008). Thus, although different VSG-ES share high homologues sequences, the few polymorphism nucleotides in particular regions, allowed us to identify which one of the telomeric VSG-ES is the one the protein under study is

occupying. Thus, to achieve discrimination among 10 different VSG-ESs, we used fragments ESPM1, and 4 since sequence analysis from PCR fragments obtained from genomic DNA showed at least 10 different sequences with polymorphism of the nucleotides (5%) (referred in the manuscript as ESPM1 & 4). This was described previously by (Lopez-Farfañán, *et al.*, Plos Pathog. 2014) where this methodology was explained and demonstrated, which allowed us to determine the position of chromatin-associated proteins that contains SUMO modification, around the VSG-ES promoter. Below, a section of Figure S5 from the cited publication:

To explain that in the new manuscript, we have included new Figure 3 A (a schematic representation of loci of interest) and Figure 3 D (schematic representations of the VSG-ES promoter area, also when the promoter are duplicated in tandem downstream referred as dw promoter). Additionally, we provide a new histogram with the updated VSG-ES nomenclature (BES) and clustering data in an appropriate manner considering the nomenclature of the BES and the sequences described in (Hertz-Fowler *et al.*, 2008).

in Figure 3 D & E:

In addition, a clear description of the methodology used for the PCR reads alignment on the BES promoter areas and thus to discriminate among the different VSG-ESs, is included in the new manuscript, in page 10:

Also in Material and Methods this is now described properly.

In sum, we think that in the new manuscript we have provide extensively more explanations and better presentation of the data in a wider context and making all these complicated set of data more accessible and we thanks the reviewer we feel that our modified manuscript is now much improved

In addition, we also added the following sentences to the result section to clarify interpretation of the results:

Page 12: “Figure 3B, shows SNF2PH is enriched in the active VSG-ES (BES1) locus by ChIP qPCR analysis using unique sequences as the RLuc reporter inserted downstream of the promoter and the VSG221 gene at the telomeric end. However, we wish to investigate in detail a possible SNF2PH occupancy at the promoter adjacent area, nevertheless highly homologues sequences shared among most of the VSG-ESs (BES1,

2, 3, 4, 7, 7dw, 10, 10dw, 12, 13, 15, 15dw, 17, 17dw described previously (Hertz-Fowler et al., 2008)). Next, using Bowtie software, alignment of the reads were assigned to the BES promoter index file, and the number of reads aligning to each BES was represented in Figure 3D. This data showed SNF2PH is enriched to the active BES1, at the ESPM1, which correspond to the core promoter of the active BES1 (VSG-ES221). SNF2PH was detected to a lesser extent in other BES promoters suggesting that SNF2PH is controlling inactive promoters as well (Fig 3D). Interestingly, the increase of read count was at the ESPM1 fragment region where the actual ES promoter is located, suggesting SN2PH is associated with the active core ES promoter rather than the upstream promoter region (Dataset EV1). Together these data indicate that SNF2PH is located in several BES promoters, however it is most enriched at the active VSG-ES promoter (ESPM1 of the BES1) (Fig 3E).”

Consequently, figure order has been altered in this section; Fig 3C (ChIP-seq data using the SNF2PH antiserum of other genomic loci), Fig 3D (Schema of VSG-Promoter region) and Fig 3E (ChIP-seq data using SNF2PH at the VSG-promoter region).

- How do the PCR regions in Fig.3A relate to the VSGs detailed in Fig.3B?

Indeed, the series of PCR fragment that map the ES promoter region were not used except for the 1 and 4. In the new manuscript we have only include those two for clarity. These are the PCR fragments that map in all VEG-ES promoter regions because the sequences share high homology and we used to locate the position of a protein in ChIP experiments. The polymorphism described above among the BESs were used previously to differentiate between active and inactive promoters (Lopez-Farfan et al., 2014). In order to clarify the correspondence of the PCR regions in Fig 3A of the new MS we provide a schema of some inactive expression sites included in Fig 3B. (VSG121, VSGJS1 and VSGVO2), as well as other Pol I (18, rDNA, rDNA promoter, rDNA spacer, procyclin, EP promoter) and Pol II (Tubulin) transcribed loci.

Figure 3A. Schema of tagged DRALI cell line: The Double Renilla-luciferase reporter gene (RLuc) was inserted downstream of the Active VSG221-ES promoter and the firefly Luciferase gene (FLuc) was inserted downstream of an Inactive VSG-ES, excluding other inactive VSG-ESs (VSG121, VSGJS1 and VSGVO2). Schematic representations for other chromosomal loci (rDNA promoter, Procyclin and Tubulin) are shown. Color code: grey (Reporter), green (active VSG-ES), red (Inactive VSG-ESs), blue (procyclin locus). Arrow (Promoter).

The ChIPseq signal across the active and silent VSG expression sites is limited by several problems, including high homologous sequences, which justify the use of short regions of high polymorphism. In the new manuscript, we have included a whole VSGES

alignment, including other telomeric and important genomic loci.

- Why has the active VSG ES promoter region been duplicated in Figs.3A and C?

That was unnecessary, in the new manuscript this has been corrected.

- What is the meaning of the numbered BES (e.g. BES4), dw, DES, VO2 in Fig.3D?

This is old and new nomenclature for the VSG-ES, now numbered BESs. We apologize for that, in new manuscript the new nomenclature is been preferentially used (BES, instead of DES or VO2). However, in some cases we maintain both, the new and old nomenclature too since we are all very familiar. For example, in the Figure 3D the histogram labels referring to VSG-ES promoter sequences are properly labeled using the new nomenclature as BES1, BES2, etc (Fig3D & Fig 4C). In addition, in the manuscript, when necessary we have included both nomenclatures, In pages 10, 11, 14, 15, 24, 28 the active 221 VSG-ES is also referred as BES1. Other examples as the VO2-ES (BES2), DES (BES3), VSG427-15 (BES10) and VSG-14 (BES8) etc , thus we mention both nomenclatures when necessary. Also in Figure legend (VSG121 (BES6), VSGJS1 (BES13) and VSGVO2 (BES2)).

Dw, DES, VO2 are denominations for other inactive VSGs (nomenclature should be clarified).

We have now included a scheme in the Figure 3D to explain the position of the downstream promoters (dw) and also explain in the text. As mentioned above, the old and new nomenclature has been clarified in the current manuscript.

- How was the ChIPseq data analysed (I could not find this in the methods)?

In the new manuscript, we included ChIPseq methodology and data analysis in the main text. In addition, the methodology used for the alignment employing Bowtie software is now described in detail in Material and Methods:

ChIP-seq analysis in Figure 3 E. To discriminate among the BES promoters we carried out ChIP-seq analysis using selected PCR ES Promoter Mapping (ESPM) regions known to have sequence polymorphisms among different BESs, as previously described (Lopez-Farfan et al., 2014). The ChIP using anti-SNPF2PH and fixed chromatin was performed as indicated above. Immunoprecipitated DNA and total input DNA were used to generate the libraries sequenced on an Illumina NextSeq 500 platform producing about 10000 reads of 36 nt. A pool of amplicons of sequences of the promoter regions ESPM1 and ESPM4 (Fig 3D, defined with the primer include in Table) containing the corresponding sequences from the BES1, BES2, BES3, BES4, BES7, BES7dw, BES10, BES10dw, BES12, BES13, BES15, BES 15dw, BES17, BES17dw and 18S, U2, C1 as control were combined together in a single lane to built a index file ebwt, next the alignment was done with bowtie -S -n0 command to consider a 0 nucleotide mismatch to distinguish among few nucleotide sequence differences in each BES, as described before (Lopez-Farfan et al., 2014). The actual number of reads aligned on each BES in represented in the histogram of Figure 3E.

3. The validity of the conclusions reached using the K2A mutant of SNF2PH appear questionable. Though it is clear that SUMOylation of the mutant protein is reduced and leads to loss of ChIP to the active VSG, the suggestion that this mutation 'drives SNF2PH occupancy towards inactive VSG-ES promoters' is not well supported. Most importantly, how can the authors exclude that the mutation renders SNF2PH inactive, rather than merely impairing SUMOylation? If the protein was inactive, perhaps through incorrect folding, the conclusions reached would be incorrect.

The phrase '*drives SNF2PH occupancy towards inactive VSG-ES promoter*', was changed. However, the data clearly suggest the mutant K2A did not renders a inactive protein neither incorrect folding, since the mutant is still detected bound to the inactive

promoters & control loci, and because the protein is stable expressed, by WB analysis (Fig EV2). Degradation and reduction of ptoirin turn over will be the inevitable consequence of Incorrect folding, contrary of we showed by WB of HA-SNF2PH K2A compared with the HA- SNF2PH expressd in the transgenic cell lines (Fig EV2). This result demonstrates that the level of expression is similar for both the mutant and WT proteins and suggests that the mutation does not affect the stability of the protein. Further, the protein was still active since ChIP analysis detected the mutant at inactive VSG-ES promoters (Fig 4B), even increased (ESPM1 in BES12, Fig 4B). Neither would maintain binding activity to 18S, U2, SL control loci, where the mutant K2Awas detected as well as wild type protein..

4. What happens to the expression of VSGs from the silent expression sites after RNAi? At the moment, the only description is that there is variation between clones. What exactly do they mean, and why did they not perform RNAseq of the different clones to clarify these confusing effects?

In the new manuscript we provide new RNA-seq data for independent clones (Dataset EV3 and Figure S4E) and we have discussed this results in detail for clarity as well. Initially, analysis of SNF2PH depleted clones showed the effect on BES reported genes by RT-qPCR analysis inserted downstream of both active and inactive BES promoters indicating that inactive promoters were derepressed in some clones but not all clones (Fig S4). Together with the new RNAseq analysis showed in Figure S4E we clearly demonstrated that SNF2PH depletion induced lost of silencing in some of the inactive BESs but not all, as recently reported by *Murray et al.* (Nature) after H3V H4V KO.

Figure S4E. Scatter plot showing telomeric BES expression by RNA-seq analysis of SNF2PH depleted cells. Inactive VSG-Ess (BESs) upregulation occurs only in a subset of the inactive telomers, while other inactive BESs appear to be unaffected. Thus, BES deregulation seems to be variable within each of the clones analysed, consistent with different reporter activities detected by the VSG-ES promoters (Fig S4C). SNF2PH is included as KD control. Data from two independent clones are presented compared with two RNAs isolated from the parental cell line.

Minor criticisms:

Page 5, paragraph 1. SNF2 proteins are helicase-like factors, most of which are DNA dependent ATPases; not all are involved in chromatin remodelling, as is being suggested. A recent, broad review should be cited.

We have included some new references about this, as well as commentaries in the Results section as suggested, to read:

Structural CD predictions suggest that Tb927.3.2140 is a member of the Snf2 family (Sucrose Nonfermenting Protein 2) from the SF2 helicase-like superfamily 2 of chromatin remodellers (Clapier et al., 2017; Giles et al., 2019; Yan et al., 2019)[15-17], which regulate DNA accessibility to facilitate central cellular processes as transcription, DNA repair, DNA replication and cell differentiation (Clapier et al., 2017).

However, to avoid confusion in the new version this ophrase is removed and a detailed Conserved Domain search analysis lead us to as include in the results (as recomentded) this new paragraph, to read: e

Searching for Tb927.3.2140 homologues using DELTA-BLAST against UniProtKB/SwissProt database, identified a protein member of the SWI/SNF family, SMARCA1 (e-value, 4e-157) (SWI/SNF-related matrix-associated actindependent regulator of chromatin subfamily A member 1) also known as the global transcription activator SNF2L1 (length, 1054aa), (homonyms SWI; ISWI; SWI2; SNF2L; SNF2L1; SNF2LB; SNF2LT; hSNF2L; NURF140) all described to be involved in transcription for either gene activation or gene repression [16].

Page 5, paragraph 2 is data and should be removed from the introduction and left to the results.

As suggested, most of this paragraph is moved in the new manuscript to results section page 6.

Page 5. What do they authors mean by nucleosome remodelling 'translocates' DNA?

The mechanism by which these enzymes couple ATP hydrolysis to translocate the nucleosome along the DNA, is the focus of many recent publications (Li M, et al., Nature. 2019), revived in (Nichols & Corces, Nat Struct Mol Biol. 2018). The structures of Snf2 in the ADP-bound state induce a one-base-pair DNA bulge at superhelix location, with the tracking strand showing greater distortion than the guide strand. The DNA distortion propagates to the proximal end, leading to staggered translocation of the two strands (Li M, et al., Nature. 2019). Several very recent publication as well as consolidated work described this process in detail, se for example (Eustermann et al., 2018; Farnung et al., 2017; Glover et al., 2013; Li et al., 2019; Saha et al., 2002; Whitehouse et al., 2003; Yan et al., 2019)

Page 6, Table S1. What control affinity purification was performed to allow the authors to list the factors in the table?

The affinity purification of SNF2PH interacting partners was carried out in a cell line that ectopically expressed 3HA tagged SNF2PH. As a control, a parental cell line protein extract processed in parallel was used to subtract contaminant proteins out of the HAtagged cell line.

Referee #2

Monoallelic expression of Variant Surface Glycoprotein, with transcription by RNA polymerase I, is critical for survival of bloodstream-form trypanosomes and is a key element in the control of antigenic variation. VSG transcription is known to take place in a focus called the "expression Site Body" which s outside of the nucleus. The authors previously found that SUMO is concentrated on the ESB and that SUMO ChIP pulls down the VSG promoter region. They identified a SUMO E3 ligase, SIZ1. Knockdown of SUMO, SIZ1 or UBC9 decreased polymerase I transcription of the expression site. There is therefore convincing evidence that SUMOylation plays and active role in promoting VSG transcription. In the current paper the authors set out to try to find out which SUMOylated protein(s) are responsible for the transcription effect. They used cells expressing His-tagged SUMO to pull down covalently modified proteins and identified them by mass spectrometry. Among the candidates was a protein with promising-looking domains that they have called SNF2PH.

They show convincingly that:

1) SNF2PH is spread over quite a large part (but not all of) of the nucleus, overlapping with the ESB and the nucleolus.

- 2) Some of the SNF2PH is SUMOylated (the proportion is not measured). An N-terminal fragment could also be SUMOylated by a lysate from *E. coli* expressing the trypanosome enzymes - a very useful experiment and set-up for future work.
- 3) ChIP using antibody to SNF2PH, followed by PCR, indicates that it is enriched not only on the active VSG promoter but also promoters for U2 (pol III), rRNA (also pol I) and the spliced leader or a strand-switch region (pol II). These results suggest that it might be a general mark for open chromatin. There was much less enrichment on inactive VSG promoters.
- 4) The results were followed up by ChIPSeq but most of the results are not shown. No data for any promoters other than VSG ones is shown.
- 5) Next the authors (I think) replace the wild-type gene with either an HA-tagged version, or an HA-tagged version that lacks lysine 2. They could only mutate one allele, suggesting that Lysine 2 is important. They expect that Lysine 2 is SUMOylated but I don't think they formally showed it. The mutant protein no longer preferentially localise to the active expression site promoter, suggesting that SUMOylation is important for association with the ES but not with other promoters (though I didn't understand some of these results).
- 6) The authors look for proteins associated with SNF2PH and indeed detect both pol I and pol II - associated proteins.
- 7) Depletion of SNF2PH results in a decrease in VSG mRNA, consistent with a need for the protein in VSG transcription.
- 8) Next the authors looked at effects of the RNAi on overall gene expression. RNASeq yielded extremely few significant changes, but one of them was an increase in EP procyclin mRNAs, which is diagnostic of the very beginning of differentiation. Unfortunately, it is also commonly seen after various other treatments that impair cell growth. Moreover, it is known that attenuating VSG RNA synthesis can initiate differentiation (Batram *et al.*, not cited). The effects of SNF2PH depletion on differentiation-type gene expression are therefore likely to be secondary. There is no evidence that SNF2PH is a direct regulator of differentiation.
- 9) The Supplement also contains some data relating to phosphorylation of SNF2PH. These results are unconvincing and should either be removed, or included as negative data.

All of the results from points 1-7 are good and interesting, although my interpretation would not be quite the same as the authors'. I conclude from that SNF2PH is localised to many different promoters, and probably generally associated with open chromatin, but that SUMOylation is important specifically for its action at the VSG expression site. This in itself is very interesting.

We would like to thank the referee for his/her comments, which have helped us to improve the quality of our manuscript. We appreciate the ample and constructive criticisms because, by addressing their questions and concerns in the new manuscript, we believe the work was not only considerably improved but also we were able to provide additional new and decisive results supporting a key function of SNF2PH.

One missing link is what difference the SUMOylation makes. This could be examined by a comparison of the interactions of mutant and wild-type, HA-tagged SNF2PH. Whether or not there are differences, this would make a stronger paper.

In the new manuscript we have included a deeper analysis of the K2A mutant showed in figure 4 and Appendix Figure S4. We hope all the new detailed incorporation of data and explanation clarify these questions.
Apart from that I have numerous detailed criticisms, all listed below.

GENERAL

A. Data availability:

a) Where are the database deposition numbers for the sequencing results? (Maybe I

missed them?) where are the detailed supplementary Tables for the OMICS results?

As requested, in the new manuscript we provided output data for all OMICS results detailed in:

- Table EV1: data for Table S2 (Pull down SNF2PH in PF).
- Table EV2: data for Table S3 (RNA-seq data SNF2PH KD in BF).
- Dataset EV1 (Annotated peaks for ChIP-seq SNF2PH).
- Dataset EV2 (Annotated peaks for HA-ChIP-seq library SNF2PH WT and mutant).
- Dataset EV3 (RNA-seq data for VSG telomeric expression upon SNF2PH depletion in BF).
- Dataset EV4 (RNA-seq data for SNF2PH overexpression in PF).

b) Mass spectrometry: Please supply full data for the mass spectrometry in Tables S1 -and S2, including the controls.

Were the peptide numbers for both purifications identical? If not, which was chosen for the simplified Tables? Please also supply the peak quantitation for all replicates and do a statistical analysis with an appropriate program. This should be possible if the analysis was done in the past 4 years, and may reveal additional interactions

In the new manuscript we clarify that this work is not focus in proteomic of SUMO, rather that was the way we found SNF2PH, thus we explain that in page 5, to read:

To identify abundant SUMOylated proteins we performed a non-exhaustive proteomic enquiry utilizing BF protein extracts from a cell line expressing an 8His-HA-tagged SUMO (see Material and Methods). LC-MS/MS analyses of His-HA-affinity-purified extracts robustly identified several proteins (see Appendix Table S1). Particularly interesting was Tb927.3.2140, (length 948 aa)....

The reason why we include the list is to provide a snapshot of hypothetical candidates as a starting point to carry on functional studies based on SUMO regulated factors that control VSG expression. For this reason, we selected the transcriptional activator SNF2PH given was consistently identified in SUMO-conjugated proteomic purification independent experiments, depletion by RNAi reduced active VSG expression, later ChIP located in a SUMO-dependent manner.

Unfortunately, we are not able to provide full datasets for Table S1 since we have a manuscript with other collaborators submitted recently to publication. Nevertheless, we make available the number of peptides and the peak quantitation for Table S2 provided by Scaffold 3 Proteome Software.

c) RNASeq: Please supply the raw read counts for the RNASeq and show the full statistical analysis for all genes using an established package (see below).

As requested, raw read counts for RNA-seq data, together with the full statistical analysis using EdgeR are provided in Table EV2.

STATISTICS

Standard errors should not be used at all for these sorts of results. Readers are interested in the variation in your measurements, not the possible error in your estimate of the mean. If you had enough measurements to know you had a normal distribution, standard deviation would be appropriate. But in this case you do not know you have a normal distribution since usually there are only about 3 measurements. Bar graphs with error bars are no longer regarded as appropriate low numbers of replicates, since they fail to illustrate the real distribution of the data. Please show dot plots for individual measurements. This often makes results more convincing because big error bars can be caused by just one out of 3-4 measurements. (You can add a line for the mean, if you like.)

We agree with referee in the fact that bar graphs with standard errors are not appropriate for a lower number of replicates. Although most of the assays include 3 or more independent measurements, it is difficult to avoid variability in ChIP assays. To improve that variability, a very problematic locus as the pseudo VSG (due to additional genes in the genome with high homology sequences), has been removed from the analysis. In addition, we supply Box plot representations for Fig 3B, Fig 4A, D and Fig

5B, to better illustrate the data. In our opinion, representing ChIP data in box plot is quite unusual, however, if the referee #2 thinks it is more clear, we have no problem to include it in the new version.

FIGURE FORMATTING

Please give all text labels similar sizes without laterally (or vertically) squashing them to fit. It should be possible to read the whole thing at the same magnification. 1B, 2E, 5C, D, E and S4E are the worst, but not the only, offenders; 2B has italics for no obvious reason. This is just laziness (cut it, paste it, and squash it no matter what it looks like) and it is extremely irritating.

In the new manuscript the figure fonts and sizes are adjusted as suggested.

DETAILED COMMENTS

1. Title: This has to be changed completely,

a) The authors present no evidence that *T. brucei* SNF2PH is a chromatin remodeller.

As reviewer recommended we changed the title in the new manuscript, to read:

A homeodomain protein SNF2PH promotes VSG and bloodstream stage gene expression in African trypanosomes

b) The differentiation data are weak - delete the second half of the title completely (Anyway, most - indeed perhaps all -stages of trypanosomes are infective for either mammals or *Tsetse flies*.)

We have reduced significantly the AMPK data to a more focus description of the differentiation data. Importantly, in addition of the data we provided initially supporting SNF2PH function controlling developmentally regulated gene expression, in the new manuscript we have incorporated new results that strongly support SNFPH function in the regulation of stage specific gene expression. First, the ChIP-seq analysis on the whole genome demonstrates that SNF2PH is bound to chromatin in genes developmentally regulated (as EP and GEEP and all the ES associated genes), to higher extend than to any other loci (Fig 3D). In fact, SNF2PH is found to be enriched in both active and silent transcriptional states, as other Chromatin remodelers do in mammals (CHD for example). In our opinion, we provide of additional results that point to the same function, but perhaps the experiment that clearly demonstrate this function is the gain of function achieved after over expression of SNF2PH in the insect-procyclics

form when is normally not expressed. Figure 6E & F clearly demonstrate that expression of the wild type SNF2PH in the procyclic form leads to upregulation of bloodstream for specific gene expression, while expression of the a deletant mutant lacking of the PHD, SFN2DeltaPH did not, ruling out the possibility of any artifact and demonstrating the is the PHD the domain in the protein responsible to direct the chromatin remodeler (positive) function, as a transcriptional activator, towards the bloodstream form specific set of genes.

2. Introduction

The results from the previous paper on SUMO and VSG transcription should be described in a lot more detail – beyond the single sentence that is currently there.

As suggested, in the new manuscript we have included a description in the Introduction and Results and Discussion sections of the new manuscript previous results published already, where it was described SUMO involving in VSG transcription (Lopez-Farfan et al. 2014 Plos Pathogens).

In the Introduction page 4, to read:

In *T. brucei* BF, SUMO-conjugated proteins were detected highly enriched in the nucleus in a single focus (HSF) associated with the ES body (ESB) and enriched in the active VSG-ES chromatin, suggesting chromatin SUMOylation acts as a positive epigenetic mark to regulate VSG expression (Lopez-Farfan et al., 2014). Chromatin SUMOylation to the active VSG-ES locus is required for efficient recruitment of RNA polymerase I in a SUMO E3 ligase (TbSIZ1/PIAS)-dependent manner suggesting protein SUMOylation facilitates the accessibility of additional transcription factors (Lopez-Farfan et al., 2014).

In the Results page 5, to read:

Protein SUMOylation is a hallmark of VSG epigenetic regulation at the level of chromatin and nuclear architecture (Lopez-Farfan et al., 2014). The highly SUMOylated focus (HSF) detected by a specific mAb against TbSUMO in the nucleus of bloodstream form (BF) trypanosomes was recently associated with the nuclear body ESB (Lopez-Farfan et al., 2014), the site for VSG-ES monoallelic expression (Navarro and Gull, 2001). Recognition of HSF together with the detection of highly SUMOylated proteins at the active VSG-ES chromatin by ChIP analysis, suggests that a number of SUMOylated proteins are mechanistically involved in regulation of VSG expression (Lopez-Farfan et al., 2014).

In the Discussion page 20, to read:

The VSG transcriptional state is maintained through several generations and is the product of epigenetic mechanisms and post-translational modifications (PTM) that mark the active VSG. Several PTMs are associated with silencing the inactive VSG-ESs, but SUMOylation is the only known modification associated with the active VSG-ES transcriptional state (Lopez-Farfan et al., 2014). A concentration of SUMOylated proteins (Lopez-Farfan et al., 2014) is located adjacent to the expression site body where VSG transcription occurs. Additionally, several chromatin-associated SUMO-modified proteins are enriched at the active VSG-ES and chromatin SUMOylation is also required for efficient RPAI recruitment (Lopez-Farfan et al., 2014).

3. Mass sepectrometry of SUMOylated proteins.

a) The list of SUMOylated protens from bloodstream forms has no overlap whatsoever with the previously published list for procyclic forms. Given that the patterns of SUMOylation in the two forms are similar (10.7717/peerj.180), this seems very odd. Although some of the proteins in the procyclic list might be high-abundance contaminants, quite a lot of them probably are not. Please comment, at least briefly. Also please, in the Supplement, say which peptides were SUMOylated. It must be clear from the MS results.

This work is not about SUMO proteomic, (additional manuscript is under review in Molecular Microbiology with the proteomic analysis in BF) to clarify that we have include in page 5 of the Results:

Protein SUMOylation is a hallmark of VSG epigenetic regulation at the level of chromatin and nuclear architecture (Lopez-Farfan et al., 2014). The highly SUMOylated focus (HSF) detected by a specific mAb against TbSUMO in the nucleus of bloodstream form (BF) trypanosomes was recently associated with the nuclear body ESB (Lopez-Farfan et al., 2014), the site for VSG-ES monoallelic

expression (Navarro and Gull, 2001). Recognition of HSF together with the detection of highly SUMOylated proteins at the active VSG-ES chromatin by ChIP analysis, suggests that a number of SUMOylated proteins are mechanistically involved in regulation of VSG expression (Lopez-Farfan et al., 2014). Therefore, identifying these proteins is a novel approach for the discovery of factors involved in VSG regulation. To identify abundant SUMOylated proteins we performed a non-exhaustive proteomic enquiry utilizing BF protein extracts from a cell line expressing an 8His-HA-tagged SUMO (see Material and Methods). LCMS/MS analyses of His-HA-affinity-purified extracts robustly identified several proteins (see Appendix Table S1). Particularly interesting was Tb927.3.2140, (length 948 aa), a protein annotated in the TriTrypDB database as a transcription activator, and which contains a conserved SNF2 family N-terminal domain.

b) Is there any other evidence for SUMOylation of any of the proteins on this list? It would be expected to result in a mobility shift on SDS_PAGE - Have any of these proteins been examined by Western blotting? Several have at least been studied before (give references).

As commented in the previous section, there are some proteins out of the Table S1 that are found to be SUMOylated in procyclic forms, such as the Structural maintenance of chromosome 4 (SMC-4, Tb927.10.740), the proliferative cell nuclear antigen (PCNA, Tb927.9.5190) and the repressor activator protein 1 (RAP1, Tb927.11.370). Experiments evidencing mobility shift are very often difficult to achieve due to the high dynamic nature of this PTM. Notwithstanding, to some of these targets we have evidences by western blot that will be submitted by some of the authors of this manuscript in several independent reports, out of the scope of this paper.

c) Please check - and correct - the annotations. There aren't many of them so it shouldn't take long. For example I couldn't find any evidence that Tb927.2.4950 has anything to do with translation. There is no evidence that Tb927.11.15830 has RNA methyltransferase activity, beyond motifs. Tb927.11.14190 has a Tudor domain, but there is no evidence that it is involved in RNA-induced silencing...

Protein annotation for Tables S1 and S2 has been corrected according to the latest release of the annotated TrypTripsDB. Please find attached the new supplied tables.

4. Figure 2, Immunoprecipitations:

a) Please show the unbound as well as the bound fractions. That way it will be possible to tell whether the immunoprecipitations were quantitative or not. If they were then the results suggest that only 0.5% of SNF2PH is SUMOylated. If they aren't quantitative no such conclusion can be drawn. This is important since the authors repeatedly claim - without (I think) any evidence, although it is likely - that the SUMOylation is highly dynamic.

To facilitate the interpretation of the % of SUMOylated SNF2PH, we quantified the signal intensity of SUMO mAb from both input samples (before and after centrifuging the whole cell lysate) and the immunoprecipitated products (SNF2PH and prebleed). Signal intensity was extrapolated to the volume relative to the percentage loaded from both input and immunoprecipitated products and the % of SUMOylated SNF2PH was referred to the 0.5 % of loaded input, leading to a 0.4 % of SUMOylated SNF2PH. Blot

Image Name	Volume (µL)	Signal	Total	Area	Total_Bkgn.	Diluted lysate (9600 µL)
1. Input non centrifuged (0.5%)	40	7258511	15168611	168300	9951983	2388475920
2. Input centrifuged (0.5%)	40	6414099	14829099	168300	9612471	2306993040
3. SNF2PH IP	20	3807631	12895831	168300	7679203	
4. Prebleed	20	2902227	11822127	168300	6605499	
5. Input centrifuged (0.2%)	19,2	2315993	11404193	168300	6187565	SUMOylated protein (%)
Background		167628	5216628	168300	0	0,41666667

quantification is shown below:

b) Since the authors have both antibody and recombinant protein, they could also very simply measure the absolute amount of SNF2PH in cells using a quantitative Western blot. That would be useful information. (This is desirable, not essential, but where else would you publish it and it isn't much work.)

We thank the referee for this comment and we agree that would be useful information. However, we consider relevant for this work that the protein level of SNF2PH is developmentally regulated, suggesting specific functions related to a specific stage.

5. ChIP and ChIP-Seq

a) Please explain the expression site terminology better, anyone who is not thoroughly versed in it would get totally lost. Explain what the pseudoVSG gene is.

A pseudogene (pseudoVSG) is a no-coding VSG gene due to sequence alteration including stop codons, no sense, etc. The pseudoVSG is located in the BES 1 (VSG-ES 221) just upstream of the VSG221 gene, and even we have used before a unique sequence, this gene fragment actually share some homology with other basic copies of the VSGs located elsewhere in the genome. Thus, similar sequences led to artifacts in qPCR analysis if the temperature is not precise. Thus, we finally decided to delete from the presented data, since the results using this PCR fragment were very variable, and so prompt to a high error.

b) The Figure legend for the first set of experiments is inadequate. It must be possible to see what everything is without hunting in the main text. What does "dw" mean? (Is this explained anywhere?)

Legend to the figure 3 has been rewritten in order to facilitate understanding of ES terminology and schematic representation of VSG-ESs. Thus, dw is defined as downstream promoter for a given ES.

c) "however it is significantly more enriched at the active VSG221 promoter region" - delete "significantly". I assume from the results that VSG221 is in ES1 but this is not stated.

As suggested, "Significantly" was deleted from the main text and BES1 is now included.

d) How big are these ES promoter region fragments? Fragment 1 is actually over, and immediately downstream of, the start site whereas (depending on the scale) fragment 2 would be in the promoter itself. Where are the results for ESPM2, 3, 5, 6 and 7?

Again, all this is included in a previous publication of our group (Lopez-Farfan et al 2014, Plos Pathog) and now in the new manuscript is described in detail. The reason why we used only ESPM4 and 1 (core promoter region, described now in the scheme of the new Figure 3D) is explained in detail the new manuscript, page 12, to read:

However, we wish to investigate in detail a possible SNF2PH occupancy at the promoter adjacent area, nevertheless highly homologous sequences shared among most of the VSG-ESs (BES1 to BES17 in (Hertz-Fowler et al., 2008)) prevent this analysis by a simple ChIP-seq alignment. We have previously

reported polymorphism in the sequence at particular regions located at the core promoter and upstream the promoter region, referred as ES promoter PCR fragments 1 and 4 (ESPM1 and ESPM4) (schematically represented in Fig 3D) (Lopez-Farfan et al., 2014). These minor polymorphisms in the sequences allowed us to differentiate among different BES promoter regions. In particular, PCR fragment ESPM 4 and 1 yielded 14 different sequences when genomic DNA was used as template (Figure S5 in (Lopez-Farfan et al., 2014)) providing considerable covering of most of the BESs (Hertz-Fowler et al., 2008). ChIP-seq data was generated from the immunoprecipitated chromatin with the anti-SNF2PH antiserum which was then PCR amplified with ESPM1 and 4 PCR primers, and the products were deepsequenced (see Material and Methods). Reads were aligned on BES promoter sequences index file built by combining in a single lane the sequences from the ESPM1 and 4 together with the sequences from the corresponding BESs (BES1, 2, 3, 4, 7, 7dw, 10, 10dw, 12, 13, 15, 15dw, 17, 17dw described previously (Hertz-Fowler et al., 2008)). Next, using Bowtie software, alignment of the reads were assigned to the BES promoter index file, and the number of reads aligning to each BES was represented in Figure 3D. This data showed SNF2PH is enriched to the active BES1, at the ESPM1, which correspond to the core promoter of the active BES1 (VSG-ES221). SNF2PH was detected to a lesser extent in other BES promoters suggesting that SNF2PH is controlling inactive promoters as well (Fig 3D). Interestingly, the increase of read count was at the ESPM1 fragment region where the actual ES promoter is located, suggesting SNF2PH is associated with the active core ES promoter rather than the upstream promoter region (Dataset EV1). Together these data indicate that SNF2PH is located in several BES promoters, however it is most enriched at the active VSG-ES promoter (ESPM1 of the BES1) (Fig 3E).

e) Please also give the CHIP-Seq results for the other promoters analysed in (B). Where is SSR 3U and where are the results for it? Is it a strand-switch region?

As requested, ChIP-seq data for the promoters analysed in Fig 3B are supplied in Dataset EV1. To clarify the terminology, we define SSR (used as a RNA pol II control loci) as a Switch Strand Region located in the Chromosome 7. Results for SSR3 are now described properly, illustrated in Figure B and reads counts included in Dataset EV2.

f) The statement "A minor enrichment was also detected for the splice leader (SL) promoter and coding region." Is actively misleading. The enrichment is the same as at the EP and rRNA promoters and the enrichment on the ES promoter is only 2-fold higher. And in Fig 4B there is no difference at all, although...

New SNF2PH ChIP-seq experiments on the entire genome included in the new manuscript clarify these questions. As suggested, we corrected and precisely described SNF2PH enrichment, considering its implication in a given locus.

Page 11, to read:

In eukaryotes, chromatin remodelers are detected at RNA pol II promoters and play important roles in their activity [26,27]. We investigated the presence of SNF2PH in chromatin across the genome, aside from the multiallelic VSG-ESs, to identify additional genes targeted by this protein. We compared quantitative enrichment profiles with ChIP-seq peak distribution, and considered 0-mismatch error to avoid bias in polymorphic sequences within repetitive chromosomal loci, leading to defined peaks (q value<0.05, dataset EV1)(Fig 3C). As demonstrated by quantitative ChIP, the site of enrichment corresponded to developmentally regulated loci EP and GPEET2 procyclin, and 18S ribosomal DNA and SL RNA-related sequences. Interestingly, SNF2PH localizes at H3.V, a histone variant recently associated with VSG monoallelic expression [28] and its own coding sequence.

g) I do not understand Fig 4B. Does it meant hat the enrichment was the same on all of the tested promoters? This is very poorly explained.

In the new manuscript this figure is much better explained as the previous one Figure 3 including the promoter ChIP-seq mapping, in page 13, to read:

In order to determine whether K2A mutation also affects to SNF2PH occupancy at the promoter region, we constructed a ChIP-seq index library including all the BES sequences from PCR regions ESPM1 and 4 (Fig 3D) and map the reads PCR amplified from DNA immunoprecipitated using antiHA antibodies and chromatin from cell lines expressing HA-SNF2PH and HA-SNF2PH K2A (Fig 4B). As controls we used gene Control 1, a promoter in chromosome 7, SSR7 (Strand Switch

Region, SSR) (Cordon *et al.*, sent to publication) a RNA pol III (U2) were included after input normalization (Fig 4B and Dataset EV2). Sequence alignments using *Bowtie1* and 0 mismatch error yielded a number of reads aligned in BES ESPM1/4 libraries at the active promoter in BES1 and BES2 (highly sequence homologue) that were reduced for the cell line expressing the mutant K2A, to approximately ~0.5-fold. Conversely, at inactive promoters of BES the number of reads was increased (BES12) or no significantly changed (BES7 and BES13). This result is consistent with mutation K2A reduced protein SUMOylation, which decreases the occupancy of SNF2PH in the active promoter chromatin (Fig 4B).

h) In the RPA1 CHIP, Fig 5B, is RLuc difference significant or not? If not it too should be labelled "NS", like rDNA and 18S.

As recommended, in the new graph in Fi 5B, now is labeled: NS, since TbRPA1 enrichment in the RLuc locus is not significant.

For the HA-tagged proteins - it is essential to confirm that the mutant is not SUMOylated - this should be possible by co-IP.

To investigate K2A mutation influences SUMOylation, we performed a co-IP assay under denaturing conditions for the HA-SNF2PH and HA-SNF2PH K2A. Nuclear extracts were immunoprecipitated using anti-HA polyclonal Ab and analyzed by WB using anti-TbSUMO (Upper panel in A) and anti-HA (lower panel in A) as a control for the IP. SUMOylation levels are distinguishable in both HA tagged proteins, suggesting that K2A mutation reduces SUMOylation. Quantification of SUMOylated levels are expressed in B, driving to ~0.5-fold less or the SNF2PH K2A mutant.

6. SNF2PH-associated proteins

a) This would be MUCH more interesting if you were to included similar results for the non-SUMOylated mutant. This needs to be quantitative to allow a comparison with wild-type. Label-free quantitation is fine.

The experiment proposed by referee is interesting unfortunately, we used proteindenaturing conditions required to stabilize SUMO, and therefore no SNF2PH associated proteins could be identified.

b) The discussion is biased. Please mention that RNA polymerase II subunits were also identified! Finding a splicing factor is also intriguing.

As suggested, we mention now in results and discussion that we found specific RNA polymerase II subunits together with the pre-RNA splicing factor 9 (TbPRP9), which suggests that there is an interaction between SNF2PH and RNA pol II associated factors. The following changes have been applied in results section:

Pages 15-16:

Among the identified proteins we found mRNA splicing factor TbPRP9 and nucleosome assembly protein, AGC kinase 1 (AEK1), a kinase essential for the bloodstream form stage [30] and a T-dependent RNA helicase SUB2, (Tb927.10.540). Importantly, several previously identified VSG transcription factors, including Spt16 included in FACT complex (Facilitates Chromosome Transcription) [31], the proliferative cell nuclear antigen (PCNA) and a subunit from the Class I transcription factor A [32] were also identified (Appendix Table S2). Furthermore, RNA pol I subunit RPA135 and the RNA pol II RPB1 were also found to co-purify with SNF2PH suggesting possibly a transient interaction with RNA polymerases subunits, as previously described for yeast SNF2. Together these data suggest that SNF2PH occupies a central position in VSG transcription regulation and likely interactions with RNA pol II transcription.

c) "We found PH (Pleckstrin)-like domain and Zing finger containing protein ZC3H22 as mediators that allow gene transcription by recognition of lysine methylated histones". What is the evidence for this statement? First, which is the protein with the Pleckstrin-like domain (not on the list)? Second, I looked at the TritypDB page (Incidentally the Gene ID is in tb427 format.) and it appears that ZC3H22 is a cytoplasmic protein which is expressed mainly in procyclic forms; and the zinc-finger domain appears, from the Interpro page, to bind mainly to RNA. Finding 7 peptides from this protein in a bloodstream-form preparation is remarkable given that, from the database, it appears never to have been detected previously in bloodstream forms. Perhaps it is moonlighting in bloodstream forms with another function?
As suggested, the annotations and defined Tb927.7.6810 as a EF-Hands domain containing protein and corrected the previous annotation (Pleckstrin-like). Regarding to the zing finger family member ZC3H22, it preserves a function related to posttranscriptional regulation (repressor) of gene expression by GeneDB source. Although this regulation is unclear (probably related to mRNA stability), we have removed the sentence 'mediators that allow gene transcription by recognition of lysine methylated histones'. In addition, the origin of the protein sample was procyclic form.

d) Check the locations and annotations of the other proteins as well. Won't take long.

As required, locations and annotations for Table S2 have been corrected from the list.

7. RNASeq methods:

a) It is not clear to me how these data were analysed. The authors only say (in the Supplement): "We used R software package" to analyse the data. Which package? there are many different packages available for analysis of RNASeq data, please say which one you chose. If you did not use a custom package, you must do so since normalization is an extremely complex problem. Using RPKM for data analysis is not acceptable since it ignores the impact of gene length on data reliability. For example, you had 10 million reads, 1 RPKM on a 10 KB ORF is 100 reads, which gives reliable results, but 1 RPKM on a 300nt ORF is 3 reads, which is not usable.

To clarify how the RNAseq data was analyzed, we included in Methods a detailed procedure regarding to this part. According to this, we use the miARma-Seq pipeline to analyze all transcriptomic data. In detail, this pipeline contains all needed software to automatically perform any kind of differential expression analysis. It uses fastqc to check the quality of the reads and aligned them using Bowtie1. Subsequently, the aligned reads are quantified and summarized for each gene using featurecounts. Finally gene counts are analyzed using the edgeR package from Bioconductor. In such a way, all samples were size corrected in order to be comparable and then normalized using the TMM method from the EdgeR package. TMM values for each gene were used for the differentially expression analysis. We kindly apology for the misunderstanding given by a lack of explanation regarding to the use of RPKM for data analysis. We would like to clarify that both gene length and the library size are taken into account by using EdgeR package. Thus, the effect of library size is corrected among samples to be comparable among them and genes RPKM values are obtained by using the *rpkm* function (EdgeR) from the TMM normalized data. For a better understanding of the

procedure, we briefly mentioned in methods section the analysis regarding RNAseq data.

b) Please present comparisons of reads using a standard package such as EdgeR or DESeq2, supply the full results in a supplementary table; show the principal component analysis; and describe the results more carefully, including a comparison with the available recent RNA Seq datasets for stumpy forms, not older microarrays.

As required, full comparisons of reads using EdgeR for both experimental conditions are provided in Table EV2. To facilitate and improve interpretation, we slightly modified the results section by including a comparison with recent RNA datasets for stumpy forms, as suggested.

c) The sentence "We used R software package to detect genes that were differentially expressed in uninduced versus induced SNF2PH RNAi cells after subtraction of the parental (DRALI) cell line." is incomprehensible. (How can you subtract a cell line?)

This phrase in the previous manuscript led to confusion, thus in the new manuscript Material and methods RNA-seq section this question is better explain, to read:

Genes transcripts isolated form uninduced versus induced SNF2PH RNAi cells with a $[\log_2FC] \geq 1$

(log2

of Fold Change) and $FDR \leq 0.05$) were considered as differentially expressed. Additionally,

mi(Andres-

Leon et al., 2016)ARma-Seq generated a volcano plot to facilitate the identification of genes that felt higher variation in expression.

However, FDR adjustment is considered to be the best approach (Jafari and Ansari-Pour, 2019). The take home message is that it does not matter whether you are interested in identifying a significant association with SNPs, differentially expressed genes (DEG) or enriched gene ontology (GO) terms, the moment you conduct multiple tests on the same samples or gene sets respectively, it would be essential to address the multiple testing issue by adjusting the overall false positive rate through calculating a Δ L or adjusting your raw P values (as shown here based on Bonferroni or FDR) for true positives to be teased out. This will in no doubt enhance reliability and reproducibility of research findings (Jafari and Ansari-Pour, 2019).

8. RNASeq results:

a) When you describe the RNASeq results you say "Eighteen representative..genes were found". Why "representative"? Which other genes were found? Or were only 18 found altogether? If it was only 18, and the FDR was <0.05, then there is nothing significant at all. You are comparing about 9000 genes, so with an FDR of 0.05 you expect at least this many false positives.

To solve referee's doubts about the applied criteria in order to discriminate representative genes, we want to clarify that FDR value takes into account the multiple testing errors accumulated through the 9000 comparisons. Thus, by using an adjusted P-value (such as FDR) we consider the number of false positives among the overall of regulated genes, as a distinct feature that p-value does not provide. In our case, we compare exactly 8673 genes from two biological replicates, that is translated into 221 transcripts if we consider a $p < 0.05$ and 18 transcripts if $FDR < 0.05$. In summary, the most preferable approach is controlling FDR as it not only reduces false positives, but also minimizes false negatives (Jafari and Ansari-Pour, 2019).

b) FDR <0.03 is not appropriate for RNASeq data, please delete this table and discussion.

We agree that $FDR < 0.03$ are not appropriate for RNAseq data and omitted data for Table S3B. However, we consider appropriate $p < 0.05$ since using this p value well known developmental markers are included as DEGs.

c) Changes of less than 2-fold (or, at a stretch, 1.5-fold) are also usually judged not to be biologically significant.

We agree, lees than 2-fold is not biologica significant. Therefore, we have used a 3-fold different cut of (Log2FC). Indeed, changes of logFC less than 1.5-fold are not usually

considered biologically significant. In our analysis, we evaluate data according to $\text{Log}_2\text{FC} \geq 1$ ($\text{FDR} < 0.05$) instead of Fold Change (FC) value. By applying the logarithmic correction, we obtain a lower FC value (e.g. Tb427.3.2140 $\text{logFC} -1.73$ results in 3 FC). In addition, stretch the fold to 1.5 (approx. 1.25 Log_2FC) would suppose not to contemplate interesting DEGs. According to the suggestion of referee #2, we show a shorter table in results excluding genes with $\text{Log}_2\text{FC} \leq 1$. Analysis of resulting 14 genes lead us to similar conclusions. Described in Expanded View EV2

Table S3. Genes expressed after SNF2PH depletion.

GeneName	Length	logFC	logCPM	F	FDR	Description
Tb427.03.2140	2847	-1.73	6.18	221.44	1.54E-45	Transcription activator
Tb427.07.2820	405	0.55	7.78	33.25	3.64E-05	histone H2A
Tb427.06.510	345	1.08	4.77	25.39	1.38E-03	GPEET2 procyclin precursor
Tb427_09_v4.snoRNA.0026	96	1.52	1.77	22.69	4.20E-03	C/D snoRNA
Tb427.01.4710	537	1.16	4.24	20.80	8.90E-03	hypothetical protein
Tb427.BES40.15	2031	0.42	8.50	20.46	8.90E-03	expression site-associated gene 8 (ESAG8) protein
Tb427.10.10910	2115	-0.32	11.87	19.64	1.17E-02	heat shock protein, putative
Tb427.10.10240	1218	1.28	1.66	19.05	1.40E-02	procyclin-associated gene 1 (PAG1) protein
Tb427.10.10260	426	0.95	4.52	18.01	1.94E-02	EP1 procyclin
Tb427.BES40.18	1413	0.37	9.94	17.97	1.94E-02	expression site-associated gene 2 (ESAG2) protein
Tb427.BES40.14	1893	0.39	8.73	17.81	1.94E-02	expression site-associated gene 8 (ESAG8) protein
Tb427.10.10230	1113	1.22	1.73	17.28	2.35E-02	procyclin-associated gene 5 (PAG5) protein
Tb427.10.7160	621	1.20	0.06	17.05	2.46E-02	procyclin-associated gene 1 (PAG1) protein
Tb427.BES64.2	1467	1.13	3.50	16.70	2.67E-02	variant surface glycoprotein (VSG)
Tb427.10.10210	1122	1.17	1.30	16.62	2.67E-02	procyclin-associated gene 4 (PAG4) protein
Tb427.10.10250	390	1.02	4.10	16.33	2.91E-02	EP2 procyclin
Tb427.10.10960	2115	-0.28	12.19	16.18	2.97E-02	heat shock protein, putative
Tb427.10.10970	2115	-0.27	12.56	15.80	3.42E-02	heat shock protein, putative

Table S3. Genes expressed after SNF2PH depletion. Displayed genes were identified from two biological replicates with a $\text{FDR} < 0.05$ and $\text{P-value} < 7.09 \times 10^{-6}$. Representative values are expressed as a log fold change (logFC), log for count per million (logCPM) and sorted by the highest F score.

Genes expressed after SNF2PH depletion ($\text{Log}_2\text{FC} \leq 1$, $\text{FDR} < 0.05$). Again, identifying differentially expressed genes (DEG) is essential to adjust the overall false positive rate through adjusting the raw P values based on FDR for true positives to be teased out (Jafari and Ansari-Pour, 2019).

d) Check the annotations, using the Tb927 numbers. If you do this you will discover that you have even fewer regulated genes than you thought. NO fewer than EIGHT of them are almost identical (HSP83). The various procyclins and PAGs also have some shared sequences. There is also no evidence that Tb427.02.4950 has anything to do with translation.

As requested, we confirmed annotations according to the 927 strain and indeed we found three identical heat shock proteins (Tb927.10.10910, Tb927.10.10960 and Tb927.10.10970). However, if we consider $\text{Log}_2\text{FC} \leq 1$, we teased out most them. Although some PAGs share sequences, the number showed different chromosome

location, especially for PAG1, which means independent genes are being expressed under this phenotype.

9. Differentiation experiments. These are done with trypanosomes that are presumably not capable of making stumpy forms or differentiating to growing procyclic forms, so interpretation is virtually impossible. Why are these experiments in the supplement, but nevertheless highlighted in the Abstract? From the data it seems that:

As referee's suggestion, we included the differentiation experiments as a Figure 6 (Fig 6A and 6B, Fig EV3) in the main text, together with the SNF2PH overexpression experiments in PF (Fig 6D-E, Fig EV3).

a) S5 The knock-down results in an increase in procyclin and PAD1-2 mRNAs. The increase is, however, very modest.

b) S5 There is no effect on MyoB but I am not sure why this gene was picked. Was it increased in procyclic forms in the recent RNASeq experiments?

In order to clarify this point, the MyoB was picked as an invariable control locus, but not used as a housekeeping gene to normalize. This gene showed no changes in the new experiments provide in this manuscript, see procyclic forms RNAseq datasets (Dataset EV4).

c) S5 Panel C: There is a slight increase in surface procyclin expression. There is also a decrease in VSG but that was already shown.

In the new EV figures, we have improved considerable the histograms and data below of previous Fig S5 panel D into in a new Figure EV3 (B).

After knocking down the expression of SBF2PH there is a growth defect, which might explain some or even all of the effects seen. Numerous studies - including decreasing VSG synthesis - have shown slight increases in some stumpy form RNAs after growth retardation. This is not necessarily anything to do with differentiation, it could just be a stress response. Cite the relevant papers, especially Batram et al. In summary, the effects seen here do not show that SNF2PH is a regulator of differentiation.

We agree there are many examples of essential factors associated to VSG expression when actually affect chromatin in general. However, as explained before we rule out this possibility by several means, perhaps the most clear is the new experiment of gain of function where the overexpression of a SNF2PH and mutant lacking of the Plant Homeodomain in procyclic forms did not have any effect in genes other than the BES and the BF housekeeping genes (Fig 6F). Thus, for example upregulation of VSG basic copies was no detected (as occurred with many other previously published VSG regulators), suggesting that SNF2PH specifically regulates differentiation by maintaining the bloodstream form gene expression, in a ES attenuation independent manner.

In the discussion section *Battarm et al.*, is now referred, to read:

Whereas AMPK α 1 activation promotes differentiation, reduced SNF2PH expression in stumpy forms observed in wild-type pleomorphic trypanosomes during mice infection confirmed the biological relevance. This result rule out a possible SIF-independent induction of differentiation by VSG-ES transcription attenuation described previously [44].

10. Finally, the authors indeed work on differentiation-competent trypanosomes. Overall the experiments in Fig S6 do not demonstrate any convincing link between AMPK α 1 and SNF2PH, and have nothing to do with SUMOylation. So I'm not sure why they were included. If you want to include them to get rid of them, merely state in the results section that you could find no convincing evidence that SNF2PH is phosphorylated. Also if you include them the following changes are needed.

In response to the referee's comment about this section, we first have toned down this section of the manuscript, also we decided to provide a new point of view of SNF2PH role in differentiation-competent trypanosomes. Thus, we focus this data in

the experiments that are more reproducible and show a rapid effect; the AMPK α 1 activation using a AMP analogue led to SNF2PH strong reduction in expression as occurs in fully capable of differentiation trypanosomes (pleomorphic strain). This results are based in previously published work that convincingly demonstrated this AMP analogue specifically activates AMPK α 1, both in vivo and in vitro phosphorylation assays (Saldivia et al., 2016)). That previous work, demonstrated that activation of AMPK induced a Stumpy-like phenotype, as we mimic by RNAi depletion of SNF2PH. This is also consistent with the results showed in Figure S5C (attached figure) whereby the activation of AMPK is concomitant with the reduction of SNF2PH expression using protein extract isolated from pleomorphic trypanosomes infected in mice. These data together clearly demonstrate, first in stumpy-like trypanosomes using a monomorphic strain (amply utilized by research groups working in differentiation) and most importantly, later in pleomorphic trypanosomes that these to protein are linked by a unknown mechanism.

Panels A and D should be put together. Otherwise it is impossible to see whether there is an effect of the knock-down or not – please don't force the viewers to jump from one panel to the other in order to see that there probably isn't. (Also fix the labels on panel D).

We thank the referee for this comment, which improved the interpretation of the panel, by putting together panels S6A and D into a new Fig S5B (Upper panel). Additionally, we altered the distribution of the figures according to the main text.

Panel B shows that 5'AMP treatment causes a decrease in SNF2PH expression (presumably because of growth arrest) and an increase in phospho-AMPKa1

We have previously published (Saldivia et al., 2016), these data in detail with all types of controls, including in vitro phosphorylation assays, which showed AMPK phosphorylation is a direct consequence of the AMP analogue treatment, and inhibited by Compound C in a purified kinase assay. We have demonstrated these and rule out that possibility in a previous publication (Saldivia et al., 2016), Activation of TbAMPK α 1 promotes the initial steps of differentiation of *T. brucei* into stumpy-like forms in monomorphic strain. Activation by AMP analogs induces the expression of developmentally regulated genes, indicating that TbAMPK can modulate gene expression in order to modulate metabolism. Importantly, expression of a constitutively (truncated) active form of AMPK, CA-HA-AMPK, also induced stumpy-like gene expression, similar to the one induced by treatment with the AMP analogue. Further, inhibition of TbAMPK function, by either compound C or RNAi, significantly counteracted the effects of AMP on growth inhibition and the changes to expression of stumpy marker genes. Taken together, these results ruled out possible indirect effects of the AMP analog and suggest that AMPK α 1 is a central regulator within the stumpy differentiation process (Saldivia et al., 2016).

Panel C is an immunoprecipitation of SNF2PH after AMP treatment, with and without inhibitor. This is a single experiment and it is impossible to tell which phospho-Tyr

band(s), if any, might be SNF2PH. IT all looks likel background to me. (I could see no difference between the lanes with and without inhibitor.) TThis panel was wholly unconvincing. I got a second opinion from a student, who just "WHAT? How can they claim that? More seriously - if the lower panel is the same blot, how come the markers have moved? If you don't even know where the markers are, how can you possibly even guess which background band might be the one you are looking for?. The claim "The phosphorylated band in upper panel corresponds to SNF2PH (arrow, lane 3)" is not justified, and neither is the conclusion "Cells treated with 5'-AMP contained phosphorylated SNF2PH".

To avoid these type of personal interpretation, we have presented in the new manuscript these type of analysis as in histograms of the measures obtained form at least 3 experiments. Quantitative western blot analysis (using IR fluorescence measures of the signals) and representation by histogram with the error bar (SEM) are much more appropriated form of presenting this results (Appendix, Fig S4). In any case, we kindly disagree with referee's comment, since a clear difference in the between bands intensity (western blot aside; black versus white arrow) using the anti-phosphothreonine antibody showed stronger signal in cell extracts without inhibitors, (black arrow), versus cells treated with the AMPK inhibitor Compound C (white arrow). In the new manuscript, we decided to remove these data out of the scope for this report and focus in the developmental regulation of gene expression data. These data will be published eventually in a specialized kinase signaling journal.

Panels E and F are missing a vital control - compound C alone - so cannot be interpreted.

We agree with referee's #2 opinion comment about the missing control treated with compound C alone. Considering that this figures cannot be interpreted, we suppressed data and discussion for Figs. S6 E and F.

11. Methods:

The rationale for including some methods, but not others, in the main text, escapes me. It would be better to mention all of them in the main text and refer to the Supplement for details. Meanwhile the Supplementary methods are missing quite a lot of details, such that repeating the experiments would not be possible.

As suggested, in the new manuscript all methods are included.

FORMATTING ISSUES AND OTHER POINTS

Fig S4 also has a confusing legend in the inset. "hs vs" is peculiar. Why not just label these with the cell type? Please show the individual curves for the replicates as cumulative cell number. That way the degree of variation will be seen – at the moment this is invisible except for the last time point.

To facilitate visualization of growth curves, in the new Fig S4 we included the individual curves as cumulative cell number and labelled with "cell type" in the legend. Consequently, we also altered the figure legend for this graph.

Some of the legends and statements are not very accurate - with a tendency towards

over-statement.

Examples are:

Fig 1A legend "highly expressed" - relative to what? The actual expression level was not measured. Is the upper band in bloodstream forms the SUMOylated version? Is there SUMOylation in procyclic forms as well?

Fig 1A. SNF2PH is highly expressed is incorrect, therefore we have now modified to SNF2PH is differentially expressed in T. brucei developmental stages.

The upper band in bloodstream seems to be related to the SUMOylated form of SNF2PH, which is absent in procyclic forms, in which there is no evidence for SNF2PH SUMOylation, as judged by IP (data not shown).

Fig 1B: "full" depletion is not an accurate description, the protein has merely dropped below the level that is detectable with the antibody - the detection limit of which was not measured. There's almost certainly still some protein there.

Fig 1B. Knockdown of SNF2PH in bloodstream form leads to protein depletion after 24h.

Fig 2A legend - the control panel is below, not to the right.

Fig 2A legend. Corrected

Fig 4 - "fold enrichment over no antibody control including a nontagged cell line".

Doesn't make sense.

Fig 4 A Data from three independent clones are represented as fold enrichment. A nontagged cell line (Single Marker) was included as a negative HA control.

Page 7: "which is probably due to the highly dynamic nature of protein SUMOylation." This should be stated as a hypothesis, not (at this stage) an explanation. "precise recognition determinants for SNF2PD are unlikely to resemble metazoan..." Well, no, especially as the domain resembles one from plants. How about targets from plants?

Would it be possible to eliminate some of the abbreviations? They are rather numerous and make the manuscript very difficult to read. For example "HSF" is very commonly used for "heat-shock factor", "PTM" isn't used that often. "IPed" is horrible...

“IPed” abbreviations eliminated. However, abbreviations like, PTM and HSF are required.

References should be given for the identities of the different proteins involved in (de)-SUMOylation, in cases where they were previously studied. For the remainder, the evidence that this is the right protein should be stated briefly.

"mature or pre-spliced rDNA+ 780" DO you mean rRNA? Whether or not it is rRNA, what is the +780?

rDNA+780 refers to the pre-spliced RNAs from ribosomal DNA. “+780” marks the nucleotide position upstream the rDNA promoter (rDNA pro).

Oligonucleotide Tables - Give TritypDB numbers please.

TritypDB numbers for oligonucleotide pair are detailed in Appendix Table S4

TYPOS and OTHER ODDITIES

There are quite a lot of textual errors, especially in the Supplement (Did you think the reviewers wouldn't read it?)

Here are some examples but the whole text needs scrutiny.

Appendix information has been updated and text and edit corrections have been applied to the new manuscript.

p4: to positive regulate" (->"positively" or "to enhance")

P4. Think is better positive regulation that enhance, as proposed.

p5 "unlikely to resemble metazoan." Dangling adjective. Should be "those from metazoa" or "metazoan ones".

P5. Unlikely to resemble metazoans changed to likely to resemble plants.

"included too in transcriptional activators as HsSMCA5 and MsSMCA1". I think "as" should be "like"?

"...transcriptional activators like HsSMCA5 and MsSMCA1".

"The SNF2 N domain is conserved across the kinetoplastida, importantly, this factor...

Sentence changes subject halfway through. Which factor?

"Importantly, Tb927.3.2140 contains a distinct homeodomain..."

"conserved with Histone Methyltransferases" - re-write.

New manuscript sentence changed page 7 to read:

The PHD is a conserved homeodomain involved in development [19] that binds H3 tails and reads unmodified H3 tails [20] as well as H3 trimethylated at Lys4 (H3K4me3) [21] or acetylated at Lys8 and

Lys14 [22,23]. The PH domain is conserved in histone methyltransferases, including murine HMT3 and

human NSD3 (Appendix, Fig S1). Thus, we named this chromatin-remodeling factor trypanosome SNF2PH.

"not a target of SUMOylation detected as a single protein band" doesn't make sense.

By the heterologous SUMOylation system in bacteria, expression of the C-terminal domain of SNF2PH (Fig 2) showed no SUMOylated upper band, thus, this fragment is not a target for SUMO.

.....To determine which domains of SNF2PH are SUMOylated, we used an *E. coli* strain expressing the complete *T. brucei* SUMOylation system [13]. We evaluated two different constructs encompassing

the SNF2PH N-terminal or C-terminal domain (SNF2PH-N and SNF2PH-C, respectively), bearing a Flag

tag. We co-expressed SNF2PH-N and SNF2PH-C in *E. coli* with TbSUMO (already exposing the diGly

motif) and both activating enzyme subunits (TbE1a/TbE1b) plus the conjugating enzyme (TbE2).

SNF2PHN appears as a single band migrating at the expected position when expressed alone in *E. coli*

(Fig 2C, lane 1) or when co-expressed with a partially reconstituted system (lane 2 and 3). However, when co-expressed with the complete SUMOylation system additional slower-migrating bands can be

detected (lane 4) suggesting that the N-terminal domain of SNF2PH can be SUMO conjugated. In contrast, the C-terminal domain is not a target of SUMOylation since it is only ever detected as a single

protein band at the expected position of unmodified protein (Fig 2D).

"Genomic was extracted by the phenol-chloroform method, previously treated with RNase" ! Also, in the same paragraph, "shared chromatin"

Crosslinks were reversed at 65.C for 15h. After RNase and Proteinase K treatment, DNA was extracted with phenol: chloroform and ethanol precipitated, (Materials and Methods)

I was also puzzled by "Percentage of immunoprecipitated product was referred to 10% of input." Surely it should be referred to 100% of input, otherwise it's misleading?

To compare the amount of DNA immunoprecipitated to the total input DNA, 10% of the pre-cleared chromatin saved as input was processed with the eluted immunoprecipitated products before the crosslink reversal step. CHIP experiments are well known to be no very efficient, as expected after such a long and complicated

procedure.

"there are at least two major sites for SUMOylation in.SNF2PH N-terminal." Nterminal what ? (Should be "the Nterminus").

In the new version of the manuscript, N-terminus is been incorporated

"downstream of the active VSG-ES to the telomeric VSG221 gene" ? Sentence also has mixed tenses.

"Promoter " was added.

downstream of the active VSG-ES promoter to the telomeric VSG221 gene

"while in silent promoters is detected"

Precisely, in silent promoters is detected to a lesser extent.

"showed both enrichment with the highest number of reads" - the "both" is hanging there without anything extra.

"Both" is removed

"suggested involvement of this transcription factor in.." You haven't actually shown it is a transcription factor.

The word "transcription" is omitted in this case. Notwithstanding, we provide a substantial amount of data supporting SNF2PH regulates gene expression. SNF2PH includes homology to a conserved domain such as, the SNF2_N, characteristic of global transcription activator SNF2L1 and SMARCA1 (Fig S1), well known transcription factors, and the PH domain that binds to histones tails to regulate transcription in developmentally regulated NURF and CHD4 (Wysocka et al., 2006). Most importantly, we showed that SNF2PH actually binds to VSG-ES promoter chromatin and depletion leads to reduction of both the VSG mRNA and RNA polymerase recruitment to the VSG-ES promoter and even VSG protein levels (Fig 5). Further, SNF2PH overexpression in the insect form, triggers stage-specific bloodstream form gene expression (Fig 6 & EV & Fig S4). In sum, our data strongly suggest that SNF2PH functions in developmental regulation of transcription. In our opinion, it seems extremely unlikely that a DNA repair or DNA replication factor might lead to the series of phenotype we described here in this work

Page 12 "Consecutively," - I think you mean "Consequently".

Indeed, we mean "Consequently" instead of consecutively.

References

- Andres-Leon, E., Nunez-Torres, R., and Rojas, A.M. (2016). miARma-Seq: a comprehensive tool for miRNA, mRNA and circRNA analysis. *Sci Rep* 6, 25749.
- Clapier, C.R., Iwasa, J., Cairns, B.R., and Peterson, C.L. (2017). Mechanisms of action and regulation of ATP-dependent chromatin-remodelling complexes. *Nature Reviews Molecular Cell Biology* 18, 407.
- Eustermann, S., Schall, K., Kostrewa, D., Lakomek, K., Strauss, M., Moldt, M., and Hopfner, K.P. (2018). Structural basis for ATP-dependent chromatin remodelling by the INO80 complex. *Nature* 556, 386-390.
- Farnung, L., Vos, S.M., Wigge, C., and Cramer, P. (2017). Nucleosome-Chd1 structure and implications for chromatin remodelling. *Nature* 550, 539-542.
- Giles, K.A., Gould, C.M., Du, Q., Skvortsova, K., Song, J.Z., Maddugoda, M.P., Achinger-Kawecka, J., Stirzaker, C., Clark, S.J., and Taberlay, P.C. (2019). Integrated epigenomic analysis stratifies chromatin remodellers into distinct functional groups. *Epigenetics & Chromatin* 12, 12.
- Glover, L., Hutchinson, S., Alsford, S., McCulloch, R., Field, M.C., and Horn, D. (2013). Antigenic variation in African trypanosomes: the importance of chromosomal and nuclear context in VSG expression control. *Cellular microbiology* 15, 1984-1993.
- Hertz-Fowler, C., Figueiredo, L.M., Quail, M.A., Becker, M., Jackson, A., Bason, N., Brooks, K., Churcher, C., Fahkro, S., Goodhead, I., et al. (2008). Telomeric Expression Sites Are Highly Conserved in *Trypanosoma brucei*. *PLOS ONE* 3, e3527.

- Jafari, M., and Ansari-Pour, N. (2019). Why, When and How to Adjust Your P Values? *Cell journal* *20*, 604-607.
- Li, M., Xia, X., Tian, Y., Jia, Q., Liu, X., Lu, Y., Li, M., Li, X., and Chen, Z. (2019). Mechanism of DNA translocation underlying chromatin remodelling by Snf2. *Nature* *567*, 409-413.
- Lopez-Farfan, D., Bart, J.M., Rojas-Barros, D.I., and Navarro, M. (2014). SUMOylation by the E3 ligase TbSIZ1/PIAS1 positively regulates VSG expression in *Trypanosoma brucei*. *PLoS Pathog* *10*, e1004545.
- Mouriz, A., López-González, L., Jarillo, J.A., and Pineiro, M. (2015). PHDs govern plant development. *Plant Signaling & Behavior* *10*, e993253.
- Muller, L.S.M., Cosentino, R.O., Forstner, K.U., Guizetti, J., Wedel, C., Kaplan, N., Janzen, C.J., Arampatzi, P., Vogel, J., Steinbiss, S., *et al.* (2018). Genome organization and DNA accessibility control antigenic variation in trypanosomes. *Nature* *563*, 121-125.
- Musselman, C.A., and Kutateladze, T.G. (2011). Handpicking epigenetic marks with PHD fingers. *Nucleic acids research* *39*, 9061-9071.
- Navarro, M., and Gull, K. (2001). A pol I transcriptional body associated with VSG mono-allelic expression in *Trypanosoma brucei*. *Nature* *414*, 759-763.
- Raab, J.R., Chiu, J., Zhu, J., Katzman, S., Kurukuti, S., Wade, P.A., Haussler, D., and Kamakaka, R.T. (2012). Human tRNA genes function as chromatin insulators. *The EMBO journal* *31*, 330-350.
- Saha, A., Wittmeyer, J., and Cairns, B.R. (2002). Chromatin remodeling by RSC involves ATPdependent DNA translocation. *Genes Dev* *16*, 2120-2134.
- Saldivia, M., Ceballos-Perez, G., Bart, J.M., and Navarro, M. (2016). The AMPKalpha1 Pathway Positively Regulates the Developmental Transition from Proliferation to Quiescence in *Trypanosoma brucei*. *Cell Rep* *17*, 660-670.
- Sanchez, R., and Zhou, M.-M. (2011). The PHD Finger: A Versatile Epigenome Reader. *Trends in biochemical sciences* *36*, 364-372.
- Shi, X., Hong, T., Walter, K.L., Ewalt, M., Michishita, E., Hung, T., Carney, D., Pena, P., Lan, F., Kaadige, M.R., *et al.* (2006). ING2 PHD domain links histone H3 lysine 4 methylation to active gene repression. *Nature* *442*, 96-99.
- Van Bortle, K., Nichols, M.H., Li, L., Ong, C.-T., Takenaka, N., Qin, Z.S., and Corces, V.G. (2014). Insulator function and topological domain border strength scale with architectural protein occupancy. *Genome Biology* *15*, R82.
- Watson, A.A., Mahajan, P., Mertens, H.D., Deery, M.J., Zhang, W., Pham, P., Du, X., Bartke, T., Zhang, W., Edlich, C., *et al.* (2012). The PHD and chromo domains regulate the ATPase activity of the human chromatin remodeler CHD4. *Journal of molecular biology* *422*, 3-17.
- Whitehouse, I., Stockdale, C., Flaus, A., Szczelkun, M.D., and Owen-Hughes, T. (2003). Evidence for DNA translocation by the ISWI chromatin-remodeling enzyme. *Mol Cell Biol* *23*, 1935-1945.
- Wysocka, J., Swigut, T., Xiao, H., Milne, T.A., Kwon, S.Y., Landry, J., Kauer, M., Tackett, A.J., Chait, B.T., Badenhorst, P., *et al.* (2006). A PHD finger of NURF couples histone H3 lysine 4 trimethylation with chromatin remodelling. *Nature* *442*, 86.
- Yan, L., Wu, H., Li, X., Gao, N., and Chen, Z. (2019). Structures of the ISWI-nucleosome complex reveal a conserved mechanism of chromatin remodeling. *Nat Struct Mol Biol* *26*, 258-266.

2nd Editorial Decision

21 August 2019

Thank you for the submission of your revised manuscript to our editorial offices. We have now received the reports from the two referees that were asked to re-evaluate your study, you will find below. As you will see, the referees now support the publication of your manuscript in EMBO reports. Referee #2 has some remaining concerns and further suggestions we ask you to address in a final revised version of your manuscript. Please address these points also in a detailed p-b-p response.

- Please revise the title of the manuscript as it is currently too long (100 characters including spaces). Please avoid abbreviations (SNF2PH is fine, but many of our readers would not know what VSG stands for).
- Please shorten the abstract. It should have not more than 175 words.
- It seems Diana Lopez-Farfan is listed as DLP in the author contributions. Please check.
- Fig. 6C is presently called out before 6A. Please call out the figures and panels sequentially. Thus, please re-arrange the text, or change the order of the panels in the figure.
- Please deposit primary datasets produced in this study (e.g. mass spec., RNA-seq. and CHIP-seq. data) are deposited in an appropriate public database. See:
<http://embor.embopress.org/authorguide#datadeposition>

The accession numbers and database should be listed in a formal "Data Availability " section (placed after Materials & Methods) that follows the model below. Please note that the Data Availability Section is restricted to new primary data that are part of this study.

Data availability

- Please indicate in field D10 in the author checklist compliance with the ARRIVE guidelines. See: <http://www.embopress.org/page/journal/14693178/authorguide#livingorganisms>
- Please have the manuscript proofread by a native speaker (see also points of referee #2).
- Please find attached a word file of the manuscript text (provided by our publisher) with changes we ask you to include in your final manuscript text, and some queries, we ask you to address. Please provide your final manuscript file with track changes, in order that we can see the modifications done.

In addition I would need from you:

- a short, two-sentence summary of the manuscript
- two to three bullet points highlighting the key findings of your study
- a schematic summary figure (in jpeg or tiff format with the exact width of 550 pixels and a height of not more than 400 pixels) that can be used as a visual synopsis on our website.

REFeree REPORTS

Referee #1:

The authors have provided important new data and have more clearly linked SNF2PH function in VSG expression and in wider gene expression and developmental control. I am happy to recommend this revision for publication and consider it to add a new and interesting angle to our understanding

of gene expression control in *T. brucei*, where numerous studies have sought to understand the processes that influence genome-wide multigene transcription.

Referee #2:

This paper is much improved but I still question whether the transcription factor is regulating differentiation. All effects of its ectopic expression in procyclic forms concern expression site genes (with one interesting exception, see below). In my opinion the new dataset suggests that there is specific regulation of expression sites, and the effects that resemble differentiation in bloodstream forms are secondary.

Othwise there are a few minor things that still need dealing with.

Details:

The new RNASeq data (EV4) showing the effect of SNF2PH expression are really interesting but not for the reason given by the authors. While the authors claim that this shows regulation of differentiation, so far as I can see nearly all of the induced genes (FDR<0.05 and a generous threshold of regulation of at least 1.5x) are from the expression site. The only exceptions are the genes encoding 65kDa invariant surface glycoprotein. (Although Tb427.02.3290 and 3300 are shown as "unspecified" they are actually also ISG65.) It would therefore be better to say that induced expression turns on the expression site. Are there any ISG65 sequences in expression sites? If not the specific effect is really interesting.

The text says that SNF2PH expression in procyclic forms causes significant "underrepresentation" of EP and GPEET procyclin and the glucose transporter 10.8510. ("All these markers were underrepresented in SNF2PH (FDR>0.05),") But it did not: differences are marginal and and the FDR values are way higher than 0.05. Since not everyone checks the supplement, this is seriously misleading. Even if I have just misunderstood the text, it needs fixing so it is clearer.

With the PH mutant there are only 8 genes with a significant change over 2x.. The authors should say this. Even after allowing for multiple testing, this is questionable. The histone H4 is interesting but there are several copies of this gene and the others aren't changed. The authors say "insect stage-specific genes such as EP2-3 procyclin and GPEET-2 procyclin precursor were significantly expressed" - but it's not clear what this means. The fold increase is much less than 1.5.

Minor points:

If you do not want to preset the remaining SUMO-protein data, simply state that it will be presented elsewhere.

Comment regarding tRNAs - the authors should mention that these regions are implicated in termination and are enriched in the relevant histon variants (Genes Dev. 23, 1063-1076, MBP172, 141-144)

The manuscript still needs some attention, partly for accuracy and aprtly for the English. At least one Figure number citation seems to be wrong, and I didn't check them all. This can easily happen when a paper is modified.

Dataset EV4 - the description on the third sheet is incorrect. It should say that this is the complete gene list. The fact that the authors dtake FDR<0.05 to be significant can be stated separately.

Abstract:

"Further, overexpression of SNF2PH in the insect form induces expression of BF-specific genes, while a deletant lacking the PH domain did not." - tense change in the middle of the sentence

"clusters of tRNA genes, which functions as insulators" -> function

"Specifically, it was identified a downregulation of single telomeric sequence" - needs correcting.

"In addition to induction of differentiation to the procyclic form after SNF2PH depletion in the BF

(Fig 3B and Fig EV3), - Fig 3B is CHIP. Should this be Fig 6?

"Bloodstream pleomorphic trypanosomes undergo accurately differentiation" Remove "accurately".
(Also ungrammatical)

"SNF2PH is negatively regulated during stumpy transition"

"SNF2PH is SUMO-modified to function as a transcriptional activator of VSG-ES monoallelic expression, while the PH-domain promotes bloodstream stage-specific gene expression and is required for the maintenance of the mammalian infective form" Change to:

"SNF2PH is SUMO-modified to function as a transcriptional activator of VSG-ES monoallelic expression, and the PH-domain is required for both this and for the maintenance of the mammalian infective form".

2nd Revision - authors' response

22 September 2019

Referee #1:

The authors have provided important new data and have more clearly linked SNF2PH function in VSG expression and in wider gene expression and developmental control. I am happy to recommend this revision for publication and consider it to add a new and interesting angle to our understanding of gene expression control in *T. brucei*, where numerous studies have sought to understand the processes that influence genome-wide multigene transcription. We are thankful that this reviewer thought that this work adds a new and interesting angle to our understanding of gene expression control in *T. brucei*.

Referee #2:

This paper is much improved but I still question whether the transcription factor is regulating differentiation. All effects of its ectopic expression in procyclic forms concern expression site genes (with one interesting exception, see below). In my opinion the new dataset suggests that there is specific regulation of expression sites, and the effects that resemble differentiation in bloodstream forms are secondary.

The main concern of this reviewer is whether the transcription factor SNF2PH is regulating differentiation altogether. We agree with this comment and so in the new version of the manuscript we have now changed the title (as recommended initially) and removed references to the function of SNF2PH in differentiation. Instead, we provide additional results that strongly support that SNF2PH functions as a transcription factor that regulates the expression of bloodstream form stage-specific surface protein genes. Thus, we focus the manuscript in a more precise function for SNF2PH regulating genes that showed significant increased expression in procyclic form, including now the invariant surface glycoproteins (ISG65), as suggested by the referee. Thus, we have focused the new manuscript by including a proper description of this in the Results and Discussion sections of the new manuscript and delete all conclusions regarding a possible broader function of SNF2PH in differentiation, which most likely requires future additional work to be completely demonstrated.

Othwise there are a few minor things that still need dealing with.

Details:

The new RNASeq data (EV4) showing the effect of SNF2PH expression are really interesting but not for the reason given by the authors. While the authors claim that this shows regulation of differentiation, so far as I can see nearly all of the induced genes (FDR<0.05 and a generous threshold of regulation of at least 1.5x) are from the expression site. The only exceptions are

the genes encoding 65kDa invariant surface glycoprotein. (Although Tb427.02.3290 and 3300 are shown as "unspecified" they are actually also ISG65.) It would therefore be better to say that induced expression turns on the expression site. Are there any ISG65 sequences in expression sites? If not the specific effect is really interesting.

Thank you for pointing this out. We have now included ISGs results implications in the new manuscript. Indeed, ISG genes are not included in the telomeric VSG-ES; rather, these genes are located in a tandem array in a chromosomal internal position. In addition, these genes are developmentally regulated as bloodstream stage-specific surface proteins (exclusively expressed in bloodstream stage. Indeed, how ISG65 induced expression upon ectopic SNF2PH overexpression is really interesting since almost the entire family was upregulated. In addition, ISGs are required for normal bloodstream form proliferation (3) and ISG genes are located in chromosomal internal positions (4) and not associated with telomeric BES expression. Altogether these results showed SNF2PH functions as a global regulator of bloodstream gene expression. We have now included a ISG65kDa increased expression upon ectopic SNF2PH overexpression in the insect stage in page 18 of the new manuscript.

We have also extensively discussed the implication of this data as a demonstration of SNF2PH functioning as a global regulator of bloodstream stage-specific surface protein gene expression. Interestingly, VSGs and ISGs gene families are transcribed by RNA polymerases type I and II, suggesting SNF2PH regulation of transcription is epigenetic.

In page 24 of Discussion we include:

In Interestingly, the ISG65 genes are transcribed by RNA pol II from polycistronic arrays located in core chromosomal regions, while VSG is transcribed by RNA pol I from the VSG-ES (BES) telomeric locus. Notwithstanding, the chromatin remodeler SNF2PH, utilizing the PH domain as an epigenetic reader, recognizes both distinct gene families. This is suggested by SNF2PH ectopic expression in procyclic which led to a developmental epigenetic reprogramming, similar to homeodomain proteins in other organisms. Furthermore, ISGs are transcribed at similar levels for all allelic variants while the VSG is monoallelically-transcribed one out.....

The text says that SNF2PH expression in procyclic forms causes significant "underrepresentation" of EP and GPEET procyclin and the glucose transporter 10.8510. ("All these markers were underrepresented in SNF2PH (FDR>0.05),") But it did not: differences are marginal and and the FDR values are way higher than 0.05. Since not everyone checks the supplement, this is seriously misleading. Even if I have just misunderstood the text, it needs fixing so it is clearer.

We agree that this phrase is confusing, so we have now deleted in the new manuscript.

With the PH mutant there are only 8 genes with a significant change over 2x. The authors should say this. Even after allowing for multiple testing, this is questionable. The histone H4 is interesting but there are several copies of this gene and the others aren't changed. The authors say "insect stagespecific genes such as EP2-3 procyclin and GPEET-2 procyclin precursor were significantly expressed" - but it's not clear what this means. The fold increase is much less than 1.5.

Again, this commentary was misinterpreted; this is the question, that very few genes were changed. Expression of the SNF2deltaPH mutant serves as a negative control of SNF2PH full-length overexpression, and thus, showed few genes changing expression. Importantly, none of those were related to the VSG-ESs or to the ISGs. In fact, this result supports the whole experiment. The fact that PH delta mutant induced expression of procyclic genes (EP2 and GPEET) genes that are normally expressed in this form, showed the mutant induced no real changes in gene expression, to do so overexpression of the full-length SNF2PH is required.

Thus, the phrase it has been deleted, since led to misunderstanding, while the data clearly showed ectopic expression of the PH mutant does not alter the expression compared to parental cell line used for ectopic expression. This result serves as control, and more importantly demonstrates that PH domain is required to direct the transcriptional activation function of SNF2PH to a specific set of genes, the Bfsurface proteins. Thus, we have now explain these results in a different manner and thus, since is it was misleading, in the new manuscript we exchanged that paragraph for this new one:

.....The induction of BES-related genes detected in cell lines overexpressing full-length SNF2PH did not occur in cell lines overexpressing the SNF2 Δ PH mutant lacking the PH domain (Fig 6F). This data suggests that the PH domain is required to provide specificity to SNF2PH recognizing chromatin targets, as RNA-seq analysis showed the mutant did not upregulated telomeric VSG-ES transcripts (Fig 6F and Dataset EV4).

Minor points:

Comment regarding tRNAs - the authors should mention that these regions are implicated in termination and are enriched in the relevant histon variants (Genes Dev. 23, 1063-1076, MBP172, 141-144)

We do not agree. This is, in our opinion, incorrect. The Siegel et al., paper reported different positions of modified histones and variants to the position we describe here for SNF2PH. SNF2PH is located at the actual tRNA gene clusters. Histones are not mapping to tRNA genes, rather *downstream from or upstream of* histone peaks, as clearly mentioned in page 1064 of this article:

Text in page 1064 of N. Siegel et al. 2009, Genes & Development:

....Many of these peaks occur immediately downstream from tRNA genes (see example in Fig. 2B). Most tRNA genes are located at convergent SSRs but some are present at non-SSRs that appeared to be embedded within pol II PTUs, which raised questions about the overlap of pol III and pol II transcription. We found that all but three tRNA genes that are not associated with convergent SSRs are located upstream of H4K10ac-rich ‘‘single’’ peaks, but not all single peaks are associated with tRNA genes.

Figure 2. Histone H4K10ac is enriched at probable pol II TSSs. (A) For a representative region of chromosome 10, the relative number of tags for histone H4K10ac (black) and histone H4 (gray) are calculated for a window size of 100 bp. Orange boxes represent ORFs and green arrows indicate direction of transcription. (B) H4K10ac-rich regions at non-SSRs are often located next to tRNA genes (blue boxes).

Considering the scale of the represented chromosome 10 in the Figure 2 showed above, tRNA cluster genes are located far away from H4K10ac. However, the fact that in trypanosomes tRNAs are located adjacent to chromatin domains where distinct histone marks (either variants or PTMs) were found, support our model whereby tRNA sequences in trypanosomes work as chromatin insulators or chromatin boundaries, similar to those described in other eukaryotes, as suggest our results and thus we described for the first time in this manuscript.

The manuscript still needs some attention, partly for accuracy and partly for the English. At least one Figure number citation seems to be wrong, and I didn't check them all. This can easily happen when a paper is modified.

We have revised the citation numbers as well as the English by a native speaker.

Dataset EV4 - the description on the third sheet is incorrect. It should say that this is the complete gene list. The fact that the authors take $FDR < 0.05$ to be significant can be stated separately.

To solve this inconvenience, we generated two additional data sheets including genes in Dataset EV4 showing $FDR < 0.05$ regarding to the data of the complete gene list.

Abstract:

"Further, overexpression of SNF2PH in the insect form induces expression of BF-specific genes, while a deletant lacking the PH domain did not." - tense change in the middle of the sentence

This now is been modified in the abstract as suggested

"clusters of tRNA genes, which functions as insulators" -> function

Page 4: *clusters of tRNA genes, which function as insulators* (Corrected).

"Specifically, it was identified a downregulation of single telomeric sequence" - needs correcting.

This sentence is now deleted since leads to confusion and was not proper English. As clearly shown in Figure 6E and F, overexpression of full-length SNF2PH induced expression of BES genes, while overexpression of PH-delta mutant did not, except for just one single BES related gene.

"In addition to induction of differentiation to the procyclic form after SNF2PH depletion in the BF (Fig

3B and Fig EV3), - Fig 3B is CHIP. Should this be Fig 6?

This sentence has been removed from the main text.

"Bloodstream pleomorphic trypanosomes undergo accurately differentiation" Remove "accurately".

(Also ungrammatical)

Page 20. *'Accurately'* was removed from the sentence.

"SNF2PH is negatively regulated during stumpy transition"

Page 20 *SNF2PH is negatively regulated during transition to stumpy form*

"SNF2PH is SUMO-modified to function as a transcriptional activator of VSG-ES monoallelic expression, while the PH-domain promotes bloodstream stage-specific gene expression and is required for the maintenance of the mammalian infective form" Change to:

Page 24-25 *"SNF2PH requires SUMO-modification to function as a transcriptional activator of VSG-ES monoallelic expression and the PH-domain is required for both this and for maintaining of the mammalian infective form surface protein coat, ensuring continuous VSG and ISG proper BF surface display, essential for pathogenicity"*. (Changed).

References

1. Organization of two invariant surface glycoproteins in the surface coat of *Trypanosoma brucei*. K Ziegelbauer, P Overath *Infection and Immunity* Nov 1993, 61 (11) 4540-4545; DOI:
2. D G Jackson, H J Windle, and H P Voorheis The identification, purification, and characterization of two invariant surface glycoproteins located beneath the surface coat barrier of bloodstream forms of *Trypanosoma brucei*. *J. Biol. Chem.* 1993 268: 8085-.
3. Allison H, O'Reilly AJ, Sternberg J, Field MC. An extensive endoplasmic reticulum-localised

glycoprotein family in
trypanosomatids. *Microb Cell.* 2014;1(10):325–345. doi:10.15698/mic2014.10.170.
4. Karl Ziegelbauer, Gloria Rudenko, Rudo Kieft, Peter Overath, Genomic organization of an
invariant surface
glycoprotein gene family of *Trypanosoma brucei*, *Molecular and Biochemical Parasitology*, Volume
69, Issue 1,
1995, Pages 53-63, ISSN 0166-6851, [https://doi.org/10.1016/0166-6851\(94\)00194-R](https://doi.org/10.1016/0166-6851(94)00194-R).

Corresponding Author Name: Miguel Navarro

Manuscript Number: EMBOR-2019-48029-T